# Enjoy Your Layer Normalization
# with the Computation Efficiency of RMSNorm

## Abstract

Layer normalization (LN) is a milestone technique in deep learning and has been widely adopted across various network architectures. However, LN introduces additional computational costs in the inference process. This issue has been addressed by its counterpart, RMSNorm, which removes the centering operation. This paper explores how to retain the theoretical advantages of LN while achieving the computational efficiency of RMSNorm. We first propose a general framework to determine whether an LN in any DNN can be equivalently replaced with RMSNorm. We introduce the methodology for removing the centering operation of LN after a linear layer with mathematical equivalence, by proposing column-based wight centering (CBWC) on linear layer. We further define the foldable LN—i.e., that can be replaced by RMSNorm without altering model behavior after applying constraints onto certain layers, and introduce zero-mean graph to analyze whether any LN in arbitrary given neural network is foldable. We present an algorithm that automatically detects foldable LNs and show that most LNs in currently widely used architectures are foldable, which provides a straightforward benefit in reducing the computational costs during inference. Additionally, we conduct extensive experiments to show that CBWC+RMSNorm achieves performance comparable to vanilla LN, while improving efficiency during training, even in cases where the LN is not foldable.

## 1 Introduction

Normalization techniques are widely used in deep neural networks (DNNs) to stabilize and accelerate training (Huang et al., 2023). As a seminar work, Batch Normalization (BN) (Ioffe & Szegedy, 2015) improves the training stability and the optimization efficiency of DNN by standardizing (centering and scaling) the activations of intermediate DNN layers within a mini-batch of data during training. During inference, BN uses population statistics calculated in training for normalization, and this operation can be folded into adjacent linear layers (Jacob et al., 2018), avoiding additional computational cost. Despite many merits, BN suffers from the train-inference inconsistent problem, leading to significantly degenerate performance in the scenarios of small batch sizes training and domain shifted distributions (Huang et al., 2023).

Layer normalization (LN) (Ba et al., 2016) addresses the train-inference inconsistent problem of BN by standardizing the inputs of layers within the neurons for each sample. It has become a key component of Transformer (Vaswani et al., 2017) and its variants (Dai et al., 2019; Xiong et al., 2020; Dosovitskiy et al., 2021), spreading from the Natural Language Processing (NLP) (Radford et al., 2018; Devlin et al., 2019; Raffel et al., 2020) to Computer Vision (CV) (Dosovitskiy et al., 2021; Carion et al., 2020; Cheng et al., 2022) communities. LN has a firm position (Huang et al., 2023) in the evolution of neural architectures and is currently a basic layer in most of the foundation models (Brown et al., 2020a; Alayrac et al., 2022; Kirillov et al., 2023). However, it has to perform additional standardization for each sample during inference, which introduces significant extra computational cost compared to the BN and normalization-free counterparts.

In order to alleviate the computational burden of LN, RMSNorm (Zhang & Sennrich, 2019) was proposed to perform a scaling-only operation and is reported to reduce the running time of LN by 7%∼64% on different models, according to the experiments in (Zhang & Sennrich, 2019). Despite its great potential in practice for computational efficiency and wide application in various architec-

tures (Zhang et al., 2024; Team et al., 2024; Mehta et al., 2024), RMSNorm is likely to miss the theoretical merits of the centering operation in improving conditioning, which has been widely investigated in previous work (LeCun et al., 1990; Schraudolph, 1998; Montavon & Müller, 2012; Huang et al., 2017). This raises the question of how we can exploit the theoretical advantages of LN while enjoy the computational cost of RMSNorm with mathematical equivalence. In a previous study (Jiang et al., 2023), the authors attempted to reduce LN to RMSNorm by removing the inherent redundant mean information in the main branch of Transformers, which stemmed from the pre-process and the residual branch. However, their method is based on pre-norm Transformers, leaving open the question of whether a more general analysis could be obtained to all types of DNNs.

In this work, we propose a general framework to analyze whether any LN can be replaced with RMSNorm, without changing the prediction of model for inference or the optimization dynamics for training. Firstly, we formally define the methodology for removing the centering operation of LN after a linear layer. We propose column-based weight centering (CBWC) (in Def. 2) which enforce a column-centered constraint (CCC) (in Def. 1) on the linear layer immediately preceding the LN *to guarantees the centering operation to be repetitive*. We demonstrate that the combined approach of CBWC+RMSNorm is an alternative training scheme mathematically equivalent to the vanilla model.

Extending our discussion to the whole neural network, we introduce the concept of a foldable LN (in Def. 3) and analyze its criteria in the perspective of graph. Specifically, an LN is foldable if all the leaf nodes of its zero-mean graph (in Def. 4) correspond to general linear layers. We show that most LNs in widely-used architectures are foldable, which provides a straightforward reduction in computational cost during inference.

Finally, we propose an algorithm to identify all foldable LNs in arbitrary DNN. Additionally, we conduct extensive experiments across 3 task families demonstrating that CBWC+RMSNorm can achieve comparable performance to LNs and provide greater training efficiency in long-sequence tasks in the same time—even in cases where the LN is not foldable.

*We highlight our contributions as follows. Our code is available at* `https://github.com/EnjoyYourLN/Enjoy-LN`.

- *We propose a general theoretically grounded framework that enables replacing LN with the cheaper RMSNorm with mathematical equivalence on arbitrarily networks, which provide 2% to 12% acceleration in both inference and training stage.*

- *Based on theoretical derivation, we propose CCC and CBWC that fold LN's centering into preceding linear layers, which does not affect the inference results or the training dynamics.*

- *We present automatic algorithm for detecting and folding qualified LN in arbitrarily neural network through the perspective of graph.*

## 2 NOTATION AND PRELIMINARY

We use $x \in \mathbb{R}$, $\mathbf{x} \in \mathbb{R}^n$, and $\boldsymbol{X} \in \mathbb{R}^{m \times n}$ to denote scalars, vectors, and matrices, respectively, where $\mathbb{R}$ is the set of real numbers, and $m$ and $n$ are positive integers. $\mathbf{1}_n$ stands for an $n$-dimensional all-one column vector.

**Neural Network.** We model a neural network as a function $f(\mathbf{x}; \boldsymbol{\theta})$, where $\mathbf{x}$ is the input and $\boldsymbol{\theta} \in \Theta$ denotes all learnable parameters, partitioned layer-wise as $\boldsymbol{\theta} = \{\boldsymbol{\theta}^{(k)}\}_{k=1}^{L}$. For an $L$-layer MLP, the forward pass consists of alternating linear transformations,[1] normalization, and activation:

$$\mathbf{h}^{(k)} = \boldsymbol{W}(\boldsymbol{\theta}^{(k)})\,\mathbf{x}^{(k-1)}, \tag{1}$$

$$\widehat{\mathbf{h}}^{(k)} = \mathrm{Norm}(\mathbf{h}^{(k)}) \tag{2}$$

$$\mathbf{x}^{(k)} = \phi(\widehat{\mathbf{h}}^{(k)}), \quad k = 1, ..., L, \tag{3}$$

---

[1]Following deep learning conventions, we do not differentiate between the linear and affine transformation. The bias term is omitted here for simplicity; we provide detailed inclusion in Appendix A.

where $\mathbf{x}^{(0)} = \mathbf{x}$ is the input, and $\mathbf{W}(\boldsymbol{\theta}^{(k)}) \in \mathbb{R}^{m \times n}$ is the weight matrix of layer $k$.[2]

**Layer Normalization.** Layer Normalization (LN) is a fundamental component in modern deep neural networks. Given a layer input vector $\mathbf{h} = [h_1, h_2, \ldots, h_n]^\top \in \mathbb{R}^n$, LN normalizes $\mathbf{h}$ across its $n$ elements (neurons) through centering and scaling: [3]

$$\text{Centering:} \quad \widetilde{h}_j = h_j - \mu, \quad j = 1, 2, \ldots, n, \tag{4}$$

$$\text{Scaling:} \quad \widehat{h}_j = \frac{\widetilde{h}_j}{\sqrt{\sigma^2 + \epsilon}}, \quad j = 1, 2, \ldots, n, \tag{5}$$

where $\mu = \frac{1}{n} \sum_{j=1}^{n} h_j$ is the mean of $\mathbf{x}$ and $\sigma^2 = \frac{1}{n} \sum_{j=1}^{n} \widetilde{h}_j^2$ is the second-order moment of the centered vector $\widetilde{\mathbf{h}} = [\widetilde{h}_1, \widetilde{h}_2, \cdots, \widetilde{h}_n]^\top$. *Centering* constrains the input to have zero mean, while *scaling* (i.e. RMSNorm) ensures unit second-order moment across elements.

*According to the definition, the calculation of RMSNorm is simpler than that of LN. While RMSNorm enjoys faster speed, it lacks constraints on the sample, leading to an unstable output range and significant changes in the sample values across layers (see Section 5.1 and Appendix F for details).*

In this paper, we aim to reduce the computational overhead of LN by replacing it with RMSNorm. However, a direct substitution may disrupt training dynamics and lead to performance degradation. To address this, in Section 3, we present a mathematically equivalent method for replacing LN with RMSNorm when LN directly follows a linear layer. In Section 4, we extend this analysis to arbitrary DNNs, identifying the conditions under which LN is foldable. Based on these insights, we propose an algorithm for the automatic detection and replacement of foldable LNs. Finally, in Section 5, we conduct comprehensive experiments to validate the effectiveness and efficiency of our approach.

## 3 EQUIVALENT REPLACEMENT OF LN AFTER LINEAR LAYERS

In this section, we present a theoretical analysis that simplifies LN when it is applied directly after a linear layer, by replacing it with RMSNorm. We propose a *column-centered constraint* (CCC) that enforce the linear layer to have zero-mean output, as shown in Figure 1. Furthermore, we introduce a *column-based weight centering* (CBWC) method ensure that linear layers satisfy CCC during training, which preserves mathematical equivalence between using RMSNorm and LN.

Figure 1: Overview of the method. $W$ denotes the weight matrix of a general linear layer, and $W^*$ is applied with CBWC which satisfies CCC.

### 3.1 COLUMN-CENTERED CONSTRAINT

Apparently, LN can be directly replaced by RMSNorm if the two produce mathematically equivalent results. Consider the common and simple case where LN is applied immediately after a linear layer: intuitively, the centering step of LN becomes redundant if the linear layer already produces outputs which have zero mean among all neurons. Since this zero-mean property must hold for every sample, we can enforce it directly by imposing a suitable constraint on the linear-layer parameters. We express the linear layer as the linear transformation $\mathbf{h}(\mathbf{x}; \boldsymbol{\theta}_k) = \boldsymbol{W}(\boldsymbol{\theta}_k)\mathbf{x}$ and denote the columns of $\boldsymbol{W}(\boldsymbol{\theta}_k)$ by $\mathbf{w}_i(\boldsymbol{\theta}_k) \in \mathbb{R}^m$, i.e., $\boldsymbol{W}(\boldsymbol{\theta}_k) = [\mathbf{w}_1(\boldsymbol{\theta}_k), \mathbf{w}_2(\boldsymbol{\theta}_k), \ldots, \mathbf{w}_n(\boldsymbol{\theta}_k)]$. Under this notation, we define *column-centered constraint* as follows.

**Definition 1** (Column-Centered Constraint (CCC))**.** *The parameter $\boldsymbol{\theta}_k \in \Theta^{(k)}$ of a linear layer satisfies the* column-centered constraint *if $\boldsymbol{\theta}_k$ satisfies:*

$$\boldsymbol{\theta}_k \in \Theta^* = \left\{ \boldsymbol{\theta} : \mathbf{w}_i(\boldsymbol{\theta})^\top \mathbf{1}_m = 0, \ i = 1, 2, \ldots, n \right\} \subseteq \Theta^{(k)}, \tag{6}$$

*i.e., the sum of the elements in each column vector $\mathbf{w}_i(\boldsymbol{\theta}_k)$ of the weight matrix is zero.*

---

[2]For convenience of notation, we simplify $\boldsymbol{\theta}^{(k)}$ to $\boldsymbol{\theta}_k$ in this paper.

[3]In practice, LN typically includes a learnable affine transformation after normalization, which is omitted here for simplicity. The term $\epsilon$ prevents division by zero.

**Proposition 1** (Zero-mean Output under CCC)**.** *The parameters of a linear layer $\boldsymbol{\theta}_k$ satisfy CCC, if for any input $\mathbf{x} \in \mathbb{R}^n$, the output $\mathbf{h} = \boldsymbol{W}(\boldsymbol{\theta}_k)\mathbf{x}$ has zero mean among all elements, i.e., $\mu_h = \frac{1}{m}\mathbf{1}_m^\top\mathbf{h} = 0$.*

Prop. 1 follows immediately from Def. 1; the full proof is given in Appendix B.1. By enforcing CCC on the linear layer, we fold the parameter space $\Theta^{(k)}$ into a subspace $\Theta^*$. Such that the linear layer effectively perform the centering step in advance, making the subsequent use of LN or RMSNorm equivalent.

**General Linear Layer**   To move beyond standard linear layers, we regard any layer that applies a linear transformation to its input as a general linear layer. Typical examples include recurrent layers with shared weights in RNNs and convolution layers in CNNs. Their parameters can be written in the unified form of Eqn. 1, and the corresponding CCC are derived in Appendix C.

## 3.2   COLUMN-BASED WEIGHT CENTERING

To enforce CCC, we introduce a re-parametrization technique.[4] We propose column-based weight centering, illustrated in Figure 2 and formally defined below.

**Definition 2** (Column-Based Weight Centering (CBWC))**.** Column-based weight centering *introduces a proxy parameter matrix $\boldsymbol{W}$ to control the effective weight matrix $\boldsymbol{V}$ via the transformation*

$$\boldsymbol{V} = \varphi(\boldsymbol{W}) = \left(\boldsymbol{I} - \frac{1}{m}\mathbf{1}_m\mathbf{1}_m^\top\right)\boldsymbol{W}, \tag{7}$$

*where $m$ is the number of output neurons. During backpropagation, the gradient is propagated as*

$$\frac{\partial\mathcal{L}}{\partial\boldsymbol{W}} = \varphi^\top\left(\frac{\partial\mathcal{L}}{\partial\boldsymbol{V}}\right) = \left(\boldsymbol{I} - \frac{1}{m}\mathbf{1}_m\mathbf{1}_m^\top\right)^\top\frac{\partial\mathcal{L}}{\partial\boldsymbol{V}}. \tag{8}$$

Note that CCC can be enforced in many ways. For example, a zero space $\Theta^* = \{\mathbf{0}\}$ satisfies Eqn. 6, yet destroys the feature representation. However, CBWC is a special transformation on the parameter, that ensures mathematical equivalence before and after the transformation.

**Proposition 2** (Equivalent Optimization Process)**.** *The optimization of an arbitrary model comprising a general linear layer followed by LN is equivalent to that of the model using a general linear layer parameterized via CBWC combined with RMSNorm.*

The proof of Prop. 2 is given in Appendix B.2. Once CBWC is applied to the linear layer, RMSNorm can replace the adjacent LN while producing identical outputs and gradients, yet at lower computational cost. The resulting CBWC+RMSNorm training protocol yields parameters that are mathematically equivalent to those of the original LN-based scheme, which is further discussed in Section 5.4. We discuss the acceleration in Section 5.2 and Appendix G.

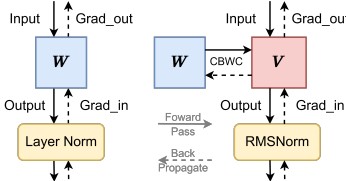

(a) Origin.   (b) CBWC+RMS.

Figure 2: Sketch map of the two training scheme

*Orthogonality with Optimization Techniques   From the perspective of any optimizer, learning-rate scheduler, gradient clipping, or weight-decay implementation which is applied on the weight matrix, CBWC+RMSNorm is completely indistinguishable from the original LN model.*

**Extension to Group Normalization**   CCC and CBWC on general linear layers can be generalized to group normalization (GN) (Wu & He, 2018), of which subsumes LN. The resulting grouped-CCC and grouped-CBWC are derived in Appendix C.3.

---

[4]Re-parametrization refers to transforming a model's parameter space to enforce desired constraints while preserving model capacity Nowlan et al. (1998).

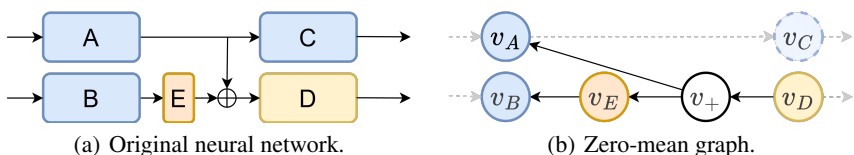

(a) Original neural network.      (b) Zero-mean graph.

Figure 3: An example network and the associated zero-mean graph. Layer D is an LN, layer E is a scalar operation, and $\oplus$ denotes residual addition. Layer D is foldable if and only if $v_A, v_B \in \mathcal{V}_l$.

## 4   A FRAMEWORK FOR FOLDING LN IN ARBITRARY DNN

In this section, we bridge the gap between theory and practice for replacing LN with RMSNorm. Firstly, we introduce the concept of foldable LN and analyze the conditions of foldable LN focusing on simple and common architectures such as sequential models and those with parallel connections. We then extend the analysis of foldable LNs to arbitrarily neural network through the perspective of graph. Finally, we propose an algorithm that automatically identifies foldable LNs and the corresponding general linear layers that require CBWC.

### 4.1   FOLDABLE LN IN SIMPLE AND COMMON CASES

We start by discussing about whether selected LN can be replaced with RMSNorm in a DNN. Intuitively, LN can be replaced by RMSNorm in the model if they can produce equivalent outputs regardless of the input. We formally define *foldable LN* as follows.

**Definition 3** (Foldable LN). *Let $f(\cdot; \boldsymbol{\theta})$ denote a subnetwork parameterized by $\boldsymbol{\theta} \in \Theta$, which directly connects to an LN. For a subset $\emptyset \neq \Theta^* \subseteq \Theta$, we say LN is* foldable *with respect to the parameter space $\Theta^*$ if for all $\boldsymbol{\theta}^* \in \Theta^*$ and all inputs $\mathbf{x} \in \mathbb{R}^n$, it holds that*

$$RMSNorm(f(\mathbf{x}; \boldsymbol{\theta}^*)) = LN(f(\mathbf{x}; \boldsymbol{\theta}^*)). \tag{9}$$

Due to the mathematical equivalence between CBWC+RMSNorm and the original LN model as proposed in Prop. 2, we only discuss the case when $\Theta^*$ is derived from CBWC in the following context for simplification. Extending from the analysis on simple scenario in Section 3, whether an LN is foldable is determined by the structure of the model, that is when the input of LN is produced by certain layers that can ensure zero-mean output, (i.e., general linear layers under CBWC). Therefore, we analyze the model by backtracking the data flow path and detect the layers that affect the input of LN.

**Sequential models**   Take MLPs as example. In this case, each LN is obviously foldable as they directly follows linear layer, which is independent of previous layers or input activations. In more complex scenarios, there may be additional layers between the general linear layer and the target LN. Notably, layer that involve only scalar operation (i.e., $\mathbf{x} \mapsto a\mathbf{x}$, where $a \in \mathbb{R}$) preserve the zero-mean property of the samples through it; these include scale layers (e.g., dropout layer in inference mode) and constant multiplication layers (e.g., temperature scaling). Therefore, the centering operation in LN can be folded into the parameters of previous linear transformation, regardless of the scalar operations interspersed between them.

**Parallel Connections**   Moving beyond sequential architectures, modern networks introduce more intricate connectivity, including parallel branches. Among them, the residual connection adds up multiple branches. To eliminate the mean of the output of the residual connection, we can eliminate the mean of each branch according to the distributive law of multiplication. So the subsequent LN can be folded by applying CBWC to each branch's general linear layers. However, other parallel connection, such as concatenation, breaks the mathematical equivalence, thus preventing the downstream LN from folding. We provide more detailed proof in Appendix D.1.

To systematically determine whether an LN is foldable in an arbitrary neural network, we analysis whether the original parameter space $\Theta$ can be effectively folded into the target parameter space $\Theta^*$ by CBWC in the next section.

## 4.2 ANALYSIS OF NEURAL NETWORK THROUGH THE PERSPECTIVE OF GRAPH

We model the neural network as a directed graph $G = (V, E)$. Each vertex $v \in V := \{f : \mathbb{R}^m \to \mathbb{R}^n, \forall m, n \in \mathbb{N}^+\}$ represents a layer, defined as any basic mathematical computations or transformations that applies on its input. The directed edges $E \subseteq V \times V$ indicate data flow: edge $\langle u, v \rangle$ indicates the output of $u$ is consumed as input by $v$.

We denote by $\mathcal{V}_l$ the subsets of vertices corresponding to general linear layers. The set of scalar operations is defined as $\mathcal{V}_s := \{f \mid \forall \mathbf{x} \in \mathbb{R}^m, f(\mathbf{x}) = a \cdot \mathbf{x}, \forall a \in \mathbb{R}\}$. Residual addition operations are represented by $\mathcal{V}_r := \{f \mid \forall \mathbf{x}_1, \mathbf{x}_2 \in \mathbb{R}^m, f(\mathbf{x}_1, \mathbf{x}_2) = \mathbf{x}_1 + \mathbf{x}_2, \}$. For layers that originally have zero-mean output, we define $\mathcal{V}_c := \{f \mid \forall \mathbf{x} \in \mathbb{R}^m, f(\mathbf{x}) \cdot \mathbf{1}_n = 0\}$. Other types of operations (i.e., $\mathcal{V}_- := \mathcal{V} \setminus (\mathcal{V}_l \cup \mathcal{V}_s \cup \mathcal{V}_r \cup \mathcal{V}_c)$), particularly nonlinear ones such as ReLU or softmax, typically disrupt the zero-mean property, thereby preventing the subsequent LN from being foldable. To determine whether an LN is foldable, we verify whether its input maintains zero mean due to parameter constraints imposed by preceding layers. We backtrack along the directed graph $G = (V, E(V))$ from the vertex $v_{LN}$ corresponding to the LN:

- If the predecessor vertex is a general linear layer ($v \in \mathcal{V}_l$), zero-mean can be ensured by applying CBWC.
- If it is a scalar operation ($v \in \mathcal{V}_s$), backtrack one step further.
- For residual addition vertices ($v \in \mathcal{V}_r$), backtrack through all input branches.
- If it is a layer originally has zero-mean output ($v \in \mathcal{V}_c$), it can fold the LN.
- If other type of vertex ($v \in \mathcal{V}_-$) is encountered, zero mean are not guaranteed, and thus LN folding fails.

To summarize the above method of judging a foldable LN, we present the *zero-mean graph*.

**Definition 4** (Zero-Mean Graph). *Given a neural network and its computation graph $G = (V, E)$, fix an target LN vertex $v_{LN} \in V$. We recursively construct a corresponding* zero-mean graph $G_z = (V_z, E_z)$ *as follows: We Initialize $V_0 = \{u \mid \langle u, v_{LN} \rangle \in E\}$ and $E_0 = \{\langle v_{LN}, u \rangle \mid \langle u, v_{LN} \rangle \in E\}$. For each iteration $k = 0, 1, 2 \cdots$:*

1. *Let $A_k = \{v \in V_k \mid v \in (V_r \cup V_s)\}$, the subset of the leaf layers.*

2. *Define $V_{k+1} = \bigcup_{v \in A_k} \{u \mid \langle u, v \rangle \in E\}$, by backtracking from the layers in $A_k$.*

3. *Define $E_{k+1} = \bigcup_{v \in A_k} \{\langle v, u \rangle \mid \langle u, v \rangle \in E\}$, by including reversed edges into the graph.*

*Define $V_z = \bigcup_{k=0}^{\infty} V_k, \ E_z = \bigcup_{k=0}^{\infty} E_k$.*

Notice that the leaf nodes of a zero-mean graph are either in $\mathcal{V}_l$, in $\mathcal{V}_c$ or in $\mathcal{V}_-$, we can easily find out that its root node LN is foldable if all the leaf nodes correspond to general linear layers or layers that ensures zero-mean output. Such that, we can replace the LN with RMSNorm with mathematical equivalence by applying CBWC on these general linear layers. We verified that all LNs in pre-norm and post-norm Transformers (Xiong et al., 2020) are foldable (see Appendix D).

## 4.3 ALGORITHM IN DETECTING FOLDABLE LN

Based on the above analysis, for arbitrary neural networks, we propose an algorithm to detect all the foldable LNs and corresponding layers that require constraint to ensure identical replacement between LN and RMSNorm. Instead of backtracking, the zero-mean graph can also be formed in the forward pass by marking states of the activation and the layers accordingly after passing through each layers. We present Algorithm 1.

Our *algorithm* can automatically detect foldable LNs in arbitrary neural network and identify the general linear layers that require CBWC to enable equivalent results after the replacement of LN with RMSNorm. We empirically tested Algorithm 1 on mainstream architectures including GPT-2 (Radford et al., 2019), BERT (Devlin et al., 2019), ViT (Dosovitskiy et al., 2021), Phi (Gunasekar et al., 2023), BLOOM (Scao et al., 2022), and OPT (Zhang et al., 2022). Remarkably, all LNs in these models were identified as foldable. We also fold most of LNs in VMamba (Liu et al., 2024). Other widely-used models such as LLaMA (Touvron et al., 2023) and Mamba2 (Dao & Gu, 2024)

---

**Algorithm 1** Detect foldable LNs and corresponding layers.

---

1: **Input:** Model $\mathcal{M}$ with input tensor $T_0^{\text{in}}$
2: **Output:** Set $\mathcal{S}$ of foldable LNs and set $\mathcal{C}$ of corresponding layers

3: $T_0^{\text{in}}.centered \leftarrow$ `False`                                       ▷ Set initial tensor state.
4: $\mathcal{S}, \mathcal{C} \leftarrow \emptyset$                   ▷ Initialize the set of foldable LNs and corresponding layers.

5: **for** each step $t$, layer $M_t \in \mathcal{M}$ **do**                              ▷ Iterate through each layer.
6:     **if** $T_t^{\text{in}} \leftarrow \sum_{i=1}^n T_i$ **then**                    ▷ Residual connection combining tensors.
7:         $T_t^{\text{in}}.centered \leftarrow \bigwedge_{i=1}^n T_i.centered$
8:         $T_t^{\text{in}}.corresponding \leftarrow \bigcup_{i=1}^n T_i.corresponding$                     ▷ New state and set.
9:     **end if**
10:    **if** $M_t = \text{LN} \wedge T_t^{\text{in}}.centered = $ `True` **then**
11:       $\mathcal{S} \leftarrow \mathcal{S} \cup \{M_t\}, \mathcal{C} \leftarrow \mathcal{C} \cup \{T_t^{\text{in}}.corresponding\}$               ▷ Mark the LN and the layers.
12:    **end if**

13:    $T_t^{\text{out}} \leftarrow M_t(T_t^{\text{in}})$                                      ▷ Get layer's output.
14:    **if** $M_t = $ *Scalar* **then**
15:       $T_t^{\text{out}}.centered \leftarrow T_t^{\text{in}}.centered$
16:       $T_t^{\text{out}}.corresponding \leftarrow T_t^{\text{in}}.corresponding$               ▷ Keep previous state and set.
17:    **else if** $M_t \in$ *General Linear Layers*, *LN* **then**
18:       $T_t^{\text{out}}.centered \leftarrow$ `True` , $T_t^{\text{out}}.corresponding \leftarrow M_t$             ▷ Update state and set.
19:    **else**
20:       $T_t^{\text{out}}.centered \leftarrow$ `False` , $T_t^{\text{out}}.corresponding \leftarrow \emptyset$            ▷ Initialize state and set.
21:    **end if**
22: **end for**

23: **Return:** The set of foldable LNs $\mathcal{S}$ and corresponding layers $\mathcal{C}$.

---

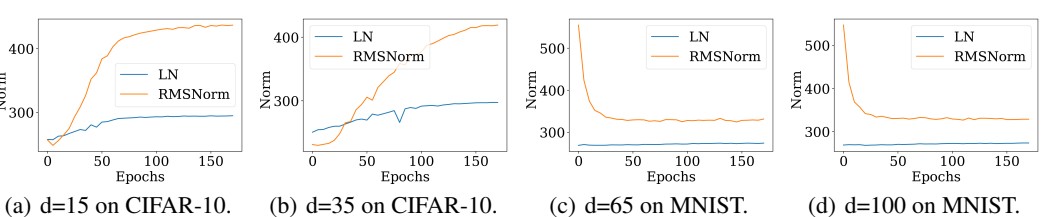

(a) d=15 on CIFAR-10.     (b) d=35 on CIFAR-10.     (c) d=65 on MNIST.     (d) d=100 on MNIST.

Figure 4: *Norm of the final layer's input for MLPs of different depth (d). LN better controls the norm of samples before the last layer during the training process.*

employ RMSNorm instead of LN by design. *We note that the runtime and memory overhead of our folding procedure is minimal, which can be covered after a few inference steps.* Further details are provided in Appendix E.1.

## 5 EXPERIMENTS

In this section, we demonstrate how to use a network with RMSNorm in practice while maintaining the theoretical advantages of LN for improving optimization conditioning Ba et al. (2016).

### 5.1 OBSERVATION OF CENTERING

Though the advantage of LN has not been universally agreed upon in the community, we observe that centering in LN helps stabilize the output range for a network. We conduct ablation experiments on MLPs of different depths using LN and RMSNorm under the classification task of CIFAR-10 Krizhevsky (2009) and MNIST LeCun et al. (1998). We monitor the parameters and input of each layer in each epoch. For detailed experiment settings, please refer to Appendix F.

## 5.2 ACCELERATION ANALYSIS OF INFERENCE

As shown in Figure 4, we find that LN better controls the norm of the last layer's input in a smaller range during the entire training process. In contrast, without the centering operation, the changes in input mean and norm are more intense. *We also report the change of input norm across all linear layers in appendix, showing the stabilizing role of LN's centering operation, particularly when residual connections are present.*

In inference, our method CBWC+RMSNorm achieves higher efficiency. Notably, CBWC only needs to be implemented once at the beginning for a pre-trained model, as the weight matrix does not update during inference. *Theoretically, our method reduces calculation usage by replacing LN with RMSNorm, resulting in an acceleration of about $50\%$ to $80\%$ in time usage compared to LN and an expected $4\%$ to $7\%$ runtime reduction for entire model.* More details are provided in Appendix G.1.

To verify the effectiveness of our method in practical applications, we conduct a comparison experiment between PyTorch's LN and our self-made CUDA-accelerated RMSNorm based on the Welford Algorithm on a single *A100-40G GPU*. We conducted experiments on *GPT-2 (Radford et al., 2019), Bert (Devlin et al., 2019), Bloom (Scao et al., 2022), OPT (Zhang et al., 2022) and Phi (Gunasekar et al., 2023). Our method reduces inference time by approximately 2% to 12% for a batch size of 2 and a sequence length of 1024 with zero accuracy degradation. as shown in Figure 5.*

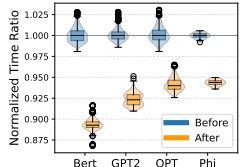 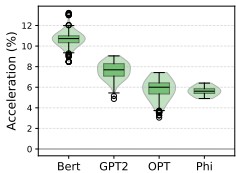

(a) *Normalized time.*    (b) *Acceleration ratio.*

Figure 5: *Comparison of inference latency across four representative models (GPT2, BERT, OPT and Phi3). Both the normalized time ratio and the acceleration percentage are reported relative to the mean baseline latency among each run.*

*Although the relative improvement appears modest, it translates into substantial absolute savings in energy and operational costs when deployed at scale. Additional experiments across varying batch sizes, sequence lengths and hardware (Appendix G.2) demonstrate that the acceleration remains robust throughout the practical operating range and even under higher memory intensity. These results confirm that CBWC+RMSNorm provides a reliable, drop-in performance enhancement with no loss in accuracy across diverse architectures and deployment scenarios.*

## 5.3 EMPIRICAL STUDY OF TRAINING

During training, CBWC+RMSNorm exceeds LN with its computational efficiency in Transformer models particularly for long-sequence tasks. Let $b$ denote the batch size, $s$ the sequence length, and $d$ the dimension of a word. Let the weight matrix be $\mathbf{W} \in \mathbb{R}^{d \times p}$. Training one epoch with $B$ samples, a centering operation incurs a computational cost of approximately $\mathcal{O}(Bsd)$, while CBWC introduces a cost of $\mathcal{O}(Bdp/b)$. CBWC is more efficient when $s \times b > p$, which is common in practical scenarios, especially for long-context learning.

Although we prove the theoretical equivalence between a foldable LN and CBWC+RMSNorm during training (in Prop. 2), the model architecture typically includes Dropout layers (Vaswani et al., 2017) between general linear layers and LN. Dropout in training mode disrupts the zero-mean property of the samples, leading to inequivalent results of model. Therefore, we conduct experiments to empirically validate the performance and effectiveness of our proposed CBWC+RMSNorm on a Transformer. We compare three variants: the baseline with LN, the variant with only RMSNorm, and the variant applying our method, CBWC+RMSNorm.

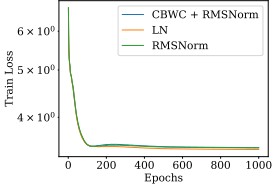 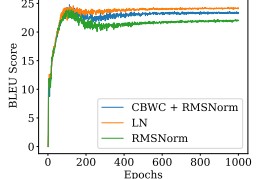

(a) Training loss.    (b) Evaluation BLEU.

Figure 6: *Training loss curves and validation BLEU scores of Transformer on the Multi30K translation task. Our proposed CBWC+RMSNorm achieves final performance between standard LN and vanilla RMSNorm.*

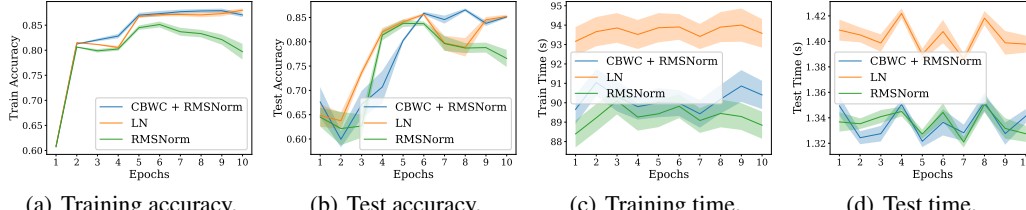

(a) Training accuracy.     (b) Test accuracy.     (c) Training time.     (d) Test time.

Figure 7: *Training and test performance of Transformer-based text classification on the AG News dataset (averaged over 5 random seeds with shaded regions indicating standard deviation). Our CBWC+RMSNorm matches the convergence behavior and final accuracy of standard LN while achieving nearly the same training and inference throughput as vanilla RMSNorm.*

**Text Translation** We train the models on Multi30K (Elliott et al., 2016) dataset, following the same experimental protocol as Vaswani et al. (2017). We measure training performance with loss value and evaluate bilingual evaluation understudy (BLEU) (Kishore Papineni & Zhu, 2002) scores in inference. Higher BLEU and lower loss values indicate better performance. All models are trained for 1000 epochs.

As shown in Figure 6, after the first 100 epochs, RMSNorm lags behind both the baseline and our method. In the end, while our method exhibits slightly inferior performance in terms of both training loss and BLEU scores compared to the baseline model, it still outperforms RMSNorm.

**Text Classification** We train on AG News (Zhang et al., 2015) dataset with the same experiment settings as the text translation task. We train the models for 10 epochs on 5 random seeds, evaluating both loss and accuracy. We measure the time usage.

We show the training loss and accuracy with standard deviation as shown in the shaded region in Figure 7. Notably, the baseline and our method CBWC+RMSNorm outperform RMSNorm during training, while RMSNorm has an unstable performance. In terms of efficiency, as shown in Figure 7, our method is faster than the baseline in both training and evaluation. The time usage is slightly inferior to RMSNorm in training due to re-parametrization costs despite the long sequence.

**Image Classification** We conduct image classification tasks based on SWIN-Tiny (Liu et al., 2021) on Imagenet100 (Chun-Hsiao Yeh, 2022). We measure the performances and evaluate the efficiency by measuring forward pass, backward propagation and validation time usage. All models are trained for 100 epochs and averaged by 3 random seeds. We provide experiment setting details and learning rate discussion in Appendix H.2.

Our method has a test performance outperforms the other two models. We list the results in Table 1 and show the best results in bold and the worst in gray. Moreover, we list the average time usage of each batch. As shown in Figure 8, our method has an improvement in time usage over the baseline for all three processes, and maintains a close time usage with RMSNorm in both forward pass stage and evaluation stage.

Table 1: *Image classification performance on ImageNet-100 using Swin-Tiny (averaged over 3 random seeds). Our method achieves the best results on both accuract and loss.*

| Method | Test Acc (%) ↑ | Test Loss ↓ |
|---|---|---|
| LN | $57.270 \pm 0.060$ | $3.352 \pm 0.015$ |
| RMSNorm | $57.079 \pm 0.317$ | $3.350 \pm 0.023$ |
| CBWC+RMS | $\mathbf{57.768 \pm 0.417}$ | $\mathbf{3.232 \pm 0.043}$ |

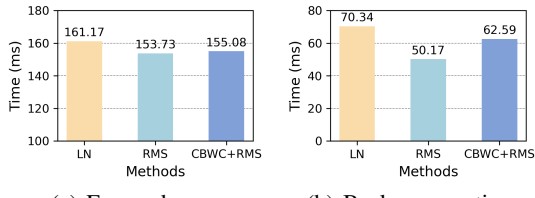

(a) Forward pass.     (b) Back propagation.

Figure 8: *End-to-end latency comparison of Swin-Tiny variants for different processes on ImageNet-100. Our CBWC+RMSNorm delivers markedly lower latency than LN in both forward and backward passes, approaching the efficiency of vanilla RMSNorm.*

**Summary**   In the case of non-equivalence, our method incurs a slightly better performance compared to RMSNorm and achieves an improvement in speed for long-sequence tasks compared to LN, approaching that of RMSNorm.

*__Trade-off__   It should be mentioned that the core goal of our method is to deliver substantial acceleration while preserving model behavior to the greatest extent possible. Therefore, under nonequivalent situation, the choice of whether to apply our method during inference or training involves a trade-off. We recommend using our method at inference time if strict equivalence and zero accuracy regression are required. Conversely, if the user prioritizes faster training and is comfortable with performance that typically lies between vanilla RMSNorm and LN (and occasionally exceeds LN), we recommend applying our method at training time.*

### 5.4 DISCUSSION ON FINE-TUNING

In practical applications, instead of training from scratch, we extensively use pre-trained models for fine-tuning. Due to the same optimization process of LN and CBWC+RMSNorm as mentioned in Prop. 2, the two sets of parameters are mathematically equivalent with the same weight matrix $W$. *We provide the setup and results of the verification experiment in Appendix H.4.1. Therefore, a model pre-trained with LN can theoretically continue to train with our method.*

We verify the feasibility of folding LN in fine-tuning stage on the Transformer structure. With the dropout layer, the two training scheme are not under the same optimization progress in the training stage. We conduct experiment under 3 different settings. After pre-training under LN, we apply our method, using CBWC+RMSNorm for training. We also convert the parameters with CBWC, continued training with RMSNorm. We compare the two models with the LN baseline. We pre-trained GPT-3 Medium (350M) Brown et al. (2020b) on Wikitext-103 Merity et al. (2016) for 3 epochs and fine-tune the models with Alpaca (Taori et al., 2023) for 1 epoch. We provide more detailed experiment settings in Appendix H.4.2.

Table 2: *Fine-tuning performance of GPT-3 Medium (350M) on the Alpaca instruction-following dataset after pre-training on WikiText-103. Our CBWC+RMSNorm achieves the best test loss and perplexity among all variants.*

| Method | Test Loss ↓ | Test PPL ↓ |
|---|---|---|
| LN | 0.2553 | 1.29 |
| RMSNorm | 0.2526 | 1.29 |
| CBWC+RMS | **0.2430** | **1.28** |

We use loss and perplexity (PPL) to evaluate the performance. The lower the perplexity, the less "confused" the model and the more accurate its predictions. The results are shown in Table 2. We show the best and the worst results in bold and gray, respectively. The models have better performance with our method while enjoying the efficiency. The modification on the parameter by our method does not affect the effect of pre-training.

## 6 CONCLUSION

This paper provided the general framework to detect whether any LN in an arbitrary DNN can be equivalently transferred to a network with RMSNorm by rigorous definition and derivation. We showed how the centering operation of LN can be replaced both in inference and training by introducing the proposed column-centered constraint (CCC) and column-based weight centering (CBWC). The proposed method can be directly applied to various pre-trained large language models (LLMs) and large vision language models (VLMs) with LN, enabling an immediate reduction in computational cost and with an equivalent forward pass during inference. The methodology is also effective in training and fine-tuning stage.

*__Limitation and Future Work__   In practical application scenarios, there are a large number of LLMs and VLMs with LN, but the number of large models we have analyzed in this paper is relatively small. In future work, we expect to have more detailed analysis of every models on how much does this method improve the effect. Moreover, in real neural networks, there are some modules that our method cannot implement, such as dropout layers. This may affect the effectiveness and utility of our simplification method, which is determined by the construction of the model.*

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

## A    BIAS IN THE WEIGHT MATRIX

In the main paper, we omit the bias for simplicity. Here, we provide the methodology for merging the bias into the weight matrix. For a linear layer $\mathbf{h} = \boldsymbol{W}\mathbf{x} + \mathbf{b}$, we denote the bias $\mathbf{b} = [b_1, b_2, \ldots, b_m]^\top \in \mathbb{R}^{m \times 1}$. For simplification, we add an additional dimension to $\mathbf{x}$, which turns $\mathbf{x}$ into $\mathbf{x}' = [x_1, x_2, \ldots, x_d, 1]^\top \in \mathbb{R}^{(d+1) \times 1}$, and adds an additional column in $\boldsymbol{W}' \in \mathbb{R}^{m \times (d+1)}$, where:

$$\boldsymbol{W}' = \begin{bmatrix} w_{1,1} & w_{1,2} & \cdots & w_{1,d} & b_1 \\ w_{2,1} & w_{2,2} & \cdots & w_{2,d} & b_2 \\ \vdots & \vdots & \ddots & \vdots & \vdots \\ w_{n,1} & w_{m,2} & \cdots & w_{m,d} & b_m \end{bmatrix}. \tag{10}$$

Therefore, we have $\mathbf{h} = \boldsymbol{W}\mathbf{x} + \mathbf{b} = \boldsymbol{W}'\mathbf{x}'$

In more general cases, for a linear transformation $f(\cdot; \boldsymbol{\theta}) : \mathbb{R}^n \to \mathbb{R}^m$, we similarly add an additional dimension to $\mathbf{x}' = [x_1, x_2, \cdots, x_d, 1]$, and a related weight vector $\mathbf{w}_{d+1} = \mathbf{b} = [b_1, b_2, \cdots, b_m]^\top \in \mathbb{R}^m$. Therefore, we have transformed expression:

$$\mathbf{h}(\cdot; \boldsymbol{\theta}) = \sum_{i=1}^{d} \mathbf{w}_i(\boldsymbol{\theta}) x_i + \mathbf{b} = \sum_{i=1}^{d+1} \mathbf{w}_i(\boldsymbol{\theta}) x_i, \tag{11}$$

such that we add the bias into the weight.

## B    PROOF OF PROPOSITIONS

### B.1    PROOF OF PROP. 1

*Proof.* For arbitrary input $\mathbf{x}$, the output is $\mathbf{h} = \sum_{i=1}^{n} \mathbf{w}_i(\boldsymbol{\theta}_k) x_i$. Taking the mean of $\mathbf{h}$ over the $m$ output neurons,

$$\mu_h = \frac{1}{m} \mathbf{1}_m^\top \mathbf{h} = \frac{1}{m} \sum_{i=1}^{n} x_i \underbrace{\mathbf{1}_m^\top \mathbf{w}_i(\boldsymbol{\theta}_k)}_{=0} = 0,$$

where the equality follows from CCC equation 6. $\qquad \square$

### B.2    PROOF OF PROP. 2

*Sketch.* Let model $A$ have a standard linear layer followed by LN, while model $B$ uses a linear layer with CBWC followed by RMSNorm. In model $A$, the centered output after LN can be written as $\widetilde{\mathbf{h}}_A = \left(\boldsymbol{I} - \frac{1}{m}\mathbf{1}_m\mathbf{1}_m^\top\right)\boldsymbol{W}_A\mathbf{x}_A$, where $\mathbf{x}_A$ is the input. In model $B$, the output is $\widetilde{\mathbf{h}}_B = \boldsymbol{V}_B\mathbf{x}_B = \left(\boldsymbol{I} - \frac{1}{m}\mathbf{1}_m\mathbf{1}_m^\top\right)\boldsymbol{W}_B\mathbf{x}_B$. If $\mathbf{x}_A = \mathbf{x}_B$ and $\boldsymbol{W}_A = \boldsymbol{W}_B$, then forward outputs are identical.

Since the parameters and outputs coincide, their gradients and losses coincide as well. Therefore, in model $A$, we have backpropagation process as:

$$\frac{\partial \mathcal{L}}{\partial \mathbf{h}_A} = \left(\boldsymbol{I} - \frac{1}{m}\mathbf{1}_m\mathbf{1}_m^\top\right)^\top \frac{\partial \mathcal{L}}{\partial \widetilde{\mathbf{h}}_A}, \qquad \frac{\partial \mathcal{L}}{\partial \mathbf{x}_A} = \boldsymbol{W}_A^\top \frac{\partial \mathcal{L}}{\partial \mathbf{h}_A}, \qquad \frac{\partial \mathcal{L}}{\partial \boldsymbol{W}_A} = \frac{\partial \mathcal{L}}{\partial \mathbf{h}_A} \mathbf{x}_A^\top. \tag{12}$$

When in model $B$, according to the definition of backward transformation $\boldsymbol{\psi}$ in CBWC, similarly we have:

$$\frac{\partial \mathcal{L}}{\partial \mathbf{x}_B} = \boldsymbol{V}_B \frac{\partial \mathcal{L}}{\partial \widetilde{\mathbf{h}}_B}, \qquad \frac{\partial \mathcal{L}}{\partial \boldsymbol{V}_B} = \frac{\partial \mathcal{L}}{\partial \widetilde{\mathbf{h}}_B} \mathbf{x}_B^\top, \qquad \frac{\partial \mathcal{L}}{\partial \boldsymbol{W}_B} = \left(\boldsymbol{I} - \frac{1}{m}\mathbf{1}_m\mathbf{1}_m^\top\right)^\top \frac{\partial \mathcal{L}}{\partial \boldsymbol{V}_B}. \tag{13}$$

It is easy to identify that the two backpropagation process are the same:

$$\frac{\partial \mathcal{L}}{\partial \mathbf{x}} = \left(\boldsymbol{I} - \frac{1}{m}\mathbf{1}_m\mathbf{1}_m^\top\right)^\top \boldsymbol{W}^\top \frac{\partial \mathcal{L}}{\partial \widetilde{\mathbf{h}}}, \qquad \frac{\partial \mathcal{L}}{\partial \boldsymbol{W}} = \left(\boldsymbol{I} - \frac{1}{m}\mathbf{1}_m\mathbf{1}_m^\top\right)^\top \frac{\partial \mathcal{L}}{\partial \widetilde{\mathbf{h}}} \mathbf{x}^\top. \tag{14}$$

Such that, CBWC+RMSNorm results in identical optimization process. $\qquad \square$

## C COLUMN-BASED WEIGHT CENTERING OF GENERAL LINEAR LAYERS

In this section, we introduce the column-based weight transformation (CCWT) and given the explicit definition of general linear layers. We provide how typical general linear layers can be converted into a linear transformation, including recurrent layers and convolution layers, and derive their corresponding CCC and CBWC. We further provide the grouped-CCC and grouped CBWC for group normalization.

### C.1 COLUMN-CENTERED WEIGHT TRANSFORMATION

To achieve the CCC of a general linear layer in inference and ensure a subsequent LN foldable, we propose *column-centered weight transformation*. The aim of CCC is to fold centering operation (in Eqn.4) into linear layer. Notice that given a layer input vector $\mathbf{x} = [x_1, x_2, \ldots, x_n]^\top \in \mathbb{R}^n$, the centering operation can be written into the form of $\widetilde{\mathbf{x}} = \left(\boldsymbol{I} - \frac{1}{m}\mathbf{1}_m^\top \mathbf{1}_m\right)\mathbf{x}$, where the matrix $\left(\boldsymbol{I} - \frac{1}{m}\mathbf{1}_m^\top \mathbf{1}_m\right)$ can be moved into the linear layer. We have the definition below.

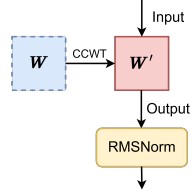

Figure 9: Sketch map of CCWT.

**Definition 5** (Column-Centered Weight Transformation (CCWT)). Column-centered weight transformation *aims to apply transformation on the weight matrix to ensure column-centered constraint. We construct a specific transformation $\varphi$, changes $\boldsymbol{W}$ into $\boldsymbol{W}'$, as:*

$$\boldsymbol{W}' = \varphi(\boldsymbol{W}) = \left(\boldsymbol{I} - \frac{1}{m}\mathbf{1}_m^\top \mathbf{1}_m\right)\boldsymbol{W}, \tag{15}$$

*where $m$ is the output neuron number.*

Apparently, CCWT ensures that the weight for each input in the transformed matrix $\boldsymbol{W}'$ has zero mean, thus guarantees CCC and ensures a zero-mean output of the layer. For different general linear layers, the transformation $\Psi$ takes different forms, but the essence of its construction based on column-centered constraint will not change.

CBWC is a special transformation on the parameter, folding the parameter space onto that of CCC while maintaining mathematical equivalence of the original model. With the construct of CCWT, it will be more easier to get the form of CBWC. For a given CCWT, we transfer it into CBWC by using a proxy weight in the place of $\boldsymbol{W}'$. Instead of directly replacing the computational weight $\boldsymbol{V}$, we ensure $\boldsymbol{V}$ is under column-centered constraint while $\boldsymbol{W}$ is calculated from $\boldsymbol{V}$ by CCWT.

### C.2 GENERAL LINEAR LAYER

In the main paper, we introduce the concept of general linear layer, which apply linear transformation on the inputs. Recurrent layers with shared weights in RNNs and convolution layers in CNNs are all general linear layers as they satisfy additivity and homogeneity. We formalize the definition of general linear layer as follows.

**Definition 6** (General Linear Layers). *A layer parameterized by $\boldsymbol{\theta}$ with transformation $\mathbf{h}(\mathbf{x}; \boldsymbol{\theta})$, where $\mathbf{x} \in \mathbb{R}^n$, is a* general linear layer *if the mapping $\mathbf{h}(\cdot; \boldsymbol{\theta})$ satisfies additivity and homogeneity:*

- $\mathbf{h}((\mathbf{x}_1 + \mathbf{x}_2); \boldsymbol{\theta}) = \mathbf{h}(\mathbf{x}_1; \boldsymbol{\theta}) + \mathbf{h}(\mathbf{x}_2; \boldsymbol{\theta}), \forall \mathbf{x}_1, \mathbf{x}_2 \in \mathbb{R}^n$.
- $\mathbf{h}(c\mathbf{x}; \boldsymbol{\theta}) = c\,\mathbf{h}(\mathbf{x}; \boldsymbol{\theta}), \forall\, \mathbf{x} \in \mathbb{R}^n,\ c \in \mathbb{R}$.

*Namely, the transformation $\mathbf{h}(\mathbf{x}; \boldsymbol{\theta})$ satisfies linearity, and is a linear transformation of $\mathbf{x}$.*

Such layers can be explicitly expressed as the form of $\mathbf{h}(\mathbf{x}; \boldsymbol{\theta}) = \boldsymbol{W}(\boldsymbol{\theta})\mathbf{x}$, allowing for direct application of CCC and CBWC.

#### C.2.1 RECURRENT LAYER

Despite linearity, the recurrent neural network is different from the original linear layer with its recurrent connection and shared weight matrix. Due to the fact that our constraints are independent

of input and output, the parameter sharing is excluded from our consideration. As for the recurrent connection, the weights for ordinary input and recurrent input can be seen as two linear layers.

For the $l$-th layer and $t$-th time step of a recurrent neural network, we define the input as $\mathbf{x}_t^{l-1} \in \mathbb{R}^{d_{l-1}}$, the recurrent input as $\mathbf{h}_{t-1}^l \in \mathbb{R}^{d_l}$ and the output of the hidden layer as $\mathbf{c}_t^l$. We have the weight matrix $\boldsymbol{W}_v = [\mathbf{w}_1^v, \cdots, \mathbf{w}_{d_{l-1}}^v] \in \mathbb{R}^{d_l \times d_{l-1}}$ and $\boldsymbol{W}_h = [\mathbf{w}_1^h, \cdots, \mathbf{w}_{d_l}^h] \in \mathbb{R}^{d_l \times d_l}$, where $\mathbf{w}_i^v, \mathbf{w}_i^h \in \mathbb{R}^{d_{l-1}}$ and which is shared between all time steps. We define $\boldsymbol{W} = [\boldsymbol{W}_v, \boldsymbol{W}_h]$ and input $\mathbf{x} = [(\mathbf{x}_t^{l-1})^\top, (\mathbf{h}_{t-1}^l)^\top]^\top$. We have recurrent layer in form of a linear transformation as follows:

$$\mathbf{c}_t = \mathbf{h}_{\mathrm{rnn}}(\mathbf{x}; \boldsymbol{W}) = \boldsymbol{W}_v \mathbf{x}_t^{l-1} + \boldsymbol{W}_h \mathbf{h}_{t-1}^l, \tag{16}$$

where we have $\mathbf{w}_i(\boldsymbol{W}) = \mathbf{w}_i^v$ when $i = 1, \cdots, d_{l-1}$ and $\mathbf{w}_i(\boldsymbol{W}) = \mathbf{w}_{i-d_{l-1}}^h$ when $i = (d_{l-1} + 1), \cdots, (d_{l-1} + d_l)$. Therefore, we have the constraint:

$$\boldsymbol{W}_0 \in \Theta_{\mathrm{rnn}} = \Big\{ \, \boldsymbol{W} : (\mathbf{w}_j^v)^\top \cdot \mathbf{1}_{d_l} = \sum_{i=1}^{d_l} w_{i,j}^v = 0, \ (\mathbf{w}_j^h)^\top \cdot \mathbf{1}_{d_l} = \sum_{i=1}^{d_l} w_{i,k}^h = 0,$$
$$j = 1, 2, \ldots, d_{l-1}, \ k = 1, 2, \ldots, d_l \, \Big\}. \tag{17}$$

Under the constraint of Eqn. 17, we have the mean of output with:

$$\mu_t^c = \frac{1}{d_l} \sum_{i=1}^{d_l} \left( \sum_{j=1}^{d_{l-1}} w_{i,j}^v x_j + \sum_{k=1}^{d_l} w_{i,k}^h h_k \right)$$
$$= \frac{1}{d_l} \left( \sum_{j=1}^{d_{l-1}} \left( \sum_{i=1}^{d_l} w_{i,j}^v \right) x_j + \sum_{k=1}^{d_l} \left( \sum_{i=1}^{d_l} w_{i,k}^h \right) h_k \right) \tag{18}$$
$$= \frac{1}{d_l} \left( \sum_{j=1}^{d_{l-1}} 0 \cdot x_j + \sum_{k=1}^{d_l} 0 \cdot h_k \right) = 0.$$

Thus, for the shared weight matrix for both input from the last layer and from the last time step, applying constraints on them centralizes the output of the hidden layer. We have the transformation $\varphi_{\mathrm{rnn},v}, \varphi_{\mathrm{rnn},h}$ of the CCWT on the recurrent neural network, as follows:

$$\boldsymbol{W}'^v = \varphi_{\mathrm{rnn},v}(\boldsymbol{W}^v) = (I - \frac{1}{m_v} \mathbf{1}_{m_v} \mathbf{1}_{m_v}^\top) \boldsymbol{W}^v$$
$$\boldsymbol{W}'^h = \varphi_{\mathrm{rnn},h}(\boldsymbol{W}^h) = (I - \frac{1}{m_h} \mathbf{1}_{m_h} \mathbf{1}_{m_h}^\top) \boldsymbol{W}^h. \tag{19}$$

We have the corresponding CBWC for the recurrent layer as below:

$$\boldsymbol{V}^v = \varphi_{\mathrm{rnn},v}(\boldsymbol{W}^v) = (I - \frac{1}{m_v} \mathbf{1}_{m_v} \mathbf{1}_{m_v}^\top) W^v,$$
$$\boldsymbol{V}^h = \varphi_{\mathrm{rnn},h}(\boldsymbol{W}^h) = (I - \frac{1}{m_h} \mathbf{1}_{m_h} \mathbf{1}_{m_h}^\top) \boldsymbol{W}^h,$$
$$\frac{\partial \mathcal{L}}{\partial \boldsymbol{W}^v} = \phi_{\mathrm{rnn},v} \left( \frac{\partial \mathcal{L}}{\partial \boldsymbol{V}} \right) = \left( \boldsymbol{I} - \frac{1}{m_v} \mathbf{1}_{m_v} \mathbf{1}_{m_v}^\top \right)^\top \frac{\partial \mathcal{L}}{\partial \boldsymbol{V}}, \tag{20}$$
$$\frac{\partial \mathcal{L}}{\partial \boldsymbol{W}^h} = \phi_{\mathrm{rnn},h} \left( \frac{\partial \mathcal{L}}{\partial \boldsymbol{V}} \right) = \left( \boldsymbol{I} - \frac{1}{m_h} \mathbf{1}_{m_h} \mathbf{1}_{m_h}^\top \right)^\top \frac{\partial \mathcal{L}}{\partial \boldsymbol{V}}.$$

### C.2.2 CONVOLUTION LAYER

Under the circumstances of the convolution layer, the convolution kernel can be regarded as a combination of a set of shared weights. All of them should fulfill the constraint of the linear layer. We hence use a vector to denote the elements among different channels among the kernel.

We denote the input tensor $\mathbf{x} \in \mathbb{R}^{d_{l-1} \times h \times w}$ and the output tensor $\boldsymbol{H} \in \mathbb{R}^{d_l \times h' \times w'}$. We have convolution kernels $\boldsymbol{W} \in \mathbb{R}^{d_l \times d_{l-1} \times F_h \times F_w}$.

For every channel of output tensor $\boldsymbol{H}_i \in \mathbb{R}^{h' \times w'}$ $(i = 1, \ldots, d_l)$, every channel of input tensor $\mathbf{x}_j \in \mathbb{R}^{h \times w}$ $(j = 1, \ldots, d_{l-1})$ and corresponding convolution kernel $\mathbf{w}_{i,j} \in \mathbb{R}^{F_h \times F_w}$ $(i = 1, \ldots, d_l, \ j = 1, \ldots, d_{l-1})$, we have:

$$\boldsymbol{H}_k = \sum_{t=1}^{d_{l-1}} \mathbf{x}_t * \mathbf{w}_{k,t}. \tag{21}$$

For more specifics, we have:

$$h_{k,i,j} = \sum_{t=1}^{d_{l-1}} \sum_{a=1}^{F_h} \sum_{b=1}^{F_w} x_{t,(is-s-p+a-1),(js-s-p+b-1)} \cdot w_{k,t,a,b}, \tag{22}$$

where the subscript sequence refers to the order number of the element in each dimension. Obviously, this equation can be written into the form of a linear transformation:

$$\boldsymbol{H} = \mathbf{h}_{\text{cnn}}(\mathbf{x}; \boldsymbol{W}) = \sum_{t=1}^{d_{l-1}} \sum_{a=1}^{h} \sum_{b=1}^{w} \mathbf{w}_{t,a,b}(\boldsymbol{W}) \cdot x_{t,a,b}. \tag{23}$$

Therefore, we have the constraint:

$$\boldsymbol{W}_0 \in \Theta_{\text{cnn}} = \left\{ \boldsymbol{W} : \sum_{i=1}^{d_l} \mathbf{w}_{i,j} = \mathbf{0}, \ j = 1, 2, \ldots, d_{l-1} \right\}. \tag{24}$$

Due to the convolution operation, we have $a * (b + c) = a * b + a * c$. Thus, under the constraint of Eqn.24, we have:

$$\begin{aligned}
\mu_h &= \frac{1}{d_l} \sum_{i=1}^{d_l} \boldsymbol{H}_i = \frac{1}{d_l} \sum_{i=1}^{d_l} \sum_{j=1}^{d_{l-1}} \mathbf{x}_j * \mathbf{w}_{i,j} = \frac{1}{d_l} \sum_{j=1}^{d_{l-1}} \mathbf{x}_j * \left( \sum_{i=1}^{d_l} \mathbf{w}_{i,k} \right) \\
&= \frac{1}{d_l} \sum_{j=1}^{d_{l-1}} \mathbf{x}_j * \mathbf{0} = 0.
\end{aligned} \tag{25}$$

It thus can be seen that the column-centered constraint on the convolution kernels achieves the effect of the centering of the LN. We have the transformation $\varphi_{cnn}$ of the CCWT on convolution layers, as follows:

$$\boldsymbol{W} = \varphi_{\text{cnn}}(\boldsymbol{W}) = (I - \frac{1}{h \times w} \mathbf{1}_{h \times w}^\top \mathbf{1}_{h \times w}) \boldsymbol{W}. \tag{26}$$

To be noted, the tensor $\boldsymbol{W}$ here is a four-dimension tensor. The transformation here is to do centering on its second dimension.

We have a corresponding CBWC for the convolution layer as below:

$$\boldsymbol{V} = \varphi_{\text{cnn}}(\boldsymbol{W}) = (I - \frac{1}{h \times w} \mathbf{1}_{h \times w}^\top \mathbf{1}_{h \times w}) \boldsymbol{W}.$$

$$\frac{\partial \mathcal{L}}{\partial \boldsymbol{W}} = \phi_{\text{cnn}}\left( \frac{\partial \mathcal{L}}{\partial \boldsymbol{V}} \right) = \left( \boldsymbol{I} - \frac{1}{h \times w} \mathbf{1}_{h \times w} \mathbf{1}_{h \times w}^\top \right)^\top \frac{\partial \mathcal{L}}{\partial \boldsymbol{V}}. \tag{27}$$

### C.3 GROUPED COLUMN-CENTERED CONSTRAINT FOR GN

We extend the conclusion to Group Normalization (GN) (Wu & He, 2018). Group normalization is first defined on the channel dimension for convolution input $\boldsymbol{X} \in \mathbb{R}^{d \times h \times w}$. So the Group Normalization here is more similar to grouped Layer Normalization, with the definition below:

**Definition 7** (Group Normalization (GN)). *Suppose the number of groups is $g$, and $d = g \times c$. Let $\mathbf{x} = [\boldsymbol{z}_1^\top, \ldots, \boldsymbol{z}_g^\top]^\top$, where $\boldsymbol{z}_i = [z_{i1}, \ldots, z_{ic}]^\top, (i = 1, \ldots, g)$. Assuming $\mathbf{x} = [x_1, \ldots, x_d]^\top$, we denote $z_{ij} = x_{(i-1) \times c + j}$. Let $\hat{\mathbf{x}} = GN(\mathbf{x})$, where $GN(\cdot)$ denotes the group normalization. GN can be calculated by $\mu_i = (z_{i1} + \cdots + z_{ic})/c$, $\sigma_i^2 = [(z_{i1} - \mu_i)^2 + \cdots + (z_{ic} - \mu_i)^2]/c$, and then $\hat{z}_{ij} = (z_{ij} - \mu_i)/\sigma_i$. Thus, we have $\hat{\mathbf{x}} = [\hat{\boldsymbol{z}}_1^\top, \ldots, \hat{\boldsymbol{z}}_g^\top]^\top$, where $\hat{\boldsymbol{z}}_i = LN(\boldsymbol{z}_i), (i = 1, \ldots, g)$.*

For every input, we divide the neurons into groups and apply normalization in every group. Thus the centering step in this normalization is to ensure the output sum of all neurons in each group is zero. For a sampled input $\mathbf{h} = [h_1, h_2, \ldots, h_m]^\top$, for $g$ groups and $c$ channels in every group ($g \times c = m$), we have:

$$\mu_{hj} = \frac{1}{c} \sum_{i=1}^{c} h_{ji} = 0 \quad (j = 1, \ldots, g). \tag{28}$$

For a given general linear layer The parameter $\boldsymbol{\theta}_0$ is under column-centered constraint of GN if $\boldsymbol{\theta}_0$ satisfies:

$$\boldsymbol{\theta}_0 \in \Sigma = \left\{ \boldsymbol{\theta} : \mathbf{w}_i^\top(\boldsymbol{\theta}) \cdot \mathbf{1}_{(c, k \times c)} = 0, \ k = 1, 2, \ldots, n \right\}. \tag{29}$$

Take the linear layer $\mathbf{h} = \boldsymbol{W}\mathbf{x}$ as an example, we have the weight matrix $\boldsymbol{W} = [\mathbf{w}_1, \mathbf{w}_2, \cdots, \mathbf{w}_{d_{l-1}}] \in \mathbb{R}^{d_l \times d_{l-1}}$, where $\mathbf{w}_i \in \mathbb{R}^{d_l}, i = 1, \cdots, d_{l-1}$. Similarly, we have $\mathbf{w}_i(\boldsymbol{W}) = \mathbf{w}_i$. Therefore, the column-centered constraint for GN can be expressed as:

$$\boldsymbol{W}_0 \in \Sigma_{\text{GN}} = \left\{ \boldsymbol{W} : \sum_{k=1}^{c} w_{j,(k+c \times i)} = 0, \ i = 1, 2, \ldots, g, \ j = 1, 2, \ldots, d \right\}. \tag{30}$$

Given $\mathbf{h} = \boldsymbol{W}\mathbf{x}$, for the $i$-th neuron output $h_i$ in the $j$-th group of $\mathbf{h}$, we have:

$$h_i = \sum_{k=1}^{d} w_{i,j} \cdot x_j. \tag{31}$$

Under the constraint of Eqn. 30, we have:

$$\mu_{hj} = \frac{1}{c} \sum_{i=1}^{c} h_{ji} = \frac{1}{c} \sum_{i=1}^{c} \sum_{j=1}^{d} w_{i,j} \cdot x_j = \frac{1}{c} \sum_{j=1}^{d} \left( \sum_{i=1}^{c} w_{i,j} \right) x_j = \frac{1}{c} \sum_{j=1}^{d} 0 \cdot x_j = 0. \tag{32}$$

Thus, we replace the centering step of GN with a grouped column-centered constraint. To be mentioned, the core idea of designing a constraint is to ensure every group of input weight has zero mean.

For the transformation $\varphi_{\text{GN}}$ of the CCWT on a normal linear layer under GroupNorm, we have:

$$\boldsymbol{W} = \varphi_{\text{GN}}(\boldsymbol{W}) = (\boldsymbol{I} - \boldsymbol{A})W. \tag{33}$$

$A$ is a matrix that we construct with the equation below:

$$\boldsymbol{A} = \boldsymbol{I} - \frac{1}{c} \sum_{k=0}^{d-1} \mathbf{1}_{(c, k \times c)}^\top \mathbf{1}_{(c, k \times c)}, \tag{34}$$

where $\mathbf{1}_{(c, k \times c)}$ refers to a vector whose elements are all zero except that the $(k \times c)$-th element to $(k \times c + c)$-th element are ones. Specifically, $A$ is a matrix with its diagonal arrayed with $c \times c$ matrices of ones.

We have a corresponding grouped CBWC for the linear layer as below:

$$\boldsymbol{V} = \varphi_{\text{GN}}(\boldsymbol{W}) = (\boldsymbol{I} - \boldsymbol{A})\boldsymbol{W}$$
$$\frac{\partial \mathcal{L}}{\partial \boldsymbol{W}} = \phi_{\text{GN}} \left( \frac{\partial \mathcal{L}}{\partial \boldsymbol{V}} \right) = (\boldsymbol{I} - \boldsymbol{A})^\top \frac{\partial \mathcal{L}}{\partial \boldsymbol{V}}. \tag{35}$$

*It is worth mentioning that InstanceNorm (Ulyanov et al., 2016) is simply the special case where the number of groups g equals the number of channels.*

*In addition, folding fails when the number of parameters influencing a single output element is smaller than the group size g (which can occur in certain shallow or highly grouped settings). In such cases, the grouped column-centered constraint becomes over-constrained and cannot be satisfied.*

*Beyond GroupNorm, any normalization (or post-processing) that consists solely of additive (among elements of the sample) and scalar multiplication operations applied per sample can be folded into the preceding general linear layer using the same principle.*

## C.4 *Discussion of Column-Based Weight Centering and Weight Decay Technology*

*According to the definition of weight decay, this technology involves adding a penalty term based on the L2 norm of the model weights to the original loss function, thereby encouraging the optimizer to favor smaller weight values and improving the model's generalization performance.*

*In our method, the model contains two weight matrices. This is because CBWC is implemented by re parameterization: the matrix ($V$) used for forward/backward calculation is different from the updated parameter matrix ($W$), but the two matrices are connected by deterministic and differentiable transformations (as described in Definition 2). The L2 norm of the calculation matrix $V$ and the storage parameter matrix $W$ are slightly different. Therefore, two implementation methods of weight decay are derived from this transformation.*

*Standard implementations (including PyTorch) apply weight decay to the trainable parameters ($W$ in our case). However, whether weight decay should technically be applied to $V$ or $W$ is an interesting and subtle question. Applying it to $W$ preserves the classic regularization interpretation (penalizing the magnitude of stored parameters), while applying it to $V$ would more directly penalize the magnitude of the pre-activation weights.*

*Both choices are defensible, and the community has not reached a consensus of a better choice, which is also orthogonal to the core contribution of our paper. Our discussed implementation (decay on $W$) follows PyTorch's default behavior and does not alter the intended regularization effect in practice.*

# D  Algorithm for Detecting Foldable LN

## D.1  Parallel Connection

In this section, we prove how to fold an LN while the sample meets residual structure during backtracking and provide the reason why other parallel connection fails to fold a downstream LN.

### D.1.1  Residual Structure

Here we consider a two branch residual structure as an example. The conclusion remains unchanged when the number of branches increases. For a residual structure, we define the input $\mathbf{x} \in \mathbb{R}^n$ and the output $\mathbf{y}\mathbb{R}^m$, as shown below:

$$\mathbf{y} = \mathcal{F}(\mathbf{x}; \boldsymbol{\theta}_F) + \mathcal{G}(\mathbf{x}; \boldsymbol{\theta}_G), \tag{36}$$

where both $\mathcal{F}(\cdot; \boldsymbol{\theta}_F)$ and $\mathcal{G}(\cdot; \boldsymbol{\theta}_G)$ are subnetworks, and $\boldsymbol{\theta}_F$ and $\boldsymbol{\theta}_G$ are learnable parameters. Due to the complexity of $\mathcal{F}(\cdot)$, it is intuitively difficult to construct a constraint on this function to eliminate the mean of $\mathcal{G}(\mathbf{x})$. Therefore, we treat the two terms separately and apply constraint based on their content each.

To eliminate the mean of $\mathbf{y}$, we have $\mathbf{y}' = (I - \frac{1}{m}\mathbf{1}_m^\top \mathbf{1}_m)\mathbf{y}$. According to the distributive law of multiplication, we have:

$$\mathbf{y}' = (I - \frac{1}{m}\mathbf{1}_m^\top \mathbf{1}_m)\mathbf{y} = (I - \frac{1}{m}\mathbf{1}_m^\top \mathbf{1}_m)\mathcal{F}(\mathbf{x}; \boldsymbol{\theta}_F) + (I - \frac{1}{m}\mathbf{1}_m^\top \mathbf{1}_m)\mathcal{G}(\mathbf{x}; \boldsymbol{\theta}_G), \tag{37}$$

Such that, the residual structure has zero-mean output if the output of each branch has zero mean.

### D.1.2  Other Parallel Connection

Other parallel connection do not satisfy linearity, such that we cannot transfer the zero-mean property to each branches.

Take concatenation as an example. When all composite vectors for concatenation have zero mean, stacking vectors along the axis produces a zero-mean tensor. But this zero-mean property of composite vectors and output vectors are not mathematical equivalent in back propagation, thus preventing the downstream LN from folding.

To put this another way, for a concatenation operation, we define the input $\mathbf{x}_i \in \mathbb{R}^{n_i}, i = 1, \cdots, n$ and the output $\mathbf{y}\mathbb{R}^m$, where $m = \sum_{i=1}^n n_i$, as shown below:

$$\mathbf{y} = f(\mathbf{x}_1, \cdots, \mathbf{x}_n) = [\mathbf{x}_1, \cdots, \mathbf{x}_n]. \tag{38}$$

To eliminate the mean of $\mathbf{y}$, we have $\mathbf{y}' = (I - \frac{1}{m}\mathbf{1}_m^\top\mathbf{1}_m)\mathbf{y}$. It is obvious that:

$$\mathbf{y}' = (I - \frac{1}{m}\mathbf{1}_m^\top\mathbf{1}_m)\mathbf{y} \neq [(I - \frac{1}{m}\mathbf{1}_m^\top\mathbf{1}_m)\mathbf{x}_1, \cdots, (I - \frac{1}{m}\mathbf{1}_m^\top\mathbf{1}_m)\mathbf{x}_n]. \tag{39}$$

Such that zero mean of each composite vector does not equal to the zero mean of the output vector. Therefore, we cannot fold the downstream LN into the concatenation.

### D.2    A Stricter Criteria of Foldable LN

To ensure mathematical equivalence, we expect all the layers required CBWC to affect only LNs. Here we define the *affected layer*.

**Definition 8** (Affected layer). *Given a neural network $G = (V, E)$ and the zero-mean graph $G_z = (V_z, E_z)$ for $v_{LN}$. We initialize $V_0 = \{u \notin V_z \mid \forall v \in (V_z \setminus \mathcal{V}_{LN}), \langle v, u \rangle \in E\}$ and $A_0 = \emptyset$. For each iteration $k = 0, 1, 2, \cdots$:*

1. *Let $A_{k+1} = (V_k \cap (\mathcal{V}_c \cup \mathcal{V}_l \cup \mathcal{V}_-))$, update affected layers.*

2. *Let $V_{k+1} = \{u \mid v \in (V_k \setminus (\mathcal{V}_c \cup \mathcal{V}_l \cup \mathcal{V}_-)), \langle u, v \rangle \in E\}$, trace the next layer through forward pass.*

*We define $A = \bigcup_{k=0}^{\infty} A_k$ as affected layer.*

If the affected layers only include LN, then the zero-mean activation in the zero-mean graph safely ensures a foldable LN. We note that the commonly used models nowadays, such as transformers, all meet this requirement.

### D.3    Self-Attention Module and Transformer Structure

As proposed in the previous section, we can replace the LN with RMSNorm if the adjacent layers are all general linear layers under CCC. However, the zero-mean output will be propagated indiscriminately to all connected layers of the adjacent layers, which may lead to nonequivalent output and unexpected results.

#### D.3.1    Self-Attention Module

To be mentioned, self-attention module can be seen as a pile of layers. We make use of its posterior linear component and thus construct the constraint on it.

For a sampled input $\mathbf{x} \in \mathbb{R}^{n \times d}$, we apply three different learnable weight matrices $\mathbf{Q}, \mathbf{K} \in \mathbb{R}^{d \times d_k}$, $\mathbf{V} \in \mathbb{R}^{d \times d_v}$ and have three input matrices $\mathbf{H}_Q, \mathbf{H}_K \in \mathbb{R}^{n \times d_k}$, $\mathbf{H}_V \in \mathbb{R}^{n \times d_v}$ with:

$$\mathbf{H}_Q = \mathbf{x} \cdot \mathbf{Q}, \quad \mathbf{H}_K = \mathbf{x} \cdot \mathbf{K}, \quad \mathbf{H}_V = \mathbf{x} \cdot \mathbf{V}. \tag{40}$$

According to the definition of scaled dot-product attention, we have:

$$\text{Attention}(\mathbf{H}_Q, \mathbf{H}_K, \mathbf{H}_V) = \text{softmax}\left(\frac{\mathbf{H}_Q\mathbf{H}_K^\top}{\sqrt{d_k}}\right)\mathbf{H}_V. \tag{41}$$

When this expression is expanded, it is written as:

$$\text{Attention}(\mathbf{x}; \mathbf{Q}, \mathbf{K}, \mathbf{V}) = \text{softmax}\left(\frac{\mathbf{x} \cdot \mathbf{Q} \cdot \mathbf{K}^\top \cdot \mathbf{x}^\top}{\sqrt{d_k}}\right)\mathbf{x} \cdot \mathbf{V}. \tag{42}$$

To simplify, we denote:

$$\mathbf{B} = \text{softmax}\left(\frac{\mathbf{x} \cdot \mathbf{Q} \cdot \mathbf{K}^\top \cdot \mathbf{x}^\top}{\sqrt{d_k}}\right)\mathbf{x} \in \mathbb{R}^{n \times d}. \tag{43}$$

We can see this module as a linear transformation $\mathbf{h} = \mathbf{h}_{\text{trans}}(\mathbf{B}, \mathbf{V}) = \mathbf{B}\mathbf{V}$, where $\mathbf{B}$ is the input. We denote $\mathbf{B} = [\mathbf{b}_1^\top, \cdots, \mathbf{b}_d^\top]^\top$, where $\mathbf{b}_i^\top \in \mathbb{R}^n, i = 1, \cdots, d$ and $\mathbf{V} = [\mathbf{v}_1, \cdots, \mathbf{v}_d]^\top \in \mathbb{R}^{d \times d_v}$, where $\mathbf{v}_i \in \mathbb{R}^{d_v}, i = 1, \cdots, d$.

We have $\mathbf{h} = \mathbf{h}_{\text{trans}}(\boldsymbol{B}, \boldsymbol{V}) = \sum_{i=1}^{d} \mathbf{v}_i \cdot \mathbf{b}_i$ and $\mathbf{w}_i(\boldsymbol{V}) = \mathbf{v}_i$. Therefore, we have the constraint:

$$\boldsymbol{W}_0 \in \Gamma_{trans} = \left\{ \boldsymbol{W} : \mathbf{v}_i^\top \cdot \mathbf{1}_{d_v} = \sum_{k=1}^{d_v} v_{j,k} = 0, \ i = 1, 2, \ldots, d_{l-1} \right\}. \tag{44}$$

By Eqn. 44, we have

$$\mu_a = \sum_{b=1}^{d_v} (\boldsymbol{B}\boldsymbol{H}_V)_{(a,b)} = \sum_{b=1}^{d_v} \sum_{j=1}^{n} b_{a,j} \cdot \boldsymbol{H}_{V(j,b)} = \sum_{b=1}^{d_v} \sum_{j=1}^{n} b_{a,j} \left( \sum_{k=1}^{d} x_{j,k} \cdot v_{k,b} \right)$$

$$= \sum_{b=1}^{d_v} \sum_{j=1}^{n} \sum_{k=1}^{d} b_{a,j} \cdot x_{j,k} \cdot v_{k,b} = \sum_{j=1}^{n} \sum_{k=1}^{d} b_{a,j} \cdot x_{j,k} \left( \sum_{b=1}^{d_v} v_{k,b} \right) \tag{45}$$

$$= \sum_{j=1}^{n} \sum_{k=1}^{d} b_{a,j} \cdot x_{j,k} \cdot 0 = 0.$$

Accordingly, we have the transformation $\varphi_{\text{trans}}$ of the CCWT on self-attention modules, as follows:

$$\boldsymbol{W}^v = \varphi_{\text{trans}}(\boldsymbol{W}^v) = (I - \frac{1}{m_v}\mathbf{1}_{m_v}^\top\mathbf{1}_{m_v})\boldsymbol{W}^v. \tag{46}$$

We have a corresponding CBWC for the attention module as below:

$$\boldsymbol{W}^v = \varphi_{\text{trans}}(\boldsymbol{W}^v) = (I - \frac{1}{m_v}\mathbf{1}_{m_v}^\top\mathbf{1}_{m_v})\boldsymbol{W}^v.$$

$$\frac{\partial \mathcal{L}}{\partial \boldsymbol{W}} = \phi_{\text{trans}}\left( \frac{\partial \mathcal{L}}{\partial \boldsymbol{V}} \right) = \left( \boldsymbol{I} - \frac{1}{m_v}\mathbf{1}_{m_v}\mathbf{1}_{m_v}^\top \right)^\top \frac{\partial \mathcal{L}}{\partial \boldsymbol{V}}. \tag{47}$$

Moreover, multi-head attention modules contain an extra linear layer at the last. Simply applying CCC and CBWC of the linear layer onto it ensures zero-mean output.

### D.3.2 POST-LN TRANSFORMER

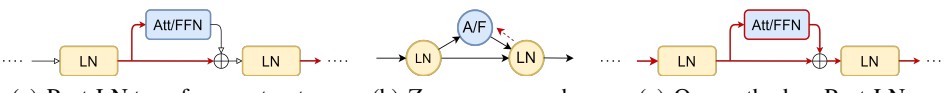

(a) Post-LN transformer structure. (b) Zero-mean graph. (c) Our method on Post-LN.

Figure 10: The proof and application of our method on Post-LN. 'Att' and 'FFN' refer to the Attention layer and the Feed-Forward Network, which are both general linear layers.

For the post-LN transformer, the residual structure connects an LN and an self-attention module or a feed-forward network layer. Obviously, all the LNs are foldable.

### D.3.3 PRE-LN TRANSFORMER

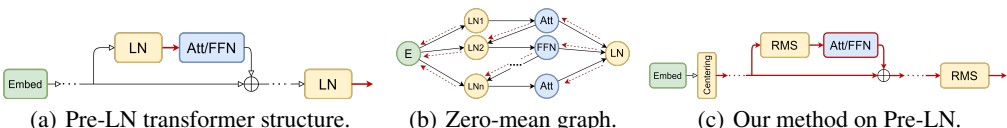

(a) Pre-LN transformer structure. (b) Zero-mean graph. (c) Our method on Pre-LN.

Figure 11: The proof and application of our method on Pre-LN. 'Att' and 'FFN' refers to the Attention layer and Feed-Forward Network, which are both general linear layers. 'Embed' refers to the Embedding layer, which it is not.

Notably, for layers $v \in V_-$, zero-mean output can still be maintained by inserting an explicit additional centering operation after them. Such insertion can form following LN foldable while introducing extra computational cost, but is still efficient if the number of folded LNs formed is larger than the number of centering operations added.

For most models like GPT2, there is an LN after the last transformer block. Therefore, all the affected layers of adjacent layers of LN are LN. Despite the embedding block fails to provide zero-mean output, we can add an extra centering after it. Thus, all the LN can be replaced if we apply CBWC on all the attention layers and feed-forward network layers. Thus, we prove that the method makes all of the LN foldable. This is also the main idea of the previous paper Jiang et al. (2023).

## E  *Implementation Details of Algorithm 1*

### E.1  FOLDABLE LN IN COMMON MODELS

Here, we list 11 common models and the number of LN and foldable LN in Table 3. Notice that some models are pre-norm Transformer structure and the number of foldable LN changed after we add an extra centering operation after embedding. The models we mentioned here are GPT-2 (Radford et al., 2019), BERT (Devlin et al., 2019), ViT (Dosovitskiy et al., 2021), Phi (Gunasekar et al., 2023), OPT (Zhang et al., 2022), BLOOM (Scao et al., 2022) and VMamba (Liu et al., 2024).

Table 3: Number of LN and foldable LN in 11 common models.

| Model | Total | Foldable | Percentage | Foldable (Embed-Centered) | Percentage |
|-------|-------|----------|------------|---------------------------|------------|
| GPT-2 | 25 | 0 | 0 | 25 | 100.00% |
| BERT | 25 | 24 | 96.00% | 25 | 100.00% |
| ViT | 25 | 0 | 0 | 25 | 100.00% |
| Phi | 25 | 0 | 0 | 25 | 100.00% |
| OPT | 25 | 0 | 0 | 25 | 100.00% |
| BLOOM | 6 | 5 | 83.33% | 6 | 100.00% |
| VMamba | 51 | 0 | 0 | 36 | 70.59% |

To be mentioned, Phi3 (Abdin et al., 2024), Qwen2 (Yang et al., 2024), T5 (Raffel et al., 2023), Mamba2 (Dao & Gu, 2024) and LLaMA (Touvron et al., 2023) originally uses RMSNorm, instead of LN.

### E.2  *Runtime and Memory Overhead*

*The runtime and memory overhead of our automatic folding algorithm is minimal. The detection phase requires only one forward pass to determine which LN layers are foldable, after which we store a small list mapping foldable layers to their corresponding modules.*

*Here, we conduct extensive experiment on different models, hardware, batch size and sequence lengths. All inference comparisons use PyTorch's official fused CUDA LayerNorm as the baseline, and our optimized CUDA RMSNorm kernel derived from it, which we have open-sourced.*

*We first conduct a confirmatory experiment on GPT-2 using a single RTX 3050 Laptop GPU. The folding algorithm takes 0.0347 seconds. For inference with batch size 32 and sequence length 512, the inference time is reduced from 0.2189 s to 0.2023 s (averaged over 100 runs), yielding a 7.6% speedup. Notably, the memory overhead introduced by the folding algorithm itself is only 19.00 MB (which is released after folding), and the total tensor memory of the model remains unchanged (486.70 MB both before and after folding). These results demonstrate that our method achieves meaningful inference acceleration without any increase in model memory footprint.*

*In order to make the conclusion more extensive, we conduct more experiments. We record time usage and memory usage for Bert, GPT2, OPT and Phi, on A100-40GB and V100-32GB. The batch size and sequence length for inference are 1 and 256 respectively. Memory footprint remains identical before and after folding. We list the results in the tables below:*

Table 4: Overhead and benefits of folding algorithm on A100-32G GPU.

| Model | Fold (s) | Init (s) | Ratio (%) | Before (s) | After (s) | Speedup (%) | Break-even |
|-------|----------|----------|-----------|------------|-----------|-------------|------------|
| Bert  | 0.0452   | 2.4056   | 1.84      | 0.0077     | 0.0073    | 5.19        | 112.9      |
| GPT2  | 0.0651   | 3.0651   | 2.08      | 0.0119     | 0.0113    | 5.22        | 105.0      |
| OPT   | 0.0484   | 2.6283   | 1.81      | 0.0092     | 0.0078    | 14.60       | 36.1       |
| Phi   | 0.0928   | 22.2692  | 0.42      | 0.0531     | 0.0507    | 4.52        | 38.7       |

Table 5: Overhead and benefits of folding algorithm on V100-32G GPU.

| Model | Fold (s) | Init (s) | Ratio (%) | Before (s) | After (s) | Speedup (%) | Break-even |
|-------|----------|----------|-----------|------------|-----------|-------------|------------|
| Bert  | 0.1263   | 2.0772   | 5.73      | 0.0095     | 0.0086    | 9.47        | 140.4      |
| GPT2  | 0.1639   | 2.6343   | 5.86      | 0.0115     | 0.0110    | 4.34        | 327.8      |
| OPT   | 0.1183   | 2.2988   | 4.90      | 0.0106     | 0.0091    | 14.73       | 75.6       |
| Phi   | 0.1779   | 21.8865  | 0.81      | 0.0693     | 0.0668    | 3.54        | 72.6       |

(a) Fold = Folding time. (b) Init = Model initialize time. (c) Ratio = Fold / (Fold + Init) × 100%. (d) Before = Inference latency without folding. (e) After = Inference latency with algorithm applied. (f) Speedup = (Before - After) / Before × 100%. (g) Break-even = Fold / (Before - After).

*The one-time folding algorithm (∼60–100 ms on A100, ∼100-200 ms on V100) is fully amortized after only a few inference steps in any real deployment. Importantly, larger batch sizes and longer sequences achieve greater acceleration with our algorithm in inference, thus requiring fewer runs to break even on the algorithm cost.*

*Moreover, time usage of folding algorithm is theoretically independent of batch size and sequence length. We conduct experiments on Bert on V100, evaluated the algorithm time usage across six batch sizes (4 to 128) and four sequence lengths (64 to 4096). The result is list below.*

Table 6: Folding algorithm time usage (ms) across different batch sizes (BS) and sequence lengths (SL).

| BS\SL   | 64    | 256   | 1024  | 4096  | Average | Std  | CV (%) |
|---------|-------|-------|-------|-------|---------|------|--------|
| 4       | 132.3 | 132.8 | 133.7 | 138.4 | 134.3   | 2.77 | 2.06   |
| 8       | 137.1 | 132.9 | 133.9 | 136.9 | 135.2   | 2.11 | 1.56   |
| 16      | 139.3 | 134.8 | 134.9 | 136.0 | 136.3   | 2.11 | 1.55   |
| 32      | 134.6 | 135.5 | 136.1 | 137.6 | 135.9   | 1.26 | 0.92   |
| 64      | 142.7 | 136.7 | 138.3 | 142.0 | 139.9   | 2.86 | 2.04   |
| 128     | 135.9 | 132.1 | 136.0 | 145.1 | 137.3   | 5.51 | 4.01   |
| Average | 137.0 | 134.2 | 135.5 | 139.3 | 136.5   | \    | \      |
| Std     | 3.65  | 1.80  | 1.73  | 3.50  | \       | 3.28 | \      |
| CV (%)  | 2.66  | 1.34  | 1.28  | 2.51  | \       | \    | 2.40   |

*As shown in Table 6, despite a 2048-fold increase in the total number of processed tokens, the standard deviation of algorithm time usage across all 24 configurations is only 3.3 ms, with a co-efficient of variation of merely 2.40%. Both marginal averages (per row and per column) and their corresponding CV values remain below 4%, confirming that algorithm time usage is essentially independent of both batch size and sequence length in practical deployment scenarios.*

## F    ABLATION EXPERIMENT OF CENTERING OPERATION

For ablation experiment of centering operation in LN *in Section 5.1*, we build up simple MLPs by stacking linear layers, LNs and ReLUs. The depth of MLP (i.e., the number of linear layers) varies among 6, 15, and 35 on CIFAR-10 Krizhevsky (2009), with a width of 256 and 512. We introduce the residual structure to help converge for deeper MLP with a depth of 65 and 100 on MNIST LeCun et al. (1998), with a width of 512. We train the model with a learning rate of 0.01 and a batch size

of 256. We train all the models for 175 epochs. The training accuracy of all the models in this experiment is 100%.

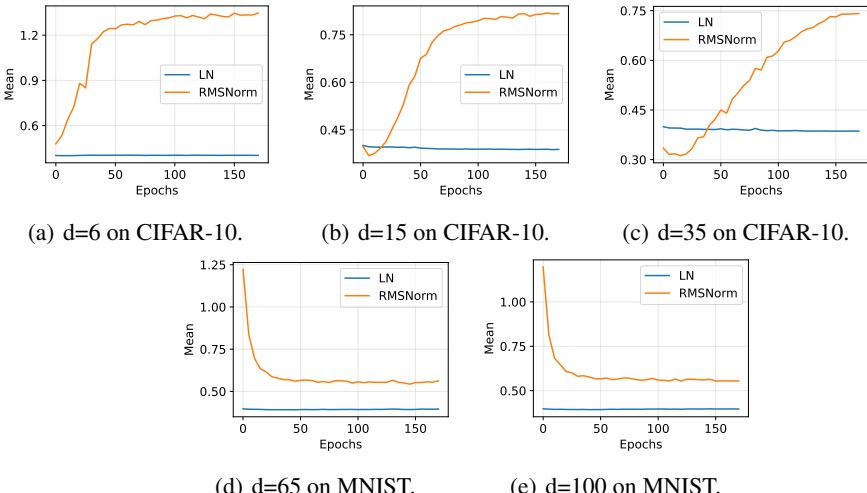

(a) d=6 on CIFAR-10.    (b) d=15 on CIFAR-10.    (c) d=35 on CIFAR-10.

(d) d=65 on MNIST.    (e) d=100 on MNIST.

Figure 12: *Mean of the final layer's input for MLPs of different depth (d). The change is similar to the change of norm.*

*We show the plot of the input norm of last layer during the training process in Figure 4. Here, we also provide the input mean value of the last layer in Figure 12. Since it is quite close to the change of the input norm, we will not go into detail.*

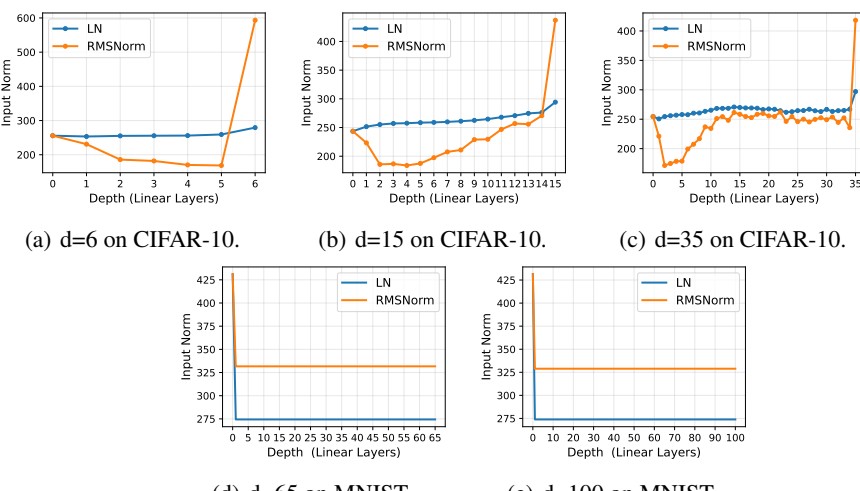

(a) d=6 on CIFAR-10.    (b) d=15 on CIFAR-10.    (c) d=35 on CIFAR-10.

(d) d=65 on MNIST.    (e) d=100 on MNIST.

Figure 13: *Norm of the input across linear layers for MLPs of different depth (d). LN better controls the norm of samples throughout the model.*

*Additionally, we recorded the input norms of every linear layer across all layers for the four models in the final training epoch in Figure 13. The results are summarized as follows:*

*- In plain MLP networks (without residual connections), the input norms of the LN model are not strictly smaller than those of RMSNorm at every layer. However, the RMSNorm-based models exhibit a sharp spike in input norm at the final layer, accompanied by clear oscillations in input norms between layers.*

*- In residual-connected MLP networks, the LN model consistently shows significantly smaller input norms than RMSNorm at every single layer. Moreover, these input norms remain remarkably stable and nearly constant across layers—an effect directly induced by the residual connections.*

*We also observed the variations in input norms across other layers. As depicted in Figure 14, the trend observed in activation function layers is similar to that in linear layers, which we will not elab-*

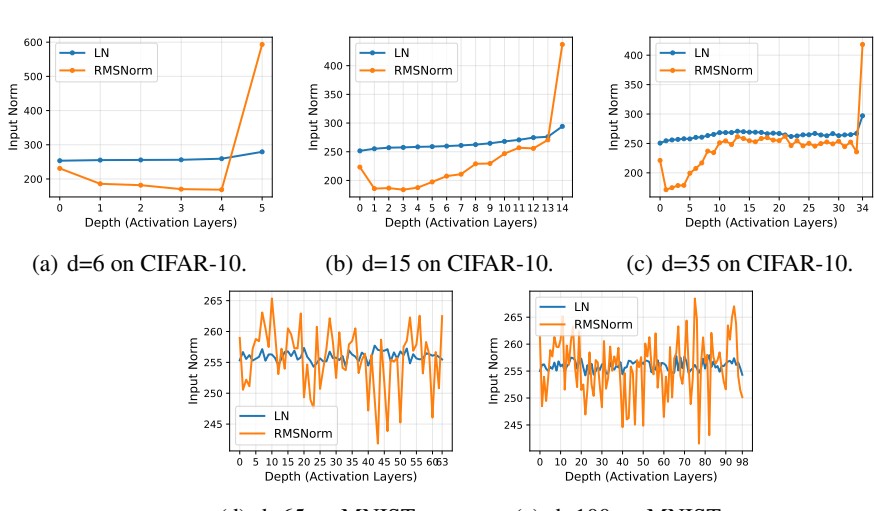

Figure 14: *Norm of the input across activation layers for MLPs of different depth (d). The change is similar to the change of norm.*

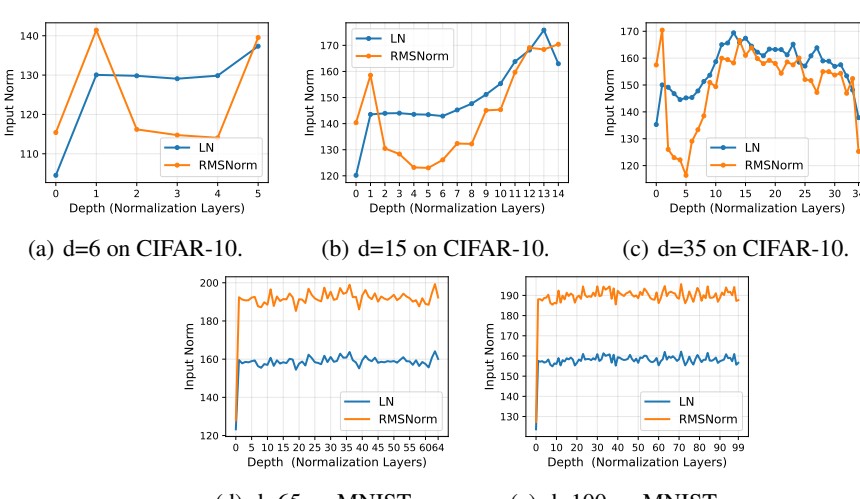

Figure 15: *Norm of the input across normalization layers for MLPs of different depth (d). The change is similar to the change of norm.*

*orate on further. To be noted, the range of input norm with residual connection is actually relatively small, between 240 and 270. As for the normalization layers shown in Figure 15, without residual connections, the range of input norms under RMSNorm exhibits more significant fluctuations, and the inter layer oscillations are greater; in the presence of residual connections, the input norms are effectively constrained by LN.*

*Therefore, we believe that the centering operation of LN controls the range of sample both across layers and throughout the training process.*

## G  ACCELERATION IN INFERENCE

In this section, we analyze how our methods accelerate model inference, both in theory and in experiments.

### G.1  *Theoretical Analysis*

In inference, CBWC modifies the weight once in the very beginning. Therefore, the acceleration of CBWC+RMSNorm can be considered as the gap between RMSNorm and LN. Since calculating RMSNorm has fewer computational steps compared to LN, replacing LN with RMSNorm has acceleration in FLOPs, thus reflecting on throughput.

FLOPs (Floating-Point Operations) measures the total number of floating-point operations required for one forward pass and is a standard metric for evaluating the computational cost of a model. Its value is decoupled from the number of parameters but directly determines inference latency and energy consumption. Therefore, it is often used for efficiency optimization, pruning, and quantization research.

We assume that addition, subtraction, and multiplication use 1 clock tick, while division uses 3 clock ticks.

Consider a sample $\mathbf{x}$, with dimension $d$. We have the equation of LN and RMSNorm as follows:

$$\text{LN}(\mathbf{x}) = \frac{\mathbf{x} - \mu}{\sqrt{\sigma^2 + \epsilon}}, \text{ where } \mu = \frac{1}{d}\sum_{i=1}^{d} x_i \text{ and } \frac{1}{d}\sum_{i=1}^{d}(x_i - \mu)^2. \tag{48}$$

$$\text{RMS}(\mathbf{x}) = \frac{\mathbf{x}}{\sqrt{\sigma_{rms}^2 + \epsilon}}, \text{ where } \sigma_{rms}^2 = \frac{1}{d}\sum_{i=1}^{d} x_i^2. \tag{49}$$

According to the formula above, we compute the number of operations in Table 7. For a $d$-dimension sample, LN needs $5d$ additions, $2d$ multiplications, and $1d$ divisions, while RMSNorm only needs $1d$ additions and $2d$ multiplications. We can see that RMSNorm requires less time and does not get stuck in some steps of the process.

Table 7: FLOPs calculation for LN and RMSNorm.

| Steps | LN | | | RMS | | |
|---|---|---|---|---|---|---|
| | $+/-$ | $\times$ | $\div$ | $+/-$ | $\times$ | $\div$ |
| Calculating average ($\mu$) | $d-1$ | 0 | 1 | — | — | — |
| Calculating variance ($\sigma^2$) | $2d-1$ | $d$ | 1 | $d-1$ | $d$ | 1 |
| Inverse square root for variance | One `rsqrt`, ignored | | | | | |
| (Centering &) scaling for each dimension | $d$ | 0 | $d$ | 0 | 0 | $d$ |
| Affine | $d$ | $d$ | 0 | 0 | $d$ | 0 |
| Total | $5d$ | $2d$ | $d$ | $d$ | $2d$ | $d$ |

In practice, PyTorch uses the Welford algorithm to do CUDA acceleration on LN, which has a different calculating process. Instead of traversing the data twice, this method requires only one

traversal, with the following equations:

$$\overline{x}_{n+1} = \overline{x}_n + \frac{x_{n+1} - \overline{x}_n}{n+1}$$

$$\widetilde{\sigma^2}_{n+1} = \widetilde{\sigma^2}_n + (x_{n+1} - \overline{x}_n)(x_{n+1} - \overline{x}_{n+1}), \tag{50}$$

where $\overline{x}_n$ and $\widetilde{\sigma^2}_n = n\sigma_n^2$ refer to the mean and variance of the first $n$ elements. For parallel computing, we combine two groups $M$ and $N$ with $m$ and $n$ elements in parallel computing with the following equation:

$$\overline{x}^{(M\cup N)} = \frac{m}{m+n}\overline{x}^{(M)} + \frac{n}{m+n}\overline{x}^{(N)}$$

$$\widetilde{\sigma^2}^{(M\cup N)} = \widetilde{\sigma^2}^{(M)} + \widetilde{\sigma^2}^{(N)} + \frac{mn}{m+n}\left(\overline{x}^{(M)} - \overline{x}^{(N)}\right)^2, \tag{51}$$

Similarly, the algorithm traverses data and merges groups for RMSNorm as outlined below:

$$\widetilde{\sigma^2}_{n+1} = \widetilde{\sigma^2}_n + x_{n+1}^2, \tag{52}$$

and

$$\widetilde{\sigma^2}^{(M\cup N)} = \widetilde{\sigma^2}^{(M)} + \widetilde{\sigma^2}^{(N)}, \tag{53}$$

which are much simpler than those for LN.

According to the formula above, we compute the number of operations in Table 8. For $d$-dimension samples separating into $g$ groups for parallel computation on CUDA, LN needs $7d$ additions, $(3d + 7g)$ multiplications, and $1d$ divisions, while RMSNorm only needs $1d$ additions and $3d$ multiplications.

Table 8: FLOPs calculation for LN and RMSNorm under Welford algorithm.

| Steps | Times | LN | | | RMS | | |
|---|---|---|---|---|---|---|---|
| | | $+/-$ | $\times$ | $\div$ | $+/-$ | $\times$ | $\div$ |
| Group initialization ($x_1^2$) | $g$ | 0 | 1 | 0 | 0 | 1 | 0 |
| Summing within groups | $d - g$ | 5 | 1 | 1 | 1 | 1 | 0 |
| Combining groups | $g - 1$ | 5 | 7 | 1 | 1 | 0 | 0 |
| Inverse square root for variance | 1 | One `rsqrt`, ignored | | | | | |
| (Centering &) scaling for each dimension | $d$ | 1 | 1 | 0 | 0 | 1 | 0 |
| Affine | $d$ | 1 | 1 | 0 | 0 | 1 | 0 |
| Total | — | $7d$ | $3d + 7g$ | $d$ | $d$ | $3d$ | 0 |

We note that the FLOPs of the LN are higher than the naive algorithm when the Welford algorithm is adopted. This is because Welford sacrifices the theoretical complexity when not grouped for parallel computation, which ultimately achieves a practical speedup. According to the equation, we simplify the time consumption from $13d + 7g$ to $4d$ clock ticks, resulting in an acceleration for about 60% to 80% time consumption.

*Theoretically,* we believe that there will be an improvement *in throughput*, which is consistent with the latency reduction. This is because throughput involves not only the speed of the model but also the memory occupation. Our method mainly involves omitting one centering operation and the relative bias in the affine transformation. Therefore, there is no significant reduction in the complexity of the model. Consequently, the main decisive factor for throughput is computational speed. Thus, we can refer to the calculations and conclusions we made above regarding inference latency.

### G.2 VERIFICATION EXPERIMENTS

We select a few common models and conduct experiments on acceleration in the inference stage. *It is worth mentioning that we compare our custom CUDA-accelerated RMSNorm kernel against PyTorch's official fused LN (the standard vendor-optimized kernel).*

*The reason for this is that PyTorch's implementation of LN already includes sophisticated acceleration algorithms, while RMSNorm does not. Existing RMSNorm kernel in PyTorch is significantly slower than its fused LN, making it unsuitable as a strong baseline. In the previous study (Jiang et al. (2023)), the authors compare self-made LN and self-made RMSNorm, implemented according to Eqn.48 and Eqn.49. The results differed from reality. As the self-made LN is slower, it will take up a larger proportion of the model's time.*

*Therefore, we implemented our own high-performance CUDA RMSNorm kernel from scratch (using the same Welford algorithm that PyTorch employs for its fused LN). This yields the comparison closest to the theoretical upper bound of achievable speedup.*

### G.2.1 *Total Time Usage and CUDA Time Usage*

We use `pytorch.profile` to trace the total time usage and CUDA time usage. We conduct 16 runs for each model, averaging 100 independent inference processes. We compare the time usage of each run with the average time usage of all runs for the vanilla model. We have noticed that the result is quite unstable. We suggest that according to our limit on devices, the start time span of the comparison groups is relatively large, leading to changes in computational resources, which will affect time usage.

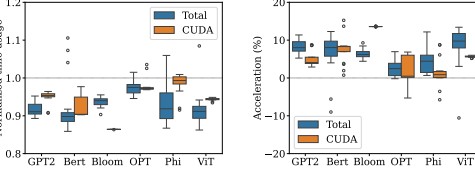

(a) *Normalized time.*    (b) Acceleration ratio.

Figure 16: *Comparison of inference latency across six representative models (GPT2, BERT, BLOOM, OPT, Phi3, and ViT). Our CBWC+RMSNorm achieves a consistent end-to-end runtime reduction of 2%–12% with zero accuracy degradation.*

We also measure the time usage proportion of LN in these models.

Table 9: Average inference time usage (ms) and proportion of LN in 5 models.

| Model | Total Time Usage | LN Time Usage | Proportion |
|---|---|---|---|
| Bert | 7.462299 | 0.713215 | 9.56% |
| Bloom | 2.321148 | 0.191480 | 8.25% |
| GPT2 | 7.125218 | 0.763676 | 10.72% |
| OPT | 10.14867 | 0.743909 | 7.33% |
| ViT | 6.552383 | 0.766613 | 11.70% |

### G.2.2 *Experiment Setting of Section 5.2*

*The inference performance of BERT, GPT-2, Phi and OPT was evaluated on an NVIDIA A100-40GB GPU. We executed 50 warm-up inferences, followed by a test phase. We run 15 runs for each model (50 inferences per run). All measurements were performed with a fixed batch size of 2 and input/output sequence length of 1024 tokens.*

*To analyze the results, we first calculated the mean inference time of the baseline method for each run. All data points within that run were then normalized against this mean to obtain the normalized time. The acceleration percentage was computed accordingly based on this normalized baseline.*

*We further investigated performance under an extended sequence length of 4096 tokens and 16384 tokens with a fixed batch size of 2. As demonstrated in Figure 17, the acceleration effect is significantly greater under these conditions, a finding that is consistent with our analytical expectations.*

### G.2.3 *Time Usage and Throughput*

*We evaluated the inference latency and throughput of BERT, GPT2 and OPT on a single NVIDIA V100-32GB and A100-40GB GPU.*

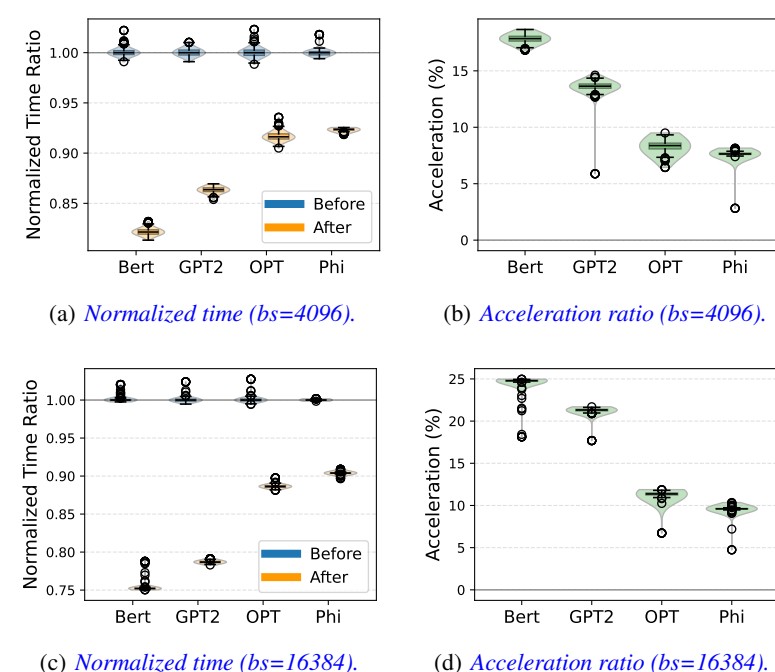

(a) *Normalized time (bs=4096).*

(b) *Acceleration ratio (bs=4096).*

(c) *Normalized time (bs=16384).*

(d) *Acceleration ratio (bs=16384).*

Figure 17: *Comparison of inference latency across six representative models (GPT2, BERT, BLOOM, OPT, Phi3). Our CBWC+RMSNorm achieves a consistent end-to-end runtime reduction of 2%–12% with zero accuracy degradation.*

Table 10: *Average inference latency (second) on a single GPU with batch size 2 and sequence length 256. Results are averaged over 5 runs × 50 inferences each after 50 warm-up steps.*

| Model | Hardware | Origin Inference Time(s) | New Inference Time (s) | Acceleration (%) |
|-------|----------|--------------------------|------------------------|------------------|
| Bert  | V100     | 0.0116                   | 0.0111                 | 3.90             |
| GPT2  | V100     | 0.0121                   | 0.0117                 | 3.57             |
| OPT   | V100     | 0.0119                   | 0.0116                 | 2.53             |
| Bert  | A100     | 0.0090                   | 0.0087                 | 2.79             |
| GPT2  | A100     | 0.0093                   | 0.0089                 | 4.22             |
| OPT   | A100     | 0.0094                   | 0.0089                 | 5.85             |

Table 11: *Average inference throughput (tokens/second) on a single GPU with batch size 2 and sequence length 256, averaged over 5 runs each consist of 50 inference steps after 50 warm-up steps.*

| Model | Hardware | Origin Throughput (tokens/s) | New Throughput (tokens/s) | Δ (tokens/s) |
|-------|----------|------------------------------|---------------------------|--------------|
| Bert  | V100     | 44329.0                      | 46126.1                   | +1797.1      |
| GPT2  | V100     | 42199.0                      | 43760.7                   | +1561.7      |
| OPT   | V100     | 43144.9                      | 44263.9                   | +1119.0      |
| Bert  | A100     | 57117.4                      | 58756.0                   | +1638.7      |
| GPT2  | A100     | 55351.4                      | 57787.8                   | +2436.5      |
| OPT   | A100     | 54468.1                      | 57853.1                   | +3385.0      |

*All measurements were performed with a fixed batch size of 2 and input/output sequence length of 256 tokens. To ensure stable timing, we first performed 50 warm-up inferences, followed by 50 measured inferences, and repeated the entire process 5 times for each configuration. The reported results are averaged over these 5 runs. Total inference time was measured using Python's* `time.time()` *.*

*As shown in Table 10 and Table 11, the proposed method consistently reduces inference latency by 2.5%–5.9% and increases throughput by 1,119–3,385 tokens/s across all three architectures and both GPU types. These gains are robust and reproducible, demonstrating that the acceleration is universal rather than specific to a particular model family or hardware platform.*

### G.2.4  *Scaling Ability of Folding Technology*

***Batch Size***   *To assess the scalability of the proposed folding technique with respect to batch size, we conducted additional experiments on BERT using a single NVIDIA V100-32GB GPU with a fixed sequence length of 256. Batch sizes were varied from 4 to 128 (powers of two). The same measurement protocol was followed: 50 warm-up inferences, followed by 50 timed inferences, repeated 5 times, with results averaged.*

Table 12: *Average inference latency (seconds) of BERT-base on a single V100 GPU (sequence length 256) across varying batch sizes. Results are averaged over 5 runs × 50 inferences after 50 warm-up steps.*

| Batch Size | Origin Inference Time(s) | New Inference Time (s) | Acceleration (%) |
| ---: | ---: | ---: | ---: |
| 4 | 0.0201 | 0.0193 | 3.99 |
| 8 | 0.0360 | 0.0347 | 3.64 |
| 16 | 0.0693 | 0.0675 | 2.58 |
| 32 | 0.1339 | 0.1269 | 5.25 |
| 64 | 0.2557 | 0.2418 | 5.45 |
| 128 | 0.5092 | 0.4763 | 6.45 |

Table 13: *Average inference throughput (tokens/second) of BERT-base on a single V100 GPU (sequence length 256) across varying batch sizes, averaged over 5 runs each consist of 50 inference steps after 50 warm-up steps.*

| Batch Size | Origin Throughput (tokens/s) | New Throughput (tokens/s) | Δ (tokens/s) |
| ---: | ---: | ---: | ---: |
| 4 | 51029.1 | 53147.9 | +2118.8 |
| 8 | 56920.5 | 59071.2 | +2150.7 |
| 16 | 59148.0 | 60715.7 | +1567.6 |
| 32 | 61185.0 | 64577.2 | +3392.1 |
| 64 | 64072.1 | 67761.8 | +3689.8 |
| 128 | 64357.4 | 68792.8 | +4435.4 |

*As shown in Table 12 and Table 13, the proposed method maintains stable and consistent performance gains across the entire batch-size range. Latency reduction ranges from 2.58% to 6.45%, while absolute throughput improvement increases progressively with batch size, reaching up to +4,435 tokens/s at batch size 128. Notably, the relative speedup remains largely unaffected by batch size, and throughput gains even grow in absolute terms at larger batches. These results confirm that the acceleration mechanism scales effectively with batch dimension and does not degrade under higher memory or compute intensity.*

***Sequence Length***   *To evaluate how performance gains evolve with sequence length, we conducted experiments on BERT, GPT2 and OPT using a single NVIDIA A100-40GB GPU with a fixed batch size of 4. We varied the input/output sequence length from 64 to 16,384 tokens. All measurements followed the identical protocol described earlier: 50 warm-up inferences, 50 timed inferences, repeated 5 times, with results averaged across runs.*

*As shown in Table 14, Table 15 and Table 16, the proposed folding technique exhibits strong positive scaling with sequence length. On BERT, absolute throughput improvement grows dramatically from +669 tokens/s at sequence length 64 to +10,536 tokens/s at 4,096 and remains substantial*

Table 14: *Average inference throughput (tokens/second) of BERT-base on a single A100 GPU (batch size 4) across increasing sequence lengths.*

| Seq Length | Origin Throughput (tokens/s) | New Throughput (tokens/s) | Δ (tokens/s) |
|---|---|---|---|
| 64 | 33464.1 | 34133.3 | +669.3 |
| 256 | 67590.8 | 72572.6 | +4981.9 |
| 1024 | 69072.5 | 77283.0 | +8210.5 |
| 4096 | 46579.5 | 57115.6 | +10536.1 |
| 16384 | 19349.4 | 25782.3 | +6432.9 |

Table 15: *Absolute throughput improvement (tokens/second) achieved by the proposed method across sequence lengths (batch size 4, single A100 GPU).*

| Method | 64 | 256 | 1024 | 4096 | 16384 |
|---|---|---|---|---|---|
| Bert | +669.3 | +4981.9 | +8210.5 | +10536.1 | +6432.9 |
| GPT2 | +1193.0 | +4890.3 | +6235.8 | +8806.8 | +8182.2 |
| OPT | +4697.6 | +3639.8 | +4145.0 | +5558.0 | +4630.1 |

*(+6,433 tokens/s) even at 16,384 tokens (Figure also extended to GPT2 and OPT in Table 15). Correspondingly, relative speedup increases monotonically with sequence length, rising from 1.96% at 64 tokens to an impressive 24.95% at 16,384 tokens (Figure also extended to GPT2 and OPT in Table 14).*

Table 16: *Relative inference speedup achieved by the proposed method across sequence lengths (batch size 4, single A100 GPU).*

| Model | 64 | 256 | 1024 | 4096 | 16384 |
|---|---|---|---|---|---|
| Bert | 1.96% | 6.86% | 10.62% | 18.45% | 24.95% |
| GPT2 | 4.23% | 6.65% | 8.50% | 13.88% | 21.47% |
| OPT | 14.13% | 5.09% | 5.17% | 8.04% | 11.51% |

*These results demonstrate that our method becomes increasingly effective as sequences grow longer, making it particularly valuable for modern long-context applications such as document-level reasoning, code generation, and retrieval-augmented systems. In contrast, as previously shown in Tables 12 and Table 13 , acceleration remains largely stable across batch sizes at fixed sequence length, confirming complementary scaling behavior along the batch and length dimensions.*

***Model Size*** *To examine the robustness of our folding technique across vastly different model scales, we further evaluated GPTJ-6B (Wang & Komatsuzaki, 2021) — a 6-billion-parameter decoder-only Transformer that is approximately 48× larger than GPT2 (124 M parameters) and 17× larger than BERT (340 M parameters) in non-embedding parameter count. All experiments were conducted on a single NVIDIA A100-40GB GPU following the identical measurement protocol used throughout this paper.*

*Table 17 and Table 18 report inference latency and throughput for representative (batch size, sequence length) configurations. Even on this significantly larger model, our method continues to deliver consistent and meaningful speedup. Relative latency reduction ranges from 4.2% to 7.7% across all in-memory configurations. The gains increase with sequence length (from ∼4.9% at 64 tokens to 7.5–7.7% at 1,024 tokens), following the same "longer-is-better" trend observed on smaller models. Absolute throughput improvement reaches +108–114 tokens/s at sequence length 1,024 — a substantial saving when amortized over billions of generated tokens in production.These results confirm that the acceleration mechanism remains highly effective when moving from hundred-million-scale models (GPT2, BERT, OPT) to multi-billion-parameter models, with no signs of diminishing returns.*

*Moreover, we compare the speed up of GPT2 and GPTJ. Even when scaling parameters by nearly 50× (GPT2 → GPTJ), relative speedup remains comparable (5–8%) demonstrating excellent parameter scalability.*

Table 17: Average inference latency of GPTJ on a single A100-40GB GPU. The proposed method achieves 4.2%–7.7% latency reduction, with larger gains at longer sequences. OOM denotes out-of-memory.

| Batch Size | Seq Length | Origin Inference Time(s) | New Inference Time (s) | Acceleration (%) |
|---|---|---|---|---|
| 4 | 64 | 0.2030 | 0.1931 | 4.89 |
| 4 | 256 | 0.7840 | 0.7453 | 4.94 |
| 4 | 1024 | 3.0976 | 2.8637 | 7.55 |
| 8 | 64 | 0.4187 | 0.4010 | 4.22 |
| 8 | 256 | 1.5709 | 1.4666 | 6.64 |
| 8 | 1024 | 6.2038 | 5.7415 | 7.45 |
| 16 | 64 | 0.7793 | 0.7409 | 4.92 |
| 16 | 256 | 2.9893 | 2.7601 | 7.67 |
| 16 | 1024 | OOM | OOM | \ |

Table 18: Inference throughput (tokens/second) of GPT-J-6B under the same settings as Table 17. The proposed method consistently improves throughput by 53–114 tokens/s, with the largest absolute gains again appearing at longer sequences.

| Batch Size | Seq Length | Origin Throughput (tokens/s) | New Throughput (tokens/s) | $\Delta$ |
|---|---|---|---|---|
| 4 | 64 | 1260.9 | 1325.7 | +64.8 |
| 4 | 256 | 1306.1 | 1374.0 | +67.9 |
| 4 | 1024 | 1322.3 | 1430.3 | +108.0 |
| 8 | 64 | 1223.0 | 1276.8 | +53.8 |
| 8 | 256 | 1303.8 | 1396.4 | +92.7 |
| 8 | 1024 | 1320.5 | 1426.8 | +106.3 |
| 16 | 64 | 1314.1 | 1382.0 | +68.0 |
| 16 | 256 | 1370.2 | 1484.0 | +113.8 |
| 16 | 1024 | OOM | OOM | \ |

Table 19: Relative speedup comparison between GPT-2-medium (124 M) and GPT-J-6B (6 B) on identical hardware and settings (single A100-40GB, batch size = 4).

| Model | Parameters | SeqLen=256 Speedup | SeqLen=256 Speedup |
|---|---|---|---|
| GPT2 | 124 M | 6.65% | 8.50% |
| GPTJ | 6 B | 4.94% | 7.55% |

# H EMPIRICAL EXPERIMENTS

*It is worth mentioning that we compare custom-PyTorch module implementations of RMSNorm and LayerNorm (implemented according to Eqn.48 and Eqn.49), both without CUDA kernels to isolate algorithmic speedup.*

## H.1 TEXT CLASSIFICATION TASK

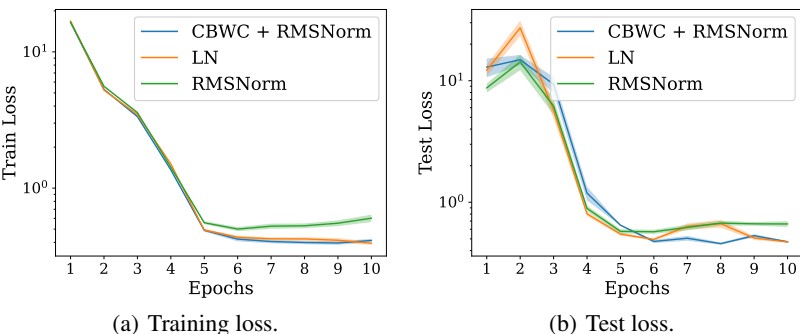

(a) Training loss.       (b) Test loss.

Figure 18: Results of transformer models for text classification task.

The experiment is conducted on a 3090Ti. The average results are shown in Table 20 and Figure 18. We mark the group with best result in bold and the worst in gray.

Table 20: Performance of Transformer on text classification.

| Method | Test Loss | Test Acc (%) |
|---|---|---|
| LN | **0.472 ± 0.010** | **85.21 ± 0.35** |
| RMS | 0.663 ± 0.037 | 76.57 ± 1.77 |
| CBWC+RMS | 0.473 ± 0.007 | 85.12 ± 0.23 |

## H.2 IMAGE CLASSIFICATION

For the Imagenet100, we select 100 classes from Imagenet1k (Deng et al., 2009) according to the given classes in (Tian et al., 2019). We chose SWIN-T for this experiment and trained on a single 3090. We apply the AdamW optimizer with a learning rate of $10^{-4}$ and a batch size of 128. Here we list the top 1 and top 5 accuracy and loss for both test and training in Table 21. We have the best results in bold and the worst results in gray.

Table 21: Average training results (mean ± std) under 3 random seeds for SWIN on ImageNet100.

| Model | TrainAcc@1 (%) | TrainAcc@5 (%) | TrainLoss |
|---|---|---|---|
| LN | **99.397 ± 0.003** | 99.919 ± 0.010 | **0.0254 ± 0.0002** |
| RMS | 99.376 ± 0.004 | 99.913 ± 0.001 | 0.0258 ± 0.0004 |
| CBWC+RMS | 99.382 ± 0.032 | **99.922 ± 0.005** | 0.0257 ± 0.0009 |

*We use* `time.time()` *to trace the time usage. This strategy is consistent with practices in prior efficiency-focused works Jiang et al. (2023), where authors also compared unoptimized module versions when no fast official RMSNorm existed.*

Moreover, to verify the training stability of our method, we applied different learning rates on the models. We trained 70 epochs for each group.

Under a higher learning rate ($10^{-3}$), the result of our method (CBWC+RMS) is between LN variant and RMSNorm one. As shown in Figure 19, with the decline of the learning rate, our method shows better stability as it can adapt to a higher learning rate. We also notice a more inclined accuracy curve, which is the evidence of faster convergence.

*Under a smaller learning rate ($10^{-5}$), the accuracy and loss of LN variant and RMSNorm are almost the same, while our method has different performance. Our method has slightly better generalization*

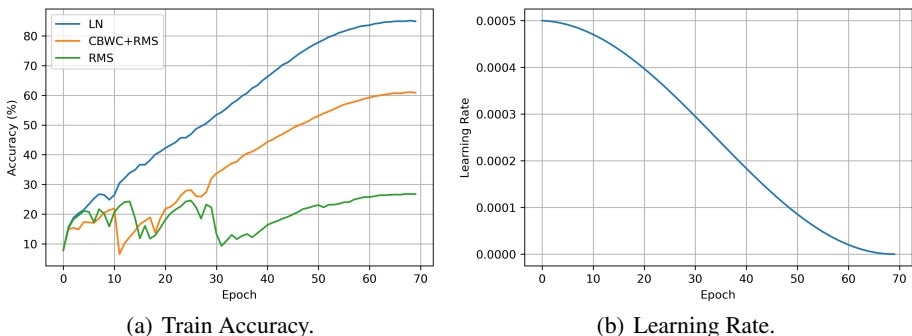

(a) Train Accuracy.

(b) Learning Rate.

Figure 19: Train accuracy of three models and learning rate setting of SWIN transformer on Imagenet100 under high learning rate setting.

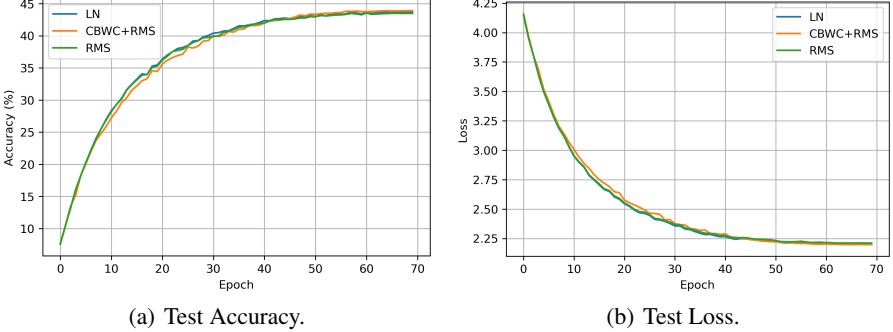

(a) Test Accuracy.

(b) Test Loss.

Figure 20: Performance of the three models in SWIN transformer on Imagenet100 under small learning rate.

*ability than model uses only RMSNorm and LN, with a slightly higher test accuracy (as shown in Figure 19) and lower test loss (as shown in Figure 20(b)).*

### H.3 TEXT GENERATION

We conduct text generation experiment on GPT-3.

Table 22: Perfomance of GPT-3 on Wikitext-103.

| Method | Train Loss | Train PPL | Test Loss | Test PPL |
|---|---|---|---|---|
| LN | 6.24 | 513 | 6.0598 | 428.30 |
| RMS | 6.32 | 557 | 6.1311 | 459.94 |
| CBWC+RMS | **6.18** | **484** | **6.0144** | **409.28** |

### H.4 VERIFICATION EXPERIMENT FOR FINE-TUNING ON PRE-TRAINED MODEL

#### H.4.1 EXPERIMENT SETTING FOR MLP

We verify that the CBWC+RMSNorm and original LN training scheme are identical in engineering by the following experiment.

We perform a classification task on CIFAR-10 Krizhevsky (2009). The MLPs have a depth of 6 and a width of 256. We train the models for 40 epochs with a learning rate of 0.01 and a batch size of 256, using a constant seed. The proxy parameter $W_A$ and the original weight matrix $W_B$ differ by less than $10^{-5}$. We can see in Figure 21 that there is little difference between the two set of parameters, which is attributable to numerical error. We therefore conclude that the two models follow indistinguishable optimization process and can be inter-onverted at any point during training without affecting the outcome.

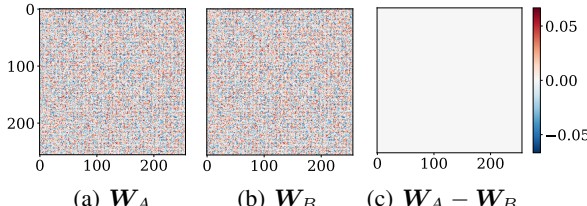

(a) $W_A$     (b) $W_B$     (c) $W_A - W_B$

Figure 21: Comparison of $W_A$ and $W_B$.

#### H.4.2 EXPERIMENT SETTING FOR TRANSFORMER

We pre-train the model with a learning rate of $5 \times 10^{-4}$ and an effective batch size of 128 (batch size is 2 and gradient accumulation is 64). For fine-tuning, we have a learning rate of $5 \times 10^{-5}$ and an effective batch size of 16 (batch size is 2 and gradient accumulation is 8).

### I *Open-source Code*

*To facilitate verification and further experimentation, we release the complete open-source implementation, including the fully automatic folding tool and optimized RMSNorm kernels:* `https://github.com/EnjoyYourLN/Enjoy-LN`.

