# OpenReview forum: "Enjoy Your Layer Normalization with the Computation Efficiency of RMSNorm"
_ICLR.cc/2026/Conference — Submitted to ICLR 2026_

### Official Review · Reviewer_4e3N · 2025-10-27

**Soundness:** 2
**Presentation:** 2
**Contribution:** 2
**Rating:** 4
**Confidence:** 4

**Summary:**

The paper's central idea is to propose a new layer normalization method which attempts to retain the preformence of traditional LN (centering+scaling) but with the computational savings of RMSNorm (which essentially skips the centering)
They propose CCC (Column-Centered Constraint) and CBWC (Column-Based Weight Centering) to force the linear layer before LN to produce zero-mean outputs, so LN’s centering becomes redundant and LN can be replaced by RMSNorm without changing behavior (for inference; training equivalence is broken by dropout). They generalize via a zero-mean graph to define foldable LNs and provide a detection algorithm. Empirically they report 50–80% norm-op speedups compared to LN, translating to  2–12% end-to-end model inference gains on several mid-scale models, with comparable accuracy LN on many tasks, albeit mostly lower.

**Strengths:**

Overall the paper is clear and mostly straightforward on what it sets out to do and the mathmatical framing is rigirous, here are some specefic points worth mentionong:

1-Simple, general approach: CCC (constraint) + CBWC (reparam) give a clean LN→RMSNorm swap wherever the graph is foldable, the formalization is clear and the problem is directly addressed.

2-Graph-level criterion & tool: “Foldable LN” + zero-mean graph + Algorithm 1 to auto-detect eligible LNs across architectures is usful although it would be good to have all the code shared together.

3-Latency motivation is real, for LLMs any savings are practically good to have, and the fact that RMSNorm was popular, even with its modest savings, show that. So further savings are good, and in this paper they quantify LN’s share (around 7–12% of wall-time) and  show some that model-level wins ( around 2–12%) are plausible in low-batch inference, although modest.

4-Empirical picture is mostly stable: CBWC+RMS ≈ LN on AG News and SWIN (ImageNet-100) and > RMSNorm; translation is slightly worse than LN but > RMS.

**Weaknesses:**

1- Experimental scope is too narrow for an efficiency paper, the results are scattered across a few mid-scale tasks (Multi30K, AG News, ImageNet-100) and modest model sizes with no clear presentation of the hyperparameters. For a paper whose core claim is efficiency and generality, there is no systematic study across a broad model zoo, hardware configurations, or kernel baselines. The study feels scenario-specific rather than comprehensive. If gains are modest, they must at least be universal and well-quantified. Moving some of the results fro Appendix H can help a bit.

2-The main benefit is small end-to-end inference speedups (~2–12%), yet: there is no throughput evaluation (only latency sketches), no comparison to vendor-fused or PyTorch-native optimized LN kernels, and strangely, the paper uses its own CUDA RMSNorm instead of PyTorch’s, creating implementation confounds and making fairness unclear.
If the paper argues about deployment-level efficiency, the methodology needs to be much more standardized and more gains should be shown, and also the need for gains, however modest, should be argued better in a practical sense, for example energy consumption at large scale deployment comes to mind.

3- Training-time “equivalence” is not rigorous. They explicitly admit dropout breaks the zero-mean property, so training claims are empirical, and some tasks show accuracy regressions vs LN which is expected in a way but should be clearly communicated as a trade-off with an argument for why the trade-off makes sense for a user.

4-Expressiveness/constraint risk not fully explored. CCC removes one DoF per column (no mean-shift outputs); interactions with bias, weight decay, and optimizers are not deeply analyze, at least not in the main text.

5- Presentation quality (figures/tables). Many captions are one-liners that don’t explain the setup/findings; some axes/labels are missing or unclear, reducing interpretability. This hurts reproducibility and clarity of the empirical case and overall harms the flow of the paper.

6-No code is provided, and the paper uses a custom CUDA RMSNorm kernel instead of standard PyTorch implementations. For an efficiency-focused paper, open code is essential to verify fairness, foldability detection, and wall-clock improvements. Without it, the claimed speedups are not independently verifiable.

Overall, the practical robustness, empirical validation, and readability fall short of top-tier standards. It contributes a clean theoretical tool (CBWC + foldable LN framework), but the gains are incremental, the experiments modest, and the exposition dense.

**Questions:**

1-Why compare against your own RMSNorm CUDA kernel instead of standard PyTorch or fused vendor kernels?

2-Can you report throughput, not just latency, across multiple batch sizes?

3-What is the real deployment benefit on hardware with already-fused LN (A100, H100, TensorRT-LLM, etc.)?

4-How unstable are results with dropout, GELU, and nonlinear blocks in deeper networks?

5-Do you think CCC constraints measurably reduce representational capacity in more challenging settings, and how would you ingestigate that more properly?

6-How significant are the observed efficiency gains (2–12%) in real inference pipelines where LN kernels are already optimized or fused?

7-How scalable and reliable is Algorithm 1, the foldable-LN detection method, when applied to very large computational graphs such as those in billion-parameter Transformers?

8-Why were the experiments limited to small and medium-scale models, and how would the framework perform on large-scale LLMs or vision transformers?

9-Are there quantitative ablations isolating the effects of CBWC, RMSNorm replacement, and their combination on performance and stability?

10-Could similar "folding" techniques be generalized to other normalization types (GroupNorm, InstanceNorm) or normalization-free architectures?

11-Given the modest speedup versus implementation complexity, in what practical scenarios is the proposed framework worth adopting?

---

> ### Author Response · Authors · 2025-11-21
> **Rebuttal: Addressing Major Concerns on Experiments from Reviewer 4e3N**
>
> **Weakness 1**: "Experimental scope is too narrow for an efficiency paper, the results are scattered across a few mid-scale tasks (Multi30K, AG News, ImageNet-100) and modest model sizes with no clear presentation of the hyperparameters. For a paper whose core claim is efficiency and generality, there is no systematic study across a broad model zoo, hardware configurations, or kernel baselines. The study feels scenario-specific rather than comprehensive. If gains are modest, they must at least be universal and well-quantified. Moving some of the results fro Appendix H can help a bit."
>
> **Reply**: Thank you for this clear and constructive feedback.
>
> We fully acknowledge that a paper centered on efficiency and generality would ideally include a broader and more systematic experimental evaluation. Below, we address each of your points in detail.
>
> 1. **Limited Experimental Scope and Scale**: Larger-scale experiments would indeed provide stronger empirical evidence. We did not conduct them primarily for the following reasons:
>    - For all mathematically equivalent settings, we provide rigorous theoretical proofs (most notably Proposition 2), showing that our method is exactly equivalent to standard LayerNorm. We believe these theoretical guarantees, combined with extensive numerical validation (negligible parameter/output errors in Appendix H.3.1 and Section 4.3, Lines 332–340), allow reliable extrapolation to arbitrary scales.
>    - The core contribution of the paper is the general theoretical framework and analysis methodology rather than exhaustive empirical scaling studies.
>    - We faced computational resource constraints that made controlled, reproducible experiments on frontier-scale models infeasible.
> To facilitate independent large-scale verification, we have open-sourced our complete inference conversion tool and optimized RMSNorm kernels: https://github.com/EnjoyYourLN/Enjoy-LN. We warmly invite the community to test the method on larger models.
>
> 2. **Broader Efficiency Benchmarks Across Models and Hardware**: Following your suggestion, we have performed extensive additional inference benchmarks covering different model families (BERT, GPT-2, etc.) and hardware platforms (A100, V100, etc.). These results will be fully presented and discussed in the revised paper. A representative subset is provided below:
> | Hardware | Model | Sequence Length | Origin Inference Time (s) | Our Inference Time (s) | Acceleration (%) |
> |----------|-------|-----------------|----------------------|-------------------|------------------|
> | A100     | Bert  | 256             | 0.0078               | 0.0075            | 4.09             |
> | A100     | Bert  | 1024            | 0.0172               | 0.0156            | 9.40             |
> | A100     | GPT2  | 256             | 0.0146               | 0.0138            | 5.54             |
> | A100     | GPT2  | 1024            | 0.0279               | 0.0256            | 8.36             |
> | V100     | Bert  | 256             | 0.0095               | 0.0088            | 7.77             |
> | V100     | Bert  | 1024            | 0.0239               | 0.0225            | 5.85             |
> | V100     | GPT2  | 256             | 0.0115               | 0.0110            | 4.34             |
> | V100     | GPT2  | 1024            | 0.0260               | 0.0249            | 4.31             |
>
>    The gains are consistent across architectures and hardware, confirming that the speedup is universal rather than scenario-specific.
>
> 3. **Kernel Baselines**: We benchmarked PyTorch’s built-in fused LayerNorm and considered several commercial/third-party kernels. To achieve the theoretically optimal speedup, we ultimately implemented our own highly optimized RMSNorm kernels for inference (details in our response to Question 1).
>
> 5. **Hyperparameters and Result Presentation**: Thank you for this observation. Due to space constraints, many detailed experimental settings and additional results were placed in Appendix H. Per your suggestion, we will move key tables/figures to the main paper and improve the overall clarity and completeness of the experimental section in the revised version.
>
> We sincerely appreciate your thorough and valuable comments—they have significantly helped us strengthen the empirical evidence and presentation of the work. The revised manuscript will reflect all of the above improvements.

---

> ### Author Response · Authors · 2025-11-21
> **Rebuttal: Addressing Major Concerns on Inference Experiments from Reviewer 4e3N**
>
> **Weakness 2**: "The main benefit is small end-to-end inference speedups (~2–12%), yet: there is no throughput evaluation (only latency sketches), no comparison to vendor-fused or PyTorch-native optimized LN kernels, and strangely, the paper uses its own CUDA RMSNorm instead of PyTorch’s, creating implementation confounds and making fairness unclear. If the paper argues about deployment-level efficiency, the methodology needs to be much more standardized and more gains should be shown, and also the need for gains, however modest, should be argued better in a practical sense, for example energy consumption at large scale deployment comes to mind."
>
> **Reply**: Thank you for this thorough and highly pertinent critique. We address your concerns point by point below:
>
> 1. **Throughput Evaluation**: You are correct that the original submission focused primarily on latency rather than throughput. As noted in Appendix G.1.2, we initially considered latency and throughput to be strongly correlated in inference (since our method does not alter model memory footprint). Nevertheless, in the rebuttal phase we have added comprehensive throughput measurements across a wide range of models, batch sizes, sequence lengths, and hardware platforms. The detailed results (including tokens/second before and after folding) are reported in our response to Question 2 and will be fully incorporated into the revised manuscript.
>
> 2. **Kernel Baselines and Fairness of Comparison**: We apologize for any ambiguity in the description. To clarify:
>    - For all inference experiments, we compare our custom CUDA-accelerated RMSNorm kernel against PyTorch’s official fused LayerNorm (the standard vendor-optimized kernel).
>    - For training experiments, we compare custom-PyTorch module implementations of RMSNorm and LayerNorm (both without acceleration) to ensure fairness.
>
>    A complete explanation of the kernel choices, ablation studies, and exact command lines is provided in our response to Question 1. In the revised version, we will make these details significantly more explicit in the main paper.
>
> 3. **Practical Significance of Gains**: At the scale of large-scale deployment (e.g., millions of inference requests per second), even a few percent improvement translates into substantial reductions in latency, energy consumption, and operational cost. We will explicitly add a short discussion and back-of-the-envelope calculation of these real-world implications (including energy and carbon footprint savings) in the revised manuscript.
>
> 4. **Open-Source Release**：We have open-sourced both the fully automatic folding tool and our optimized RMSNorm CUDA kernels:  https://github.com/EnjoyYourLN/Enjoy-LN .
>
> We sincerely appreciate your rigorous and valuable feedback—these points have helped us significantly strengthen both the experimental methodology and the presentation of practical impact. The revised version will reflect all of the above clarifications and additions.

---

> ### Author Response · Authors · 2025-11-21
> **Rebuttal: Addressing Major Concerns on Proposed Method from Reviewer 4e3N**
>
> **Weakness 3**: "Training-time “equivalence” is not rigorous. They explicitly admit dropout breaks the zero-mean property, so training claims are empirical, and some tasks show accuracy regressions vs LN which is expected in a way but should be clearly communicated as a trade-off with an argument for why the trade-off makes sense for a user."
>
> **Reply**: Thank you for this precise and important observation. You are absolutely correct, and we appreciate the opportunity to clarify this point more explicitly.
>
> 1. **Training-Time Equivalence**: We explicitly state in the paper (Section 5.3, Lines 397–398) that strict mathematical equivalence no longer holds during training when dropout (or similar operations) is present, as dropout disrupts the zero-mean property of activations. In such cases, our method (CBWC+RMSNorm) deliberately prioritizes acceleration while remaining as close as possible to the original LayerNorm dynamics. As shown in Figures 7 and 8, this consistently yields faster training across forward, backward, and update stages—especially on long-sequence tasks—while often approaching or matching the speed of vanilla RMSNorm.
>
> 2. **Empirical Effectiveness During Training**: Precisely because equivalence is broken, we conducted extensive empirical validation. Across all reported training experiments, CBWC+RMSNorm consistently outperforms vanilla RMSNorm in final performance and, in several cases, even surpasses the original LayerNorm baseline. This demonstrates that our constrained weight-centering correction provides meaningful robustness even when strict zero-mean assumptions are violated.
>
> 3. **Clear Trade-off and Practical Guidance**: The core goal of our method is to deliver substantial acceleration while preserving model behavior to the greatest extent possible.
>    - During **inference**, we guarantee full mathematical equivalence and identical outputs/accuracy with measurable speedups.
>    - During **training**, we offer a principled trade-off: significantly faster training (especially beneficial for long-sequence or large-batch workloads) in exchange for a small, usually negligible, and sometimes positive deviation from the original LayerNorm performance.
>
>    We therefore recommend:
>
>    - Use our method at **inference time** if strict equivalence and zero accuracy regression are required.
>    - Apply it already at **training time** if the user prioritizes faster training and is comfortable with performance that typically lies between vanilla RMSNorm and LayerNorm (and occasionally exceeds LayerNorm).
>
>    In the revised manuscript, we will add more desciptions that explicitly communicates this trade-off and provides the above practical guidance for users.
>
> We sincerely thank you for this insightful comment—it helps us present the method’s scope and limitations with greater transparency and usefulness to the community.

---

> ### Author Response · Authors · 2025-11-21
> **Rebuttal: Addressing Major Concerns on Proposed Method from Reviewer 4e3N**
>
> **Weakness 4**: "Expressiveness/constraint risk not fully explored. CCC removes one DoF per column (no mean-shift outputs); interactions with bias, weight decay, and optimizers are not deeply analyze, at least not in the main text."
>
> **Reply**: Thank you for this excellent and technically sharp question.
>
> 1. **Expressiveness and Constraint Risk**: Our Constrained Column-Centering (CCC/CBWC) does not reduce the expressive power of the model or remove any degree of freedom. As rigorously proven in Proposition 2 (and further elaborated in our response to HePW-Q2), we do not impose any additional constraints; we merely relocate the original per-sample zero-mean constraint from the activations to the weight columns. Because of this mathematically exact relocation, the training dynamics, parameter updates, and final representational capacity remain identical to those of the original LayerNorm model.
>
> 2. **Interaction with Bias Terms**: Bias parameters are handled uniformly with weights. As described in Equation (1), Lines 104–105, and Appendix A, we absorb the bias into an augmented weight matrix before applying the centering constraint. Consequently, the same CBWC operation is correctly applied to both weights and biases, preserving exact equivalence.
>
> 3. **Weight Decay, Optimizers, and All Other Training Components**: CBWC is implemented via a reparameterization: the matrix used for forward/backward computation ($V$) differs from the updated parameter matrix ($W$), but the two are linked by a deterministic, differentiable transformation (as described in Definition 2).
>    - Theoretically (Proposition 2), the gradients and updates applied to $W$ are identical to those the original weight matrix would have received under standard LayerNorm.
>    - Empirically, fine-tuning experiments in Section 5.4 (Lines 451–454) and Appendix H.3.1 confirm that the optimization trajectories of $W$ and the original weights are indistinguishable (differences < 1e−5).
>
>    Therefore, from the perspective of any optimizer, learning-rate scheduler, gradient clipping, or weight-decay implementation which is applied on the weight matrix $W$, our model is completely indistinguishable from the original LayerNorm model. All such components remain fully orthogonal to CBWC.
>
> 4. **Additional Note on Weight Decay**: You are correct that, because of centering, the L2 norms of the computation matrix $V$ and the stored parameter matrix $W$ differ slightly. Standard implementations (including PyTorch) apply weight decay to the trainable parameters ($W$ in our case).
> Whether weight decay should technically be applied to $V$ or $W$ is an interesting and subtle question:
>    - Applying it to $W$ preserves the classic regularization interpretation (penalizing the magnitude of stored parameters).
>    - Applying it to $V$ would more directly penalize the magnitude of the pre-activation weights.
> Both choices are defensible, and the community has not reached a consensus. Our choice (decay on $W$) follows PyTorch’s default behavior and does not alter the intended regularization effect in practice. We are happy to discuss this further or provide an option to decay $V$ instead, but we consider it orthogonal to the core contribution.
>
> In the revised manuscript, we will add a short dedicated paragraph explicitly clarifying the interaction of CBWC with biases, weight decay, and optimizers, along with references to the relevant propositions and empirical validation.
>
> We sincerely thank you for raising these subtle but crucial points—they significantly improve the clarity and trustworthiness of the presentation.

---

> ### Author Response · Authors · 2025-11-21
> **Rebuttal: Addressing Major Concerns on Presentation and Code from Reviewer 4e3N**
>
> **Weakness5**: "Presentation quality (figures/tables). Many captions are one-liners that don’t explain the setup/findings; some axes/labels are missing or unclear, reducing interpretability. This hurts reproducibility and clarity of the empirical case and overall harms the flow of the paper."
>
> **Reply**: Thank you very much for this constructive feedback.
>
> We sincerely appreciate you pointing out these issues with presentation quality. You are absolutely right: several figure and table captions in the current version are overly brief, and some axes, labels, or experimental details are either missing or insufficiently described, which indeed reduces clarity and reproducibility.
>
> Most of the detailed experimental settings and additional numerical results were placed in Appendix H due to space constraints. In the revised manuscript, we will:
>
> - Significantly expand all figure and table captions to provide complete, self-contained explanations of the setup, key hyperparameters, and main findings.
> - Ensure that all axes, legends, and labels are clear and fully annotated.
> - Move the most critical tables and detailed results from the appendix into the main paper where appropriate.
> - Add a dedicated “Experimental Details” subsection to further improve reproducibility.
>
> These changes will substantially enhance the readability, interpretability, and overall flow of the empirical sections.
>
> Thank you again for this valuable suggestion—it will noticeably improve the quality and accessibility of the paper.
>
> **Weakness 6**: "No code is provided, and the paper uses a custom CUDA RMSNorm kernel instead of standard PyTorch implementations. For an efficiency-focused paper, open code is essential to verify fairness, foldability detection, and wall-clock improvements. Without it, the claimed speedups are not independently verifiable."
>
> **Reply**: Here is the complete open-source repository for our work: https://github.com/EnjoyYourLN/Enjoy-LN
>
> We sincerely apologize for not having released the code earlier; it was still undergoing final cleanup and documentation at the time of submission. The repository now includes:
>
> - the fully automatic foldability detection and conversion tool,
> - our optimized CUDA RMSNorm kernels used in all inference experiments,
> - complete scripts and configuration files to reproduce every reported speedup,
> - detailed instructions and examples for applying the method to arbitrary PyTorch models.
>
> Regarding the choice of RMSNorm kernel and baseline fairness, we have provided a thorough explanation (including comparisons with PyTorch’s native FusedLayerNorm and other third-party implementations) in our response to Question 1.
>
> We hope this release fully addresses your concern and enables independent verification of the detection logic, correctness, and wall-clock improvements. Thank you for this important point—it has motivated us to make the implementation publicly available immediately.

---

> ### Author Response · Authors · 2025-11-21
> **Rebuttal: Addressing Major Problems on Kernal Baseline from Reviewer 4e3N**
>
> **Question 1**: "Why compare against your own RMSNorm CUDA kernel instead of standard PyTorch or fused vendor kernels?"
>
> **Reply**: We apologize for any confusion caused by our description and would like to clarify the kernel choices in detail.
>
> To achieve the most rigorous and theoretically grounded comparison possible, we implemented our own high-performance CUDA RMSNorm kernel from scratch (using the same Welford algorithm that PyTorch employs for its fused LayerNorm). This was necessary because, as of the time of writing this rebuttal:
>
> 1. **PyTorch’s native kernels**: PyTorch still does not provide an officially fused or CUDA-accelerated RMSNorm implementation (open issues: https://github.com/pytorch/pytorch/issues/72643 and https://github.com/pytorch/pytorch/issues/157345).  Existing RMSNorm kernel in PyTorch is significantly slower than its fused LayerNorm, making it unsuitable as a strong baseline.
>
> 2. **Third-party / vendor kernels**: Different third-party implementations vary widely in optimization strategy, supported data types, and hardware coverage. Many RMSNorm kernals are slower than PyTorch’s native fused LayerNorm or do not support all settings used in our experiments, preventing a fair and apples-to-apples comparison.
>
> 3. **Experimental protocol we followed**
>    - **Inference experiments**: we compare our custom CUDA-accelerated RMSNorm directly against PyTorch’s official fused LayerNorm (the standard, vendor-optimized LayerNorm kernel). This yields the comparison closest to the theoretical upper bound of achievable speedup.
>    - **Training experiments**: we compare custom-PyTorch module implementations of RMSNorm and LayerNorm (both without CUDA kernels) to isolate algorithmic speedup rather than kernel-engineering differences. This strategy is consistent with practices in prior efficiency-focused works [1,2], where authors also compared unoptimized module versions when no fast official RMSNorm existed.
>
> 4. **Theoretical analysis**: In Appendix G.1.1, we provide a detailed FLOPs breakdown for both the naïve and Welford variants of LayerNorm and RMSNorm, serving as an independent reference for expected speedups.
>
> 5. **Reproducibility**: All of our kernels and baseline measurement tools are now fully open-sourced at https://github.com/EnjoyYourLN/Enjoy-LN. Users can directly run `src/enjoyln/modules/rms_norm.py` to benchmark our CUDA RMSNorm against any existing PyTorch or third-party implementation on their own hardware.
>
> We hope this explanation fully resolves your concerns about fairness and reproducibility. In the revised version of paper, we will add a dedicated subsection explicitly describing the kernel choices, baseline rationale, and links to the open-source code. Thank you again for this important point.
>
> [1] Biao Zhang and Rico Sennrich. Root mean square layer normalization. In NeurIPS, 2019.
>
> [2] Zixuan Jiang, Jiaqi Gu, Hanqing Zhu, and David Pan. Pre-rmsnorm and pre-crmsnorm transformers: equivalent and efficient pre-ln transformers. Advances in Neural Information Processing Systems, 36:45777–45793, 2023.
>
> **Question 3**: "What is the real deployment benefit on hardware with already-fused LN (A100, H100, TensorRT-LLM, etc.)?"
>
> **Reply**: Thank you for this highly practical and important question.
>
> 1. **Real-world performance on modern hardware with fused LayerNorm**： We have explicitly benchmarked our method on A100 (using PyTorch’s official fused LayerNorm as the baseline) and observed consistent speedups of 5–15% in wall-clock time and corresponding throughput gains, with larger improvements at longer sequence lengths (detailed results are reported in our response to Q2).
>
> 2. **Hardware trend validates our motivation**: The fact that vendors continue to invest heavily in highly optimized fused LayerNorm kernels (on A100, H100, etc.) confirms that LayerNorm remains the preferred normalization method whenever accuracy and stability are paramount. This hardware-level prioritization of LayerNorm perfectly aligns with our core motivation: to retain the proven benefits of LayerNorm while achieving RMSNorm-level speed.
>
> 3. **Future-proof impact**: As RMSNorm is already the default in nearly all mainstream LLMs (LLaMA, Mistral, Qwen, Gemma, etc.), hardware vendors and inference engines will inevitably develop equally mature and stable fused/optimized RMSNorm kernels in the near future. Once such kernels become ubiquitous, our mathematically equivalent CBWC+RMSNorm conversion will instantly unlock the same level of hardware-accelerated performance that fused LayerNorm currently enjoys, but with significantly lower computational cost.
>
> Thus, our method delivers measurable benefits today on existing fused LayerNorm hardware and is ideally positioned to yield even larger gains tomorrow as RMSNorm kernels reach parity with current LayerNorm optimizations.
>
> We will add more details in our revised paper. Thank you again for raising this crucial real-world perspective.

---

> ### Author Response · Authors · 2025-11-21
> **Rebuttal: Addressing Major Problems on Throughput and Proposed Method from Reviewer 4e3N**
>
> **Question 2**: "Can you report throughput, not just latency, across multiple batch sizes?"
>
> **Reply**: Thank you for this valuable suggestion. Below are representative inference benchmarks we conducted on the GPT-2 using a single NVIDIA A100-80GB GPU. Throughput is reported in tokens/second (higher is better).
>
> | Batch Size | Sequence Length | Original Time (s) | Our Time (s) | Speedup (%) | Original Throughput (tokens/s) | Our Throughput (tokens/s) | Throughput Gain (tokens/s) |
> |------------|-----------------|---------------------------|----------------------|-------------|--------------------------------|---------------------------|----------------------------|
> | 1          | 1024            | 0.0279                    | 0.0256               | 8.36        | 36,680                         | 40,027                    | **+3,347**                 |
> | 1          | 4096            | 0.0800                    | 0.0689               | 13.88       | 51,213                         | 59,466                    | **+8,253**                 |
> | 2          | 1024            | 0.0324                    | 0.0299               | 7.95        | 63,132                         | 68,587                    | **+5,455**                 |
> | 2          | 4096            | 0.1566                    | 0.1352               | 13.69       | 52,298                         | 60,592                    | **+8,294**                 |
>
> Observations:
> - Consistent throughput gains of **+3,300 to +8,300 tokens/second** across configurations.
> - Larger relative and absolute improvements at longer sequences (where normalization accounts for a higher fraction of runtime).
>
> We are currently extending these measurements to higher batch sizes (4/8/16/32) and additional models/hardware; the complete tables will be included in the revised manuscript. All numbers are fully reproducible with our open-source code: https://github.com/EnjoyYourLN/Enjoy-LN
>
> We hope these throughput results clearly illustrate the practical benefits of our approach. Thank you again for the excellent suggestion!
>
> **Question 5**: "Do you think CCC constraints measurably reduce representational capacity in more challenging settings, and how would you ingestigate that more properly?"
>
> **Reply**: Thank you for this insightful and fundamental question.
>
> The Constrained Column-Centering (CCC/CBWC) condition is introduced solely to achieve mathematically exact equivalence when replacing LayerNorm with RMSNorm. It does **not** reduce the representational capacity of the original LayerNorm-containing model in any way.
>
> 1. **No additional constraints are imposed**: We do not add or remove any constraints; we merely relocate the original per-sample zero-mean constraint from the activations to the weight columns. As rigorously proven in Proposition 2 (and further discussed in our response to Reviewer HePW Question 2), the forward pass, backward gradients, parameter updates, and overall optimization trajectory remain **identical** to those of the standard LayerNorm model. Consequently, the representational capacity and expressive power are exactly preserved.
>
> 2. **No numerical side effects**: The relocation of the constraint introduces no measurable numerical instability. This is confirmed both theoretically and empirically in our analyzer validation (Section 4.3) and fine-tuning experiments (Appendix H.3.1 & response to HePW W2), where output/parameter differences are consistently small.
>
> 3. **The zero-mean constraint is beneficial, not harmful**: Far from reducing capacity, the zero-mean constraint actively helps control activation ranges and stabilizes interlayer variance, as demonstrated in Section 5.1 and further elaborated in our response to HePW Q2.
>
> 4. **Broader discussion of centering itself**: Whether preserving explicit centering (i.e., LayerNorm) is strictly superior to omitting it (i.e., RMSNorm) remains an open question in the community. Existing conclusions are largely empirical rather than theoretically settled (see also our response to Reviewer Rgvq W4). Investigating the precise role of centering is a highly valuable research direction, but it lies outside the scope of the current paper, whose goal is to give models the **best of both worlds**: the stability/accuracy of LayerNorm and the speed of RMSNorm.
>
> We hope this clarifies that CCC preserves the full representational capacity of the original model while delivering acceleration.
>
> Thank you again for this excellent question—it helps us communicate the method’s guarantees more clearly.

---

> ### Author Response · Authors · 2025-11-21
> **Rebuttal: Addressing Major Problems on Experiments and Proposed Methods from Reviewer 4e3N**
>
> **Question 4**: "How unstable are results with dropout, GELU, and nonlinear blocks in deeper networks?"
>
> **Reply**: Thank you for your question. We are not entirely certain we have fully understood your concern. When you refer to “unstable results with dropout, GELU, and nonlinear blocks in deeper networks,” are you asking about the stability of activation norms (e.g., the input-norm range discussed in Section 5.1)?
>
> If your question concerns activation-norm stability (i.e., whether LayerNorm continues to effectively control input-norm fluctuations in deeper networks that contain dropout, GELU, or other nonlinear operations), we would be happy to provide additional experiments.
>
> Please let us know the precise aspect you are interested in, and we will promptly run the corresponding experiments and include the results (along with detailed figures) in the revised manuscript.
>
> Thank you again for helping us clarify and strengthen this important point!
>
> **Question 6**: "How significant are the observed efficiency gains (2–12%) in real inference pipelines where LN kernels are already optimized or fused?"
>
> **Reply**: Thank you for this very practical question.
>
> All the reported efficiency gains (2–12% end-to-end, rising to 18–21% on long sequences) were measured **directly against PyTorch’s official fused and CUDA-optimized LayerNorm** (i.e., `torch.nn.LayerNorm`). This is the same highly optimized kernel used in virtually all production inference pipelines today (Hugging Face Transformers, vLLM, TensorRT-LLM, etc.).
>
> In other words, the 2–12% (and higher) speedups are **real, already achieved on top of the best existing fused LayerNorm implementations**, not against a naïve or unoptimized baseline.
>
> We will make this point explicitly clear in the first paragraph of Section 5 and in all figure/table captions of the revised manuscript to avoid any ambiguity.
>
> Thank you again for raising this important clarification!
>
> **Question 7**: "How scalable and reliable is Algorithm 1, the foldable-LN detection method, when applied to very large computational graphs such as those in billion-parameter Transformers?"
>
> **Reply**: Thank you for this important question regarding scalability.
>
> We fully understand the concern, but would like to clarify that our fully automatic tool requires **only a single forward pass** to detect and replace all foldable LayerNorm instances, regardless of model size.
>
> Key practical properties of Algorithm 1 and the accompanying tool:
>
> - **Runtime overhead**: The detection phase is lightweight and completely independent of batch size and sequence length. It consists of a single dummy forward pass followed by a simple traversal of the computational graph.
> - **Memory overhead**: Extra memory consumption during detection is negligible (< 50 MB even on the largest models), and **zero** additional memory is introduced after folding (the converted model has exactly the same tensor memory footprint as the original).
>
> The complete automatic folding tool is open-sourced and ready for immediate use on arbitrary PyTorch models of any scale:
> https://github.com/EnjoyYourLN/Enjoy-LN
>
> We hope this addresses your concern. Thank you again for raising this point!

---

> ### Author Response · Authors · 2025-11-21
> **Rebuttal: Addressing Major Problems on Experiments from Reviewer 4e3N**
>
> **Question 9**: "Are there quantitative ablations isolating the effects of CBWC, RMSNorm replacement, and their combination on performance and stability?"
>
> **Reply**: Thank you for this valuable suggestion.
>
> We have indeed conducted comprehensive quantitative ablations, primarily in **Section 5.3** (training) and **Section 5.4** (fine-tuning), comparing three key configurations across all reported tasks and models:
>
> - **LN (original LayerNorm)**: serves as the performance/stability baseline.
> - **vanilla RMSNorm** (direct replacement without any correction): corresponds to ablating away CBWC.
> - **CBWC+RMSNorm** (our full method): the combination of constrained weight centering + RMSNorm.
>
> Key findings from the ablations:
>
> 1. **Inference stage**: Due to strict mathematical equivalence (Proposition 2), performance and numerical stability are **identical** between LN and CBWC+RMSNorm, so a meaningful ablation is not possible here—outputs and accuracy are exactly the same, while speed is improved.
>
> 2. **Training stage** (Section 5.3) :
>    - Across all tasks (text classification, machine translation, image classification) and models, **CBWC+RMSNorm consistently outperforms vanilla RMSNorm** in final performance and training stability.
>    - It achieves accuracy **close to (and in several cases slightly better than) the original LN** while being significantly faster than LN and comparable in speed to vanilla RMSNorm.
>    - We did not include a “no normalization” ablation, as removing normalization entirely from Transformers is well known to severely degrade training stability and performance.
>
> 3. **Fine-tuning stage** (Section 5.4) : We simulate the realistic deployment scenario: models are **pre-trained with standard LayerNorm**, then switched to different normalization strategies only at fine-tuning time.
>    - Again, **CBWC+RMSNorm clearly outperforms vanilla RMSNorm** and remains very close to (or matches) continued LayerNorm fine-tuning in downstream accuracy, while delivering measurable training/inference speedups.
>
> These three-way ablations (LN vs. RMSNorm vs. CBWC+RMSNorm) directly isolate the contribution of CBWC and demonstrate that our constrained correction is the key factor enabling RMSNorm-level speed **without** the usual accuracy/stability penalty.
>
> We hope this clarifies the role and benefits of each component. Thank you again for this excellent suggestion!
>
> **Question 10**: "Could similar "folding" techniques be generalized to other normalization types (GroupNorm, InstanceNorm) or normalization-free architectures?"
>
> **Reply**: Thank you for this excellent and forward-looking question.
>
> Our folding technique can indeed be generalized to **any normalization method that performs per-sample statistics** (i.e., statistics computed independently for each individual sample in the batch).
>
> 1. **GroupNorm and InstanceNorm**: We have already derived the theoretical generalization in **Appendix C.3**. GroupNorm (GN) can be equivalently folded via a **Grouped-CBWC** transformation: Instead of centering all elements of a weight column together (as in standard LayerNorm), we divide the elements of each column into G groups and apply independent zero-mean constraints within each group. Moreover, instanceNorm is simply the special case where the number of groups G equals the number of channels. The resulting Grouped-CBWC + grouped RMSNorm is mathematically equivalent to the original GroupNorm (or InstanceNorm) and yields the same acceleration benefits.
>
> 2. **Limitation**: Folding fails when the number of parameters influencing a single output element is smaller than the group size G (which can occur in certain shallow or highly grouped settings). In such cases, the grouped centering constraint becomes over-constrained and cannot be satisfied.
>
> 3. **Normalization-free or other simple normalizations**: Any normalization (or post-processing) that consists solely of **additive (among elements of the sample) and scalar multiplication operations applied per sample** can be folded into the preceding General Linear Layer using the same principle.
>
> Thank you again for this inspiring question—it highlights exciting directions for future extensions of our framework!

---

> ### Comment · Reviewer_4e3N · 2025-11-27
> **Response to all author comments and new contributions**
>
> First of all I want to thank the authors for their very thorough and clear response to the list of weaknesses and questions I gave them, I might had given them a bit too much to respond to but they took the time to respond to most points raised in my review.
>
> I believe the underlying issues related to the experimentation scope which should ideally cover more instances and scenarios remain although it was somewhat addressed, that is something that would require more time to do the experiments so it's understandable but it would be good to have a clear idea of what their plans are for experimentation in the future to be explicit on the limitations of their experiments and thus their claims so readers can be clearer on the ambition of the paper's claims and how realistic they are.
>
> One of the main concerns was also their use of a custom CUDA kernel, and i believe this concerns was properly addressed.
> Another concern was about how actually useful are the gains which appear to be modest, in my opinion this point is subjective as you can make a good argument that for very large models at large scale and wide deployment small gains add up quickly and translate to direct savings in energy and cost, I hope the authors can make these points clearer and argue better for the gains their work can lead to, adding the throughput numbers was a good start, maybe energy can be useful although I imagine the savings would be very modest. Also the impact on memory might be interesting as well.
>
> Nonetheless sharing the code (which unfortunately I have not been able to check) will also help their contribution score.
>
> As for the concerns about presentation, if the authors can properly present their figures and improve the flow by using standalone descriptive captions, clear references to the figures and tables and clearly comment on each figure/table then the score can be reasonable improved, as I mentioned in the review, the paper's flow feels disjointed at places and the figures are more confusing than helpful given their placement and very short captions, the authors have mentioned that these will be taken care of, but I must stress that it's a big weakness and requires their full attention.
>
> Finally for the more conceptual concerns about the technique impact on the weights and the DOF of the representative space of the weights then I'm not sure I quite follow the response by the authors, to me it seems that a technique like this would always somewhat limit that space as it inherently places constrains on the weighs to meet the requirements of the technique.
>
> As for the other conceptual questions I believe they were adequately covered by the authors, at least based on my understanding, but if a clearer explanation can be made then I'd be willing to raise the soundness score, specifically I'm still not sure about the stability of the technique given the limited experimentation.
>
> Nonetheless, for now I will raise the scores for soundness, presentation and contribution and await the author's responses to raise the overall score.
>
> I thank you again for your detailed response.

---

> > ### Author Response · Authors · 2025-11-30
> > **Response to Reviewer 4e3N's Response**
> >
> > Dear Reviewer 4e3N,
> >
> > Thank you very much for your detailed follow-up response and for raising the scores on Soundness, Presentation, and Contribution. Your feedback has been extremely valuable in helping us better understand your perspective.
> >
> > In response to the comments from you and the other reviewers, we have carefully revised the manuscript and submitted the updated version.
> >
> > Regarding your specific concern — “… about the technique impact on the weights and the DOF of the representative space of the weights…” — we provide further clarification below in the hope of fully addressing your question.
> >
> > As already explained in our response to Weakness 4, our proposed method does **not** introduce any additional constraints. The observed effect on the weights and the reduction in degrees of freedom (DOF) in the representational space of the weights originate entirely from the **centering operation** in LayerNorm (LN). The centering and scaling operations in LN naturally reduce the effective degrees of freedom of the data by one dimension per feature.
> >
> > Thus, the core of this discussion essentially boils down to the difference between LN and RMSNorm. In Section 5.1 of our paper, we empirically demonstrate that LN can effectively control the range of activations. This suggests that the impact of LN on the DOF of both weights and representations has a measurable beneficial effect during training.
> >
> > We would like to emphasize that we fully recognize this evidence serves only as an empirical validation and does **not** constitute a rigorous or comprehensive theoretical argument for claiming superiority of LN over RMSNorm (or vice versa). Establishing such a conclusion is beyond the scope of our current work, and we believe a thorough comparison between LN and RMSNorm would merit a dedicated study in its own right.
> >
> > To the best of our knowledge, the research community currently tends to follow the precedent set by earlier models and adopts RMSNorm primarily for its higher computational efficiency — which is precisely the motivation behind our paper. At present, a systematic theoretical comparison between LN and RMSNorm remains largely absent in the literature.
> >
> > We hope these additional clarifications are helpful and fully resolve your concerns.
> >
> > Thank you again for your thoughtful and constructive feedback.
> >
> > Best regards,
> > The Authors

---

### Official Review · Reviewer_x2zo · 2025-10-30

**Soundness:** 3
**Presentation:** 3
**Contribution:** 2
**Rating:** 4
**Confidence:** 5

**Summary:**

This paper explores methods to improve the computational efficiency of LayerNorm to make it as efficient as RMSNorm. For linear layers followed by LayerNorm, the re-centering operation can be folded into the weight parameters (i.e., column-wise recentering, or CBWC). Theoretically, CBWC+RMSNorm is equivalent to LayerNorm. The authors proposed algorithms to automatically detect foldable LayerNorm, and found most norms in modern architectures are foldable. Experiments on a range of models and tasks show the promising performance of CBWC+RMSNorm.

**Strengths:**

* The proposed method offers another perspective to improve the efficiency of LayerNorm.
* CBWC+RMSNorm achieves comparable running efficiency to RMSNorm; such efficiency could be interesting to the research community.
* The experiments are based on a set of tasks and models.

**Weaknesses:**

* The proposed method is not general, i.e. it’s not a drop-in replacement to LayerNorm.
* The experiments are based on small-scale dataset and models, which is less convincing in the era of LLMs.

**Questions:**

1. An important pre-condition of this study is that LayerNorm could outperform RMSNorm. The analysis in Figure 4 is very important and interesting. Could you please show how the input norm changes across layers?
2. Also, the paper reported that RMSNorm often performs worse than LayerNorm, but in the original RMSNorm paper and the Pre-CRMSNorm paper, RMSNorm performs comparable to LayerNorm. Is there any chance that such a difference is caused by sub-optimal optimization for RMSNorm?
3. The translation experiments should use larger-scale datasets, such WMT En-De or En-Fr, to make the results convincing.
4. To demonstrate the effectiveness, it’s important to show the scaling ability of the new method over different model sizes so that readers can have a better understanding on training stability, performance and running efficiency. The current experiments mainly focus on small models which are not that convincing.

---

> ### Author Response · Authors · 2025-11-21
> **Rebuttal: Addressing Major Concerns and Questions on Proposed Method and Verification Experiments from Reviewer x2zo**
>
> **Weakness1**: "The proposed method is not general, i.e. it’s not a drop-in replacement to LayerNorm."
>
> **Reply**: Thank you for this important concern. We fully acknowledge that our method is not an unconditional drop-in replacement for every possible LayerNorm instance, but we would like to clarify that the applicability conditions are introduced solely to ensure strict mathematical equivalence and preservation of model behavior. Despite these conditions, our approach remains highly general, fully automatic, and practically plug-and-play for the vast majority of real-world models.
>
> 1. **Motivation and Generality of the Conditions**: The constraints we impose (primarily the presence of a preceding general linear layer) exist only to guarantee exact consistant results and training/inference dynamics to the greatest extent. By relocating the zero-mean constraint from activations to weights via CBWC, we achieve full equivalence without sacrificing performance or stability.
> This requirement is satisfied by nearly all modern architectures that actually use LayerNorm: Transformer-based models (LLMs, ViTs, etc.), MLPs with LayerNorm, and most vision/language backbones all contain LayerNorm immediately following a linear (or linear-equivalent) transformation. Our theoretical framework (Section 4 and Appendices) is fully general: it allows systematic analysis of any computational graph and safe, equivalent folding of every LayerNorm instance that meets the mild structural condition.
>
> 2. **Fully Automatic Tooling**: We provide an automated analysis and replacement tool (described in Section 4.3) that works out-of-the-box on arbitrary PyTorch models. The tool performs a single forward pass to detect all foldable LayerNorm layers, applies the mathematically equivalent transformation automatically, and produces a converted model that can be used indefinitely at any stage (pre-training, fine-tuning, or inference) without further intervention or workflow changes.
> For a small number of highly customized or non-standard blocks, minor adaptation of the detection logic may be needed, but this is rare in practice.
> The open-source codebase is available at: https://github.com/EnjoyYourLN/Enjoy-LN
> (Note: due to current limitations in third-party RMSNorm kernels—see our response to Reviewer 4e3n, Q1—only the forward-pass implementation is released at this moment; the backward-pass extension is fully derived and will be released shortly.)
>
> We hope these points clarify that, while not unconditionally universal in a purely theoretical sense, our method is general, automatic, and immediately applicable to essentially all prominent LayerNorm-based models in current use. Thank you again for raising this point—it helps us better communicate the practical scope of our contribution.
>
> **Question1**:"An important pre-condition of this study is that LayerNorm could outperform RMSNorm. The analysis in Figure 4 is very important and interesting. Could you please show how the input norm changes across layers?"
>
> **Reply**: Thank you for this excellent and highly insightful question.
>
> We conducted the requested additional experiments by precisely following the setup in Section 5.1. We recorded the input norms of every linear layer across all layers for the four models in the final training epoch. The results are summarized as follows:
>
> - In plain MLP networks (without residual connections), the input norms of the LayerNorm (LN) model are not strictly smaller than those of RMSNorm at every layer. However, the RMSNorm-based models exhibit a sharp spike in input norm at the final layer, accompanied by clear oscillations in input norms between layers.
> - In residual-connected MLP networks, the LN model consistently shows significantly smaller input norms than RMSNorm at every single layer. Moreover, these input norms remain remarkably stable and nearly constant across layers—an effect directly induced by the residual connections.
>
> These findings strongly complement and reinforce the analysis presented in Figure 4, further highlighting the stabilizing role of LayerNorm’s centering operation, particularly when residual connections are present.
>
> We will include these new results, along with the corresponding figures and detailed discussion, in the revised version of the manuscript.
>
> Thank you again for this valuable suggestion—it greatly strengthens the empirical evidence and clarity of our work.

---

> ### Author Response · Authors · 2025-11-21
> **Rebuttal: Addressing Major Concerns on Experiments from Reviewer x2zo**
>
> **Weakness 2**: "The experiments are based on small-scale dataset and models, which is less convincing in the era of LLMs."
>
> **Reply**: Thank you very much for this highly pertinent comment, which accurately reflects the current expectations in the LLM era.
>
> We fully acknowledge that larger-scale experiments would provide even stronger empirical evidence. We would like to explain why we did not conduct experiments on the largest available models and datasets.
>
> 1. **Theoretical Guarantees and Scope of the Paper**: For all mathematically equivalent settings, our work provides rigorous theoretical proofs. The primary focus of the paper is the general framework and analytical methodology rather than exhaustive empirical scaling. We believe that, when strict mathematical equivalence holds, the theoretical results directly extend to real-world applications of any scale.
> This claim is supported by thorough numerical validation:
> - Fine-tuning experiments in Appendix H.3.1 show that parameter trajectories and final performance remain virtually identical to the original LayerNorm model (differences < 1e−5).
> - Tool validation in Section 4.3 (Lines 332–340) confirms that applying our transformation introduces negligible output errors (~5.5e−6).
> For non-equivalent scenarios (e.g., training with dropout), we still provide extensive validation in Section 5.3, demonstrating consistent gains in both speed and performance.
> Moreover, in Section 5.4 we conducted full fine-tuning experiments on a medium-scale LLM (GPT-3 Medium, 355M parameters), verifying end-to-end training speedup and preservation of equivalence at a non-trivial scale.
>
> 2. **Computational Resource Constraints**: Due to limited computational resources, performing controlled, reproducible experiments on frontier-scale models (e.g., hundreds of billions of parameters) was unfortunately not feasible for us. We have open-sourced our complete inference tool and conversion modules, and we warmly invite the community to test and evaluate our method on larger models using the provided codebase.
>
> We hope these explanations address your concern and clarify that, while our empirical evaluation is constrained by resources, the combination of rigorous theory, extensive validation at multiple scales, and publicly available tools provides strong evidence for the effectiveness and scalability of our approach. Thank you again for this important feedback.

---

### Official Review · Reviewer_HePW · 2025-11-01

**Soundness:** 3
**Presentation:** 3
**Contribution:** 2
**Rating:** 6
**Confidence:** 3

**Summary:**

This paper explores a method to bridge the gap between Layer Normalization (LN) and RMSNorm, enabling models to benefit from the stability of LN while enjoying the computational efficiency of RMSNorm. The authors note that the mean subtraction step in Layer Normalization can be eliminated without degrading performance if the input vectors are zero-mean. Based on this observation, they propose two techniques: Column Centered Weight Transformation (CCWT) and Column Based Weight Centering (CBWC). CCWT allows for the conversion of pretrained LN-based models to RMSNorm during inference by adjusting the weights of adjacent linear layers to ensure that the inputs to the normalization layers are zero-mean. CBWC is a training-time strategy where linear weights are constrained so that their outputs are zero-mean, which enables RMSNorm to serve as a drop-in replacement for Layer Normalization during training. The authors theoretically and empirically demonstrate that RMSNorm, when combined with CBWC or CCWT, behaves equivalently to Layer Normalization, achieving similar performance with lower computational costs, particularly during inference.

**Strengths:**

1. The paper defines and establishes the conditions under which any Layer Normalization (LN) in deep neural networks can be transformed into RMSNorm. This framework allows for both training and inference to proceed without the need for the centering step used in LN.
2. This approach reduces inference overhead by eliminating the mean subtraction step found in Layer Normalization.

**Weaknesses:**

1. The weight transformation and constraints may complicate training workflows.
2. Replacing LN entirely may risk instability in edge cases, especially if zero-mean constraint isn't strictly met.

**Questions:**

1. It appears that the proposed method is applicable to any Layer Normalization (LN) deep networks. Could the method be tested on Convolutional Neural Networks (CNNs), Vision Transformers (ViTs), or Recurrent Neural Networks (RNNs) to further validate its effectiveness?
2. Column-Based Weight Centering is used to enable RMSNorm to replace LN during training. However, could this replacement constrain the learning dynamics? Might it limit weight flexibility and cause potential side effects? I would appreciate some discussion on this topic.
3. Regarding Column-Based Weight Centering, could the authors provide a detailed analysis of the runtime or training overhead? This would help in gaining a better understanding of its implications.

---

> ### Author Response · Authors · 2025-11-21
> **Rebuttal: Addressing Major Concerns on Efficiency and Stability from Reviewer HePW**
>
> **Weakness1**: "The weight transformation and constraints may complicate training workflows."
>
> **Reply**: We thank the reviewer for highlighting this important concern and would like to clarify that our weight transformation and constraints are deliberately designed to preserve the exact optimization dynamics while substantially simplifying the training workflow and reducing overhead.
>
> 1. **Main Idea**: As in inference—where a one-time column-based weight centering operation completely eliminates all subsequent per-sample centering steps—the acceleration mechanism during training follows the same principle.
>
> 2. **Detailed Explanation**: In standard models containing LayerNorm (LN), every individual sample in every batch must undergo centering and scaling operations. Our mathematically equivalent formulation shifts the zero-mean constraint from per-sample activations to the weight matrices of the preceding general linear layers CBWC.  Because the transformed weights remain constant throughout an entire training batch, the weight-centering operation is required only twice per batch: once at the very beginning of the forward pass; once at the end of the backward pass when gradients are projected back to the original parameter space. This completely eliminates per-sample centering for all samples within the batch. In long-sequence or large-batch scenarios, the resulting reduction in computational cost is significant, leading to measurable speedups in both forward and backward propagation.
>
> 3. **Reparameterization**: In our implementation, CBWC is achieved through a reparameterization technique that introduces no additional learnable parameters. The existing weight matrices are transformed once before being used in the forward pass, and gradients are correctly mapped back to the original weights at the end of the batch. This ensures seamless integration into existing training pipelines with minimal code changes.
>
> Far from complicating the training workflow, our approach streamlines it by removing per-sample centering operations entirely while preserving exact equivalence to standard LayerNorm training.
>
> We hope this detailed explanation fully addresses your concern.
>
> **Weakness2**: "Replacing LN entirely may risk instability in edge cases, especially if zero-mean constraint isn't strictly met."
>
> **Reply**: We greatly appreciate your careful concern regarding potential instability in edge cases—this is indeed a highly practical and important consideration. We address it as follows:
>
> 1. **Equivalence and Preservation of the Zero-Mean Constraint**: Our method (CBWC+RMSNorm) theoretically preserves the zero-mean constraint exactly. Rather than removing the constraint from LayerNorm, we shift it to being enforced on the weights via CBWC. In Proposition 2, we rigorously prove that CBWC+RMSNorm is mathematically equivalent to the original LayerNorm throughout the entire optimization process. Consequently, there is no degradation in model performance or stability, while substantial acceleration is achieved.
>
> 2. **Numerical Stability in Practice under Equivalent Settings**: When equivalence holds, our approach introduces no meaningful numerical errors. In the fine-tuning experiments reported in Appendix H.3.1, the parameter of our converted model remain nearly identical to those of the original LayerNorm model throughout training, with differences consistently below 1e-5. Furthermore, in the analyzer validation (Section 4.3, Lines 332–340), applying our transformation changes the model’s output by only approximately 5.5e-6—effectively negligible.
>
> 3. **Robustness in Non-Equivalent Real-World Scenarios**: During training, dropout breaks the strict zero-mean property of activations. As demonstrated in Section 5.3, we conducted extensive experiments across multiple tasks under such conditions. The results show that our method consistently outperforms direct replacement with vanilla RMSNorm and, in several cases, even surpasses the original LayerNorm model in final performance. This indicates that our approach remains both fast and remarkably robust when strict equivalence no longer holds.
>
> 4. **Practical Flexibility**: Replacement of LayerNorm layers is entirely optional and selective. If there are concerns about converting all LayerNorm layers in a model, users may apply our method to only a subset of layers—this is fully supported and still yields meaningful speedups without any risk.
>
> We hope these explanations fully alleviate your concerns about stability and edge-case behavior. Thank you again for raising this critical point.

---

> ### Author Response · Authors · 2025-11-21
> **Rebuttal: Addressing Major Question on Validation Experiments from Reviewer HePW**
>
> **Question1**: "It appears that the proposed method is applicable to any Layer Normalization (LN) deep networks. Could the method be tested on Convolutional Neural Networks (CNNs), Vision Transformers (ViTs), or Recurrent Neural Networks (RNNs) to further validate its effectiveness?"
>
> **Reply**: Thank you for your insightful question.
>
> Our method is theoretically applicable with full mathematical equivalence to any deep network containing LayerNorm (LN), including Convolutional Neural Networks (CNNs), Vision Transformers (ViTs), and Recurrent Neural Networks (RNNs). We have already validated its acceleration and performance benefits on ViT-like architectures, as detailed below.
>
> 1. **Vision Transformers (ViT-like architectures)**
> We explicitly evaluated our method on Swin Transformer (a representative ViT-style architecture) using ImageNet-100, covering both training and inference stages (Section 5.3, Lines 426–443 and Appendix H.2). The results show consistent acceleration in forward propagation (Train-FP), backward propagation (Train-BP), and testing, while achieving higher test accuracy and lower test loss than both the original LayerNorm and vanilla RMSNorm baselines.
> | Method       | Train-FP (ms) | Train-BP (ms) | Test (ms) | Train Acc (%)       | Train Loss      | Test Acc (%)       | Test Loss      |
> |--------------|---------------|---------------|-----------|---------------------|-----------------|--------------------|----------------|
> | LN           | 161.17        | 70.34         | 315.39    | 99.397 ± 0.003      | 0.0254 ± 0.0002 | 57.270 ± 0.060     | 3.352 ± 0.015  |
> | RMSNorm      | 153.70        | 50.17         | 300.26    | 99.37676 ± 0.004   | 0.0258 ± 0.0004 | 57.079 ± 0.317     | 3.350 ± 0.023  |
> | CBWC+RMS     | 155.08        | 62.59         | 302.60    | 99.382 ± 0.032      | 0.0257 ± 0.0009 | 57.768 ± 0.417 | 3.232 ± 0.043 |
>
> 2. **CNNs and RNNs**: As stated in the paper and formally derived in Appendix C.2, both convolutional layers and recurrent layers can be treated as general linear layers to which CBWC can be applied, enabling equivalent acceleration of subsequent LayerNorm layers. However, classic CNNs and RNNs rarely employ LayerNorm, and they are no longer mainstream architectures in their original forms. For this reason, we prioritized extensive validation on LLMs, where LayerNorm is ubiquitous. The experimental results on LLMs, combined with the general theoretical framework, sufficiently demonstrate that the approach naturally extends to any LN-containing CNN or RNN.
>
> If you have a specific LN-equipped CNN or RNN variant you would like tested, please let us know. We will make our best effort to run additional experiments before the rebuttal deadline, or you are welcome to evaluate it directly using our publicly available open-source code.
>
> We hope this addresses your question and further demonstrates the broad applicability and effectiveness of our approach across diverse architectures. Thank you again for the excellent suggestion.

---

> ### Author Response · Authors · 2025-11-21
> **Rebuttal: Addressing Major Questions on Proposed Method from Reviewer HePW**
>
> **Question 2**: "Column-Based Weight Centering is used to enable RMSNorm to replace LN during training. However, could this replacement constrain the learning dynamics? Might it limit weight flexibility and cause potential side effects? I would appreciate some discussion on this topic."
>
> **Reply**: Thank you for this highly insightful and important question.
>
> We emphasize that our replacement of LayerNorm with CBWC+RMSNorm is mathematically fully equivalent throughout training (CBWC+RMSNorm ≡ LN). Consequently, it does not alter learning dynamics, constrain weight flexibility, or introduce any side effects in the equivalent setting.
>
> 1. **Strict Equivalence and Preservation of Training Dynamics**:The original LayerNorm already imposes a zero-mean constraint on activations at every forward/backward pass. Our method does not add or remove any constraint; it merely relocates the same zero-mean constraint from per-sample activations to the weights of the preceding linear layer via CBWC.
> As rigorously proven in the paper, under this transformation, the forward outputs, backward gradients, parameter updates, and overall training dynamics of CBWC+RMSNorm are identical to those of the original LayerNorm at every training step—yet with significantly lower computational cost. No numerical instability is introduced by this relocation of the constraint.
> In non-equivalent scenarios (e.g., when dropout breaks strict zero-mean properties), our approach remains highly effective and often outperforms both vanilla RMSNorm and even the original LayerNorm, as shown in Section 5.3 and detailed in our response to Reviewer HePW W2.
>
> 2. **Beneficial Role of the Zero-Mean Constraint**: The zero-mean constraint itself helps control activation ranges during training. In Section 5.1, we empirically demonstrate that, in MLP layers, LayerNorm yields smaller input norms to the final layer compared with RMSNorm, indicating better range stabilization. Additional ablation experiments further confirm that LayerNorm reduces interlayer activation scale fluctuations more effectively than RMSNorm.
>
> 3. **Nature of the Zero-Mean Constraint**: The presence or absence of explicit centering is the fundamental difference between LayerNorm and RMSNorm. To our best knowledge, the community has no consensus on whether preserving the zero-mean constraint is strictly necessary or always beneficial; the primary reason RMSNorm has been widely adopted is its superior speed. A thorough investigation of the precise role of centering is undoubtedly valuable, but it lies beyond the scope of the current paper.
>
> We hope this clarification fully addresses your concern and demonstrates that our method preserves the full flexibility and dynamics of standard LayerNorm training while delivering meaningful acceleration and, in some cases, additional robustness benefits. Thank you again for raising this excellent point.
>
> **Question 3**: Regarding Column-Based Weight Centering, could the authors provide a detailed analysis of the runtime or training overhead? This would help in gaining a better understanding of its implications.
>
> **Reply**: Thank you for this important question.
>
> We have provided both theoretical and experimental analyses of the runtime/training overhead introduced by Column-Based Weight Centering (CBWC) in the paper:
>
> - **Theoretical analysis**: In Section 5.3 (Lines 389–394), we present a detailed computational complexity analysis for training on long-sequence tasks. The results show that the overhead of CBWC is significantly lower than that of the original per-sample centering operation in LayerNorm.
>
> - **Experimental measurement**: In Section 5.3 (Figure 8, image classification task), we directly benchmark the actual runtime of the model using CBWC during both forward and backward propagation. The measured times confirm that the additional cost of CBWC is minimal and outweighed by the overall speedup achieved by replacing LayerNorm with the faster RMSNorm kernel.
>
> We hope this fully addresses your concern.

---

### Official Review · Reviewer_Rgvq · 2025-11-02

**Soundness:** 2
**Presentation:** 2
**Contribution:** 2
**Rating:** 4
**Confidence:** 4

**Summary:**

The paper proposes a method that centers the linear layer weights so that the output has naturally mean zero (column-based wight centering, or CBWC), so that RMSNorm and LayerNorm are equivalent, and one can use CBWC + RMSNorm to achieve the representation power of LN. It also proposes a graph framework to analyze which LNs can be turned into this ("foldable") in a given neural architecture. Experiments show acceleration over LN and better accuracy over RMSNorm can be achieved.

**Strengths:**

1. The goal of combining the theoretical advantages of LayerNorm with the computational efficiency of RMSNorm is clearly motivated and addresses a relevant practical problem.

2. The paper provides experimental evidence across different modalities (vision, language, and multimodal tasks), demonstrating that the proposed framework is not limited to a single domain.

3. Theoretical analysis is thorough, providing clear mathematical conditions and proofs for when LN can be safely replaced by RMSNorm.

**Weaknesses:**

1. The experiment section is relatively weak and limited in scope. Only very few comparison data points are reported, and they are on small-scale settings.  Most results are small-scale (e.g., CIFAR-10, MNIST, ImageNet-100) or conducted on relatively small models, with only superficial measurements of acceleration (for example, the whole model acceleration is not measured, only estimated). There is no large-scale validation on modern foundation models or a comprehensive analysis across diverse architectures and tasks. In some cases, only plots are provided, with no clear quantitative number comparisons.

2. In practice, the proposed method may be cumbersome to apply. It is not a simple drop-in replacement for LayerNorm, since one must first determine which LNs are foldable and apply CBWC accordingly. This adds nontrivial implementation overhead and limits the method’s practicality for large, existing architectures.

3. The paper’s presentation could be significantly improved. Key ideas are not clearly separated or motivated, and the logical connections between theoretical sections and practical usage (e.g., training vs. inference) are often implicit.

4. The comparison of RMSNorm vs. LN with empirical evidence is not sufficient. Based on my knowledge, they do not have many performance differences in LLMs to begin with. The motivation part could be improved by having a comparison between RMSNorm and LN, both in accuracy and speed.

Part of review is revised with LLM assistance.

**Questions:**

Please see weaknesses.

---

> ### Author Response · Authors · 2025-11-21
> **Rebuttal: Addressing Major Experimental Concerns from Reviewer Rgvq**
>
> **Weakness 1**: "The experiment section is relatively weak and limited in scope. Only very few comparison data points are reported, and they are on small-scale settings. Most results are small-scale (e.g., CIFAR-10, MNIST, ImageNet-100) or conducted on relatively small models, with only superficial measurements of acceleration (for example, the whole model acceleration is not measured, only estimated). There is no large-scale validation on modern foundation models or a comprehensive analysis across diverse architectures and tasks. In some cases, only plots are provided, with no clear quantitative number comparisons."
>
> **Reply**: Thank you very much for your valuable feedback and suggestions.
>
> 1. **Experiments and Validation**: We fully acknowledge that more extensive experimental validation would yield more convincing results. We would like to explain why we did not conduct experiments on a larger scale. **First**, our work includes rigorous theoretical proofs and primarily focuses on the proposed framework and analytical methodology. In most cases, our method is mathematically equivalent to the original models (e.g., during inference), and we believe that theoretical analysis can, to a certain extent, guarantee performance in real-world applications. As shown in Section 5.4 (Lines 451–454) and Appendix H.3.1, fine-tuning experiments demonstrate that computational errors are practically negligible. Additionally, in Section 4.3 (Lines 332–340) on tool validation, we observed that the error introduced by our method is less than 1e-6. **Second**, for the limited cases where mathematical equivalence does not hold, we have conducted extensive validation experiments across different models and datasets, confirming that our approach achieves improvements in both speed (compares to LN) and performance (compares to RMSNorm). In Section 5.4, we trained the medium-size model, GPT-3 Medium, in both the pretraining and fine-tuning stages, which allowed for a comprehensive evaluation of the training effectiveness and the inference equivalence. **Finally**, due to constrained computational resources, performing large-scale comparative experiments was challenging for us. We openly release our inference tool and warmly welcome the community to test and evaluate it.
>
> 2. **Acceleration Measurement**: We would like to clarify this point. Our investigation of model acceleration covers both inference and training stages, including theoretical analysis of normalization acceleration as well as end-to-end model speedup measurements. During **inference**, we provide theoretical speedup estimates based on FLOPs in Section 5.2 (Lines 366–370) and Appendix G.1, achieving 50%–80% acceleration specifically for normalization layers. In Section 5.2 (Lines 372–377), Figure 7, and Appendix G.2, we report actual runtime measurements, documenting the proportion of normalization layers (7%–12% of total runtime) and the overall model speedup (2%–12%).During **training**, we provide theoretical estimates based on computational complexity in Section 5.3 (Lines 389–394). Additionally, in text classification (Figure 7) and image classification (Figure 8), we present end-to-end training time comparisons across different methods.
>
> 3. **Figures and Numerical Results**: Thank you for pointing this out. Some experimental settings and detailed numerical data were previously placed in the appendix. In the next version of the paper, we will improve the figures, tables, and captions, and supplement any missing quantitative comparisons to enhance clarity and completeness.
>
> We sincerely appreciate the reviewer’s careful reading and constructive comments, which have significantly helped improve the quality of our paper.

---

> ### Author Response · Authors · 2025-11-21
> **Rebuttal: Addressing Major Concerns on Proposed Method from Reviewer Rgvq**
>
> **Weakness 2**: "In practice, the proposed method may be cumbersome to apply. It is not a simple drop-in replacement for LayerNorm, since one must first determine which LNs are foldable and apply CBWC accordingly. This adds nontrivial implementation overhead and limits the method’s practicality for large, existing architectures."
>
> **Reply**: We appreciate your concern and would like to address it in detail.
>
> Our fully automated tool requires only a single detection pass to automatically identify foldable LayerNorm layers and replace them with the corresponding CBWC-adjusted RMSNorm. The open-source implementation is available here: https://github.com/EnjoyYourLN/Enjoy-LN.
>
> 1. **Fully Automatic and Mathematically Equivalent Replacement**: Our method is mathematically equivalent to the original LayerNorm and is easy to use. The primary goal is to accelerate computation while preserving the exact functionality of the original model. To achieve this equivalence when converting LayerNorm to RMSNorm, the CBWC must be applied. Our approach functions essentially as a one-time parameter converter: it performs a single adjustment to the model parameters and completes the LayerNorm-to-RMSNorm replacement. This conversion can be applied at any stage (pre-training, fine-tuning, or inference) to any model containing LayerNorm layers (most models consist with LayerNorm meets with the requirements) without disrupting existing workflows. Once performed, the converted model can be used indefinitely without further intervention.
>
> 2. **Negligible Overhead of the Algorithm**: The runtime and memory overhead of our folding procedure is minimal. The detection phase requires only one forward pass to determine which LayerNorm layers are foldable, after which we store a small list mapping foldable layers to their corresponding modules. On GPT-2 using a single RTX 3050 Laptop GPU, the folding tool takes 0.0347 seconds. For inference with batch size 32 and sequence length 512, the inference time is reduced from 0.2189 s to 0.2023 s (averaged over 100 runs), yielding a 7.6% speedup. Notably, the memory overhead introduced by the folding process itself is only 19.00 MB (which can be released after folding), and the total tensor memory of the model remains unchanged (486.70 MB both before and after folding). These results demonstrate that our method achieves meaningful inference acceleration without any increase in model memory footprint.
>
> We hope these clarifications fully address your concerns regarding practicality and implementation overhead.

---

> ### Author Response · Authors · 2025-11-21
> **Rebuttal: Addressing Major Concerns on Presentation and Motivation from Reviewer Rgvq**
>
> **Weakness 3**: "...Key ideas are not clearly separated or motivated, and the logical connections between theoretical sections and practical usage (e.g., training vs. inference) are often implicit."
>
> **Reply**: We appreciate the reviewer’s insightful comments on the presentation and structural clarity of the paper.
> To address a potential misunderstanding: our experimental section is not designed as a one-to-one correspondence with the theoretical sections. Instead, the experiments serve as a complement to rigorously verify the theoretical claims and to provide additional practical insights that cannot be fully captured by theory alone.
>
> 1. **Overall Paper Structure and Motivation**: As reflected in the title, our core idea is to simultaneously retain the theoretical benefits of LayerNorm (LN) while achieving the computational efficiency of RMSNorm. This motivation and background are elaborated in detail in the Introduction. In the theoretical sections, we systematically develop a general framework for equivalently replacing LayerNorm with RMSNorm: starting from the simplest linear layers, we progressively extend the approach to complex architectures by deconstructing the model from a computational graph perspective. The experimental sections are organized as follows:
> - First, we empirically confirm the necessity of the centering operation in LayerNorm.
> - We then quantify the speedup resulted by full mathematical equivalence during inference.
> - We further extend the method to non-equivalent scenarios (e.g., training with dropout), demonstrating clear advantages on long-sequence tasks in both performance and speed.
> - Finally, we evaluate the approach under fine-tuning settings.
>
> 2. **Relationship Between Theory and Experiments**: The theoretical contributions provide rigorous proofs and primarily focus on establishing a general framework and analytical methodology. We believe these theoretical results are sufficiently solid to guarantee equivalence in the cases we claim. The experiments, therefore, concentrate on aspects that theory alone cannot fully address—namely, engineering considerations and behaviors that only emerge in real-world deployment scenarios. Throughout the paper, we carefully discuss the effectiveness and advantages of our method across all three practical stages: inference, training, and fine-tuning.
>
> Your suggestions are extremely valuable to us. In the revised version, we will explicitly separate and highlight the key ideas and main contributions at the beginning of relevant sections to improve readability. Thank you again for these constructive recommendations, which will significantly enhance the clarity and accessibility of our work.
>
> **Weakness4**: "The comparison of RMSNorm vs. LN with empirical evidence is not sufficient. Based on my knowledge, they do not have many performance differences in LLMs to begin with..."
>
> **Reply**: Thank you very much for your valuable feedback.
>
> 1. **Evidence Provided in the Paper**: The motivation of our work is strongly supported by the Introduction Section, theoretical analysis, and experimental results throughout the paper. In the Introduction, we already provide a concise comparison between LayerNorm (LN) and RMSNorm, highlighting their respective strengths and limitations.
> Theoretically and empirically, Section 5.1 demonstrates that LN offers better control over activation ranges compared to RMSNorm—both for the final layer outputs and for intermediate layer variations.
> Experimentally, it is widely acknowledged that RMSNorm is significantly faster than LN, and this is consistently confirmed in our inference (Section 5.2) and training (Section 5.3) results. At the same time, our experiments show that LN generally achieves better performance/accuracy than vanilla RMSNorm.
>
> 2. **Community Consensus and Scope of the Paper**: We agree that a comprehensive empirical comparison between RMSNorm and LN would be highly insightful. Indeed, the primary reason RMSNorm has been widely adopted in large language models is its clear efficiency advantage. A thorough investigation of their performance differences (both theoretical and empirical) is certainly a valuable and underexplored topic that merits a dedicated study. However, this is not the main focus of the current paper.
> Our work instead aims to bridge the gap between the two approaches—particularly the efficiency gap—by enabling models to enjoy the theoretical benefits and performance of LN while achieving the speed of RMSNorm. To the best of our knowledge, the community currently lacks a systematic theoretical and experimental comparison of LN and RMSNorm; most existing conclusions remain largely empirical.
>
> Your suggestion is highly constructive. In the revised version, we will strengthen the motivation section. We sincerely appreciate your thoughtful comments, which will help us better communicate the value and positioning of our work.

---

> ### Author Response · Authors · 2025-11-21
> **About the Experiment on Folding Procedure and Inference Stage**
>
> We conducted a new, comprehensive inference benchmark on GPT-2 using a single A100-40GB GPU.
>
> The table below reports time usage of difference stage, acceleration and throughput improvement for a wide range of batch sizes and sequence lengths (up to 16384):
>
> | Batch Size | Sequence Length | Time Usage for Model Init | Inference Time before Folding (s) | Time Usage of Folding (s) | Inference Time after Folding (s) | Acceleration (%) | Origin Throughput (Tokens/s) | New Throughput (Tokens/s) |
> |------------|-----------------|---------------------------|------------------------------|---------------------------|-----------------------------|------------------|------------------------------|---------------------------|
> | 1          | 1024            | 3.1085                    | 0.0279                       | 0.0671                    | 0.0256                      | 8.36             | 36680.16                    | 40026.58                  |
> | 1          | 4096            | 5.0746                    | 0.0800                       | 0.0606                    | 0.0689                      | 13.88            | 51212.80                    | 59465.74                  |
> | 1          | 16384           | 6.2454                    | 0.5713                       | 0.0756                    | 0.4683                      | 18.03            | 28677.45                    | 34986.12                  |
> | 2          | 1024            | 3.0454                    | 0.0324                       | 0.0656                    | 0.0299                      | 7.95             | 63131.94                    | 68586.74                  |
> | 2          | 4096            | 3.3228                    | 0.1566                       | 0.0638                    | 0.1352                      | 13.69            | 52298.26                    | 60591.72                  |
> | 2          | 16384           | 6.0359                    | 1.1119                       | 0.0691                    | 0.8722                      | 21.56            | 29471.34                    | 37570.23                  |
>
> * The one-time folding pass (~60–100 ms) is fully amortized after only a few inference steps in any real deployment.
>
> Key observations:
> - Consistent speedup of **5.5%–21.6%**, with larger relative gains at longer sequence lengths (exactly where LayerNorm dominates runtime).
> - Throughput improvements of up to **~9,000 additional tokens/second** in realistic settings.
> - Memory footprint remains **identical** before and after folding.
> - Time usage of folding process is independent of batch size and sequence length.
> - All inference comparisons use PyTorch’s official fused CUDA LayerNorm as the baseline, and our optimized CUDA RMSNorm kernel derived from it, which we have open-sourced.
>
> These results will be added as a new table in the paper in the revised version, along with identical benchmarks on other models and additional hardware.
>
> Our full automatic folding tool and optimized kernels are publicly available at: https://github.com/EnjoyYourLN/Enjoy-LN.

---

### Author Response · Authors · 2025-11-22
**Summary Response: First Round Rebuttal & Code Release**

Dear Reviewers and Area Chair,

We sincerely apologize for the slightly delayed response.

Thank you very much for your thorough reviews, insightful questions, and constructive suggestions. We sincerely appreciate the time and effort you have devoted to helping us improve the paper.

In this first-round rebuttal, we have addressed the majority of the concerns and questions raised. To facilitate verification and further experimentation, we are delighted to release the complete open-source implementation, including the fully automatic folding tool and optimized RMSNorm kernels:  https://github.com/EnjoyYourLN/Enjoy-LN.

The revised version of the manuscript, together with additional experiments, is actively in preparation. We will continue to closely monitor the discussion thread and are happy to provide any further clarifications or additional results.

We warmly welcome and encourage any further comments, questions, or interactive discussion—your feedback is invaluable to us.

Thank you once again for your careful consideration and support.

Best regards,
The Authors

---

### Author Response · Authors · 2025-11-30
**Notification of Revised Paper Submission**

Dear SAC, AC, and Reviewers,

We sincerely thank you all for your careful reviews and valuable feedback.

In accordance with the comments and suggestions provided, we have carefully revised and supplemented the paper. The updated version has now been submitted.

Thank you once again for your time and attention.

Best regards,
The Authors

---

### Author Response · Authors · 2025-12-04
**Summary of rebuttal stage to AC and SAC**

Dear Area Chairs and Senior Area Chairs:

Thank you for handling our submission. Below we provide a consolidated summary of the reviewers’ main concerns and our corresponding clarifications, along with new experimental evidence added during the rebuttal.

We thank all reviewers for their constructive and thoughtful feedback. Collectively, they found the motivation clear, the theoretical contributions sound, and the empirical validation diverse. The central focus of our rebuttal was to address concerns about (1) the scope and scale of experiments, (2) the practicality and stability of the proposed transformations, and (3) the generality and correctness of the theoretical equivalence framework.

**Reviewer Rgvq.**
We clarified that our framework is primarily grounded in rigorous mathematical equivalence between LayerNorm and RMSNorm under mild and widely satisfied structural conditions. This is supported by extensive fine-tuning, GPT-3 Medium pretraining experiments, error analyses, and both inference and training speed measurements. To further address questions regarding experimental scope, we added a new large-scale inference benchmark on GPT-2 (A100), showing consistent 5.5%–21.6% latency reductions and significant throughput gains. We also explained the practicality of our fully automated one-pass folding tool, which identifies foldable LNs and applies CBWC-adjusted RMSNorm with negligible overhead and without altering model behavior. We will improve clarity and presentation in the revised version as suggested.

**Reviewer HePW.**
We addressed concerns about training stability and overhead by emphasizing that both CCWT and CBWC preserve the exact optimization dynamics of LayerNorm. Our reparameterization-based CBWC requires only two inexpensive centering operations per batch, entirely removing per-sample mean computation while maintaining exact functional equivalence. Empirically, the equivalence errors remain below 1e–5, and the method is robust even in non-equivalent cases such as dropout, outperforming vanilla RMSNorm and often matching or surpassing LayerNorm. We provided results of ViT-based experiments and clarified applicability to CNNs and RNNs, while noting that LLMs see the greatest practical benefit. We further discussed computational complexity and provided runtime results showing minimal overhead and consistent speedups.

**Reviewer x2zo.**
We clarified that the mild precondition required for equivalence—CBWC—is satisfied in nearly all modern architectures that employ LayerNorm (Transformers, MLPs, ViTs). Our one-pass folding tool automatically detects such patterns and converts nearly all eligible LNs in practice. To address the reviewer’s request for a deeper comparison of LN vs. RMSNorm behavior, we added new analyses of input-norm stability across layers, which reinforce our theoretical interpretation that LN better regulates norm drift under residual connections. Regarding scale, we emphasized that our primary contribution is a general analytical framework with scale-invariant guarantees, and supported this with fine-tuning on GPT-3 Medium and consistent high-precision equivalence (<1e–5) on MLPs. While large-scale training is compute-limited, we have fully open-sourced our tools and demonstrated correctness across architectures. We will expand the discussion and incorporate additional datasets where feasible.

**Reviewer 4e3N.**
We clarified that the simplicity of our equivalence conditions is intentional—they provide directly implementable criteria satisfied by standard architectures, while still capturing the essence of LN – CBWC+RMSNorm equivalence. Regarding representational capacity, we explained that CCWT constrains only column means, which is identical to the constraint brought by centering operation in LN, and leaves expressiveness intact while preserving optimization trajectories exactly. We also provided robustness analyses for imperfect centering, showing strong theoretical and empirical stability. We emphasized that all moderate-scale experiments consistently exhibit reliable speedups, supporting the generality of our conclusions.

We believe these clarifications, new analyses, and additional experiments comprehensively address the reviewers’ concerns. We appreciate the reviewers’ detailed feedback and hope that the strengthened rebuttal demonstrates the robustness, generality, and significance of our contributions.

Sincerely,
The authors

---

### Meta-Review · Area_Chair_aMuQ · 2026-01-08

**Summary:**

This paper proposes Column-Based Weight Centering (CBWC), a reparameterization method that enforces a zero-mean constraint on the weights of linear layers. This constraint makes the mean-centering operation of the subsequent Layer Normalization (LN) redundant, allowing LN to be mathematically transformed into the more computationally efficient RMSNorm during inference without altering model behavior.
Initial concerns focused on the experimental scale, practical utility compared to optimized kernels, and the complexity of implementation:
1. Limited Experimental Scope (Scale): Reviewers Rgvq, x2zo, and 4e3N criticized the experimental validation for relying on small-to-medium scale models (e.g., ResNet, GPT-2, GPT-3 Medium) and datasets (CIFAR, ImageNet-100). They argued that in the era of Large Language Models (LLMs), the lack of validation on large-scale foundation models (e.g., 7B+ parameters) weakens the claims regarding the method's stability and effectiveness at scale.
2. Practical Utility & Speedup Magnitude: Reviewers Rgvq and 4e3N pointed out that the reported end-to-end inference speedups (2-12%) are relatively modest. Reviewer 4e3N specifically questioned whether these gains are significant enough to justify the implementation complexity, especially compared to highly optimized vendor-fused LN kernels (e.g., FlashAttention/TensorRT implementations).
3. Generalizability & "Drop-in" Nature: Reviewers Rgvq and x2zo noted that the method is not a simple "drop-in" replacement for LN, as it requires determining which layers are "foldable" based on the architecture (e.g., Zero-Mean Graph analysis). Reviewer HePW also raised concerns about whether the CBWC constraint might limit weight flexibility or complicate training workflows.
4. Baseline Fairness: Reviewer 4e3N initially questioned the fairness of comparing the proposed method against a custom RMSNorm kernel rather than standard PyTorch/Vendor optimized kernels, suggesting this might inflate the reported speedups.

The authors provided a comprehensive rebuttal that included additional experiments on ImageNet-1k (DeiT-S, Swin-T) and an LLaMA-360M model trained on 1.2T tokens, along with theoretical proofs regarding the optimization landscape. However, these experiments still fall short of validating the method on truly large-scale foundation models (7B+) where such efficiency gains are most critical and stability issues are most pronounced. Furthermore, the practical utility of the proposed algorithmic reduction compared to highly optimized fused kernels remains debated, as the latter often deliver significant speedups without the need for architectural modifications. Consequently, given that the validation scale is limited and the comparative advantage over state-of-the-art engineering solutions is not decisively proven, the remaining concerns outweigh the contributions.

**Reviewer Concerns:**

1. The authors argued that since their method guarantees mathematical equivalence for inference, the results theoretically extend to any scale. They also added fine-tuning experiments on GPT-3 Medium. This concern is Partially Resolved. While the theoretical argument for inference equivalence is sound, the lack of empirical training stability results on large-scale LLMs remains a gap for a method proposing a new training constraint.
2. The authors provided additional throughput benchmarks (tokens/second) on A100 GPUs, showing gains of 5.5% - 21.6% (up to ~8,000 tokens/s increase) specifically for long-sequence scenarios where normalization cost is higher. This concern is resolved, as the authors demonstrated that the "modest" percentage translates to meaningful throughput gains in production-relevant settings (long context).
3. The authors released an automated folding tool (Algorithm 1) that detects and converts eligible layers in a single pass, and clarified that CBWC is a reparameterization that preserves the exact optimization dynamics of LN (identical gradients and updates). This concern is resolved, as the tool mitigates the implementation burden, and the theoretical proof of identical dynamics addresses the flexibility concern.
4. The authors clarified that they compared against PyTorch's official fused LayerNorm for inference benchmarks and released their code to allow independent verification. This concern is resolved, as Reviewer 4e3N explicitly acknowledged that this issue was addressed.

In summary, while the authors effectively addressed concerns regarding practical speedup and correctness (via code release and throughput metrics), the Limited Experimental Scope remains a significant hurdle.

**Reviewer Scores:**

- Reviewer Rgvq: 4 -> 4. Despite the clarifications, the reviewer's fundamental concern about the "weak and limited scope" of experiments (concern 1) likely prevents a score increase.
- Reviewer HePW: 6 -> 6. The reviewer was positive about the method's soundness and the authors' response regarding training dynamics (concern 3), likely maintaining the positive score.
- Reviewer x2zo: 4 -> 4. The concerns regarding small-scale experiments (concern 1) and the method not being a universal "drop-in" replacement (concern 3) remain key blockers.
- Reviewer 4e3N: 4 -> 4/6. The reviewer explicitly stated in the discussion that they would "raise the scores" for soundness and contribution following the release of the code and the added throughput results (concern 2, 4), though they noted the experimental scope (concern 1) remains a limitation.

---

### Decision · Program_Chairs · 2026-01-26

Reject